

# Abundances of isotopologues and calibration of $CO_2$ greenhouse gas measurements

Pieter P. Tans[1,3], Andrew M. Crotwell[2,3], and Kirk W. Thoning[1,3]

[1]Global Monitoring Division, Earth System Research Laboratory, National Oceanic and Atmospheric Administration, Boulder, Colorado, 80305, USA.
[2]Cooperative Institute for Research in Environmental Sciences, University of Colorado, Boulder, Colorado, 80309, USA.
[3]Central Calibration Laboratory, World Meteorological Organization Global Atmosphere Watch program (WMO/GAW)

*Correspondence to:* Pieter Tans (Pieter.Tans@noaa.gov)

**Abstract**

We have developed a method to calculate the fractional distribution of $CO_2$ across all of its component isotopologues based on measured $\delta^{13}C$ and $\delta^{18}O$ values. The fractional distribution can be used with known total $CO_2$ to calculate each component isotopologue individually, in units of mole fraction. The technique is applicable to any molecule where isotopologue-specific values are desired. We used it with a new $CO_2$ calibration system to account for isotopic differences among the primary $CO_2$ standards that define the WMO X2007 $CO_2$ in air calibration scale and between the primary standards and standards in subsequent levels of the calibration hierarchy. The new calibration system uses multiple laser spectroscopic techniques to measure amount of substance fractions (in mole fraction units) of the three major $CO_2$ isotopologues ($^{16}O^{12}C^{16}O$, $^{16}O^{13}C^{16}O$, and $^{18}O^{12}C^{16}O$) individually. The three measured values are then combined into total $CO_2$ (accounting for the rare unmeasured isotopologues), $\delta^{13}C$, and $\delta^{18}O$ values. The new calibration system significantly improves our ability to transfer the WMO $CO_2$ calibration scale with low uncertainty through our role as the World Meteorological Organization Global Atmosphere Watch Central Calibration Laboratory for $CO_2$. Our current estimates for reproducibility of the new calibration system are $\pm\,0.01$ μmol mol$^{-1}$ $CO_2$, $\pm\,0.2$‰ $\delta^{13}C$, and $\pm\,0.2$‰ $\delta^{18}O$, all at 68% confidence interval (CI).

## 1 Introduction

Long-term atmospheric monitoring of the greenhouse gases relies on a stable calibration scale to be able to quantify small spatial gradients and temporal trends. Small changes in trends and spatial gradients result from realignments in the strengths of emissions ("sources") and removals ("sinks") of the greenhouse gases. Inconsistent scale propagation to atmospheric measurements would give biased results from one monitoring station or network to the next that would be attributed incorrectly to sources/sinks by atmospheric transport models. Preventing biased results from various national monitoring networks enables improved understanding of the carbon cycle and its response to human intervention and climate change. It has now become even more important as countries have pledged



emissions reductions. The capability to independently and transparently verify emission reductions could be helpful for creating trust in the agreements.

The World Meteorological Organization Global Atmosphere Watch (WMO GAW) program facilitates cooperation and data sharing among participating national monitoring programs. Atmospheric data collected over small regional
scales is difficult to interpret without global coverage that provides boundary conditions and also insight into influences outside of the region. WMO GAW sets stringent compatibility goals so measurements from independent laboratories can be combined in scientific studies. This greatly enhances the value of the individual data sets since it allows processes occurring within the region to be better distinguished from processes external to that region. In combining data sets it is imperative that systematic biases between the monitoring networks be small enough that
they do not influence scientific interpretation of patterns and strengths of sources and sinks. For $CO_2$, the consensus of the scientific community is that network biases should be below 0.1 $\mu$mol mol$^{-1}$ in the Northern Hemisphere but less than 0.05 $\mu$mol mol$^{-1}$ in the Southern Hemisphere where atmospheric gradients are smaller (WMO, 2016). One initial requirement to accomplishing this network compatibility goal is that measurements are comparable, that is each independent laboratory uses a single common calibration scale. The use of a single calibration scale makes
spatial gradients and temporal changes insensitive to large components in the full uncertainty budget of the scale itself. The calibration scale must be maintained indefinitely to ensure that measurements from various organizations are compatible and that measurements over long time scales can be directly compared to infer rates of changes. The WMO GAW has designated a single laboratory as the Central Calibration Laboratory (CCL) whose mission is to maintain a stable reference scale over time and to disseminate it to other organizations with very low uncertainty
(WMO, 2016).

The WMO X2007 $CO_2$ in air calibration scale is maintained and propagated by the National Oceanic and Atmospheric Administration, Earth System Research Laboratory, Global Monitoring Division (NOAA) in its role as the WMO GAW CCL for $CO_2$. The scale is defined by 15 primary standards covering the range 250 – 520 $\mu$mol mol$^{-1}$. The primary standards are modified real air standards made in the early 1990's by filling cylinders with dried
($H_2O$ < 2 $\mu$mol mol$^{-1}$) natural air at Niwot Ridge, CO, USA, a remote site at approximately 3040 masl in the Rocky Mountains. It typically is exposed to clean tropospheric air and is only occasionally influenced by local sources. $CO_2$ abundances of the primary standards were adjusted either by scrubbing $CO_2$ from a portion of the natural air using a trap with sodium hydroxide coated silica to lower the $CO_2$ or by spiking with a mixture of $CO_2$ in air (approximately 10%) to raise it. This differs slightly from the current practice of targeting lower than local ambient $CO_2$ by diluting
with ultra high purity zero air, $CO_2$ nominally < 1 $\mu$mol mol$^{-1}$ (Scott Marrin Inc., Riverside CA, USA) (Kitzis, 2009).

The assigned values of the primary standards come from repeated (approximately every two years) manometric determinations of the primary standards. The manometer, an absolute measurement method, described fully in Zhao et al. (1997), essentially measures the $CO_2$ amount of substance fraction in units of mole fraction ($X_{CO2}$) by accurate
measurement of pressure and temperature of a whole air sample and then of pure $CO_2$ extracted from the whole air



sample in fixed volumes. The manometer is enclosed in an oven capable of maintaining a constant temperature (within ± 0.01 °C). A 6 L volume borosilicate glass bulb (the large volume) is flushed with the dried whole air sample (dew point < -70 °C) and the pressure and temperature are measured after the large volume temperature equilibrates with the oven. $CO_2$ plus $N_2O$ and trace amounts of $H_2O$ are cryogenically extracted from the whole air

sample using two liquid nitrogen cold traps. $CO_2$ and $N_2O$ are then cryogenically distilled from $H_2O$ and transferred to a ~10 mL cylindrical glass vessel (the small volume). Pressure and temperature of the small volume are measured after the oven temperature has stabilized following the transfer. The volume ratio of the small to large volumes, determined by an off-line sequential volume expansion experiment, is used with the measured temperatures and pressures to calculate the ratio of moles $CO_2$ (corrected for the $N_2O$) to total moles of air in the sample using the

virial equation of state. Uncertainty of the method is ± 0.1 µmol mol$^{-1}$ (68% CI) at 400 µmol mol$^{-1}$ (Zhao et al., 2006, Brad Hall, personal communication).

The transfer of the scale from primary to secondary standards and hence to tertiary standards (which are used as working standards by NOAA and delivered to other organizations) has been done historically using nondispersive infrared absorption spectroscopy (NDIR). The secondary standards are used to prolong the lifetime of the primary

standards. The current primary standards have been in use for nearly 25 years and provide a consistent scale over that time period. All measurements by NOAA and WMO GAW contributing programs are directly traceable to this single set of primary standards through a strict hierarchy of calibration.

The transfer of the scale from primary to secondary standards has typically been done using a subset of 3 or 4 primary standards rather than the entire set of 15 primary standards. This was done because we wanted to perform a

local curve fit of the non-linear NDIR response while also minimizing use of the primary standards. The subset of primary standards chosen was a function of the expected $CO_2$ abundance in the secondary standards and was designed to closely bracket the expected values with a small range of $CO_2$ in the primary standards. The relatively large uncertainty of the individual manometric assigned values would potentially introduce significant biases due to the use of subsets of primary standards. To prevent these biases, the individual manometrically assigned values of

the primary standards were corrected based on the residuals to a consistency fit of almost all primary standards (usually without the highest and the lowest primary) run on the NDIR. The re-assigned values (average manometer value minus the residual) were assumed to be the best assigned value for the primary standards. This in theory should allow the use of subsets of the primary standards when transferring the scale from primary to secondary. In practice, as will be shown, there are still possible biases due to the grouping of primary standards based on expected

$CO_2$ of the secondary standards. Tertiary standards were calibrated similarly against closely spaced subsets of secondary standards that bracketed the expected values of the tertiary standards.

New analytical methods developed over the last several years have greatly improved the ability of monitoring stations to measure $CO_2$. These new analytical techniques and improved diligence of monitoring network staff are pushing the uncertainties of measurements lower and improving the network compatibility. Current scale

reproducibility using the NDIR calibration system is 0.03 µmol mol$^{-1}$ (68% CI) ("Carbon Dioxide WMO Scale",



2017). This is a significant component of the targeted 0.1 µmol mol⁻¹ (or 0.05 µmol mol⁻¹ in the Southern Hemisphere) network compatibility goal (WMO, 2016). Improvements in the scale propagation uncertainty would help monitoring programs achieve the compatibility goals. We have therefore undertaken to improve our calibration capabilities and to address key uncertainty components of the scale transfer. These key components are the

reproducibility of the scale transfer, the potential for mole fraction dependent biases, and of most importance to this paper the potential issues relating to the isotopic composition of the primary standards and subsequent standards in the calibration hierarchy.

## 2 Isotopic influence on CO₂ measurement

The WMO CO₂ mole fraction scale is defined as the number of molecules of CO₂ per mole of dry air, without regard to its isotopic composition. An isotopologue of CO₂ has a specific isotopic composition. The five most abundant CO₂ isotopologues, in order of abundance, are: $^{16}O^{12}C^{16}O$, $^{16}O^{13}C^{16}O$, $^{18}O^{12}C^{16}O$, $^{17}O^{12}C^{16}O$ and $^{18}O^{13}C^{16}O$ (referred to in equations in this work by 626, 636, 826, 726, and 836 respectively). For CO₂ the two oxygen positions are equivalent due to the symmetry of the molecule so the position of the oxygen isotopes does not matter. The

abundance of the radioactive $^{14}C$ relative to $^{12}C$ is ~10⁻¹²; which is too small to be of significance in this context. Analysts need to take into account differences in the relative sensitivity of their analyzers to different isotopologues (or isotopomers, see below) as well as differences in the isotopic composition of sample and standard gases.

Isotopic composition is typically measured by isotope ratio mass spectroscopy (IRMS) and is reported as the difference in the minor isotope to major isotope ratio (i.e. $^{13}C/^{12}C$) from the ratio of an accepted standard reference

material, typically in units of per mil (‰). For example, the $^{13}C$ isotopic value ($\delta^{13}C$) is defined as:

$$\delta^{13}C = \frac{\left(^{13}C/_{12}C\right)_{Sample} - \left(^{13}C/_{12}C\right)_{Standard}}{\left(^{13}C/_{12}C\right)_{Standard}} * 1000 \qquad (1)$$

Where $(^{13}C/^{12}C)_{sample}$ and $(^{13}C/^{12}C)_{standard}$ are the $^{13}C$ to $^{12}C$ ratios for the sample and the standard reference material respectively. The international accepted scale for $^{13}C$ is the Vienna Pee Dee Bellemnite (VPDB) scale, realized as calcium carbonate. Oxygen isotopic ratios ($^{18}O/^{16}O$ or $^{17}O/^{16}O$) in CO₂ are described with a similar delta notation relative to an accepted reference material. For many applications, the $^{17}O$ isotope is not actually measured but is

assumed to follow a mass dependent relationship with $^{18}O$ where $\delta^{17}O \approx 0.528 * \delta^{18}O$. This approximation is adequate for the purpose of defining the oxygen isotopic effects on atmospheric CO₂ measurements. For more detailed descriptions of this relationship see Santrock et al. (1985), Assonov and Brenninkmeijer (2003), Brand et al. (2010) and references therein. Oxygen isotopes can be related to either Vienna Standard Mean Ocean Water

(VSMOW) or to VPDB-CO₂, with the latter commonly used in the atmospheric CO₂ community. The VPDB-CO₂



scale relates to the $CO_2$ gas evolved from the calcium carbonate material itself during the reaction with phosphoric acid and accounts for oxygen fractionation that occurs during the reaction (Swart et al., 1991). In this paper all oxygen isotope values are referenced to the VPDB-$CO_2$ scale unless otherwise noted.

$CO_2$ analysers are not equally sensitive to the isotopologues of $CO_2$. For example, gas chromatography where $CO_2$ is

reduced to $CH_4$ and detected with a flame ionization detector (GC-FID) (Weiss, 1981) is equally sensitive to all isotopologues whereas laser based absorption techniques that measure an absorption line from the single major $^{16}O^{12}C^{16}O$ isotopologue are blind to all of the minor isotopologues. NDIR instruments are much more complicated in their response to the various minor isotopologues of $CO_2$. Most NDIR analyzers use an optical band pass filter to limit the wavelengths of light reaching the detectors. These filters often exclude part of the absorption bands of the

minor isotopologues (e.g. Tohjima et al. 2009), but are more sensitive to the $^{16}O^{13}C^{16}O$ lines within the pass band because absorption of the much stronger $^{16}O^{12}C^{16}O$ lines is partially saturated. The width and shape of the transmission window of the filter is generally not identical between instruments. Tohjima et al. (2009) found significant differences in the sensitivity to the minor isotopologues between three different LI-COR NDIR analyzers. In addition, Lee et al. (2006) found the response of a Siemens ULTRAMAT 6E NDIR analyzer to be almost

completely insensitive to the minor isotopologues.

The range of $\delta^{13}C$ and $\delta^{18}O$ encountered in the background atmosphere (~ -7.0 to -9.0‰ $\delta^{13}C$ and 2 to -2‰ $\delta^{18}O$) is too small to cause a significant bias on the total $CO_2$ measurements with any of these techniques. At 400 μmol mol$^{-1}$ total $CO_2$, neglecting $\delta^{13}C$ values leads to errors of 0.0044 μmol mol$^{-1}$ per 1‰, and neglecting $\delta^{18}O$ values leads to errors of 0.0018 μmol mol$^{-1}$ per 1‰. A problem arises however when standards with significantly different isotopic

compositions from the atmosphere are used to calibrate instruments that have partial or no sensitivity to the minor isotopologues. This occurs when standards are made from fossil fuel sourced $CO_2$ (often from combustion of natural gas) which results in significant depletion in $^{13}C$ and $^{18}O$ (Andres et al., 2000; Schumacher et al., 2011).

In the past we have neglected the dependency of the NDIR response to isotopic composition during scale transfer. The manometer measurement of the primary standards is not sensitive to isotopic composition, all isotopologues are

included in the total. However, the primary standards have a range of $\delta^{13}C$ and $\delta^{18}O$ values (-7‰ to -18‰ $\delta^{13}C$ and 0 to -15‰ $\delta^{18}O$) with higher $CO_2$ standards being more depleted due to the use in the early 1990's of a spike gas that was isotopically depleted. This probably introduced a slight bias in the results when the scale was transferred to secondary standards (often with ambient isotopic values) via NDIR measurements. It was assumed that the bias was small relative to the measurement noise in the NDIR analysis.

We intend to provide standards to the atmospheric monitoring community with isotopic values similar to the background atmosphere by using natural air whenever possible. To adjust the $CO_2$ content in the natural air standards, the current practice is to dilute using essentially $CO_2$ free natural air (ultra high purity air, Scott Marrin, Inc. Riverside CA, USA) or enrich using high $CO_2$ (10 – 20%) spike gases with $\delta^{13}C \approx$ -9‰. The $\delta^{13}C$ isotopic composition of the resulting mixture is not significantly different from ambient background air. Currently, urban air

highly enriched in $CO_2$ would have $\delta^{13}C$ values significantly lower than the spiked standards of similar $CO_2$ made


by us. However, the WMO scale is designed to track the slow isotopic depletion of background air as the global burden of $CO_2$ increases over the next decades due to burning of fossil fuels rather than approximate the composition of air influenced by local emission sources. We started using the isotopically correct spike gases in November 2011, prior to this the spike gas was fossil fuel sourced and was depleted in $\delta^{13}C$. Background

atmospheric $\delta^{18}O$ is not well matched with the current spike gases or the historical spike gases ($\delta^{18}O \approx$ -30 to -40‰) and does result in depleted $\delta^{18}O$ values in cylinders that are spiked to targeted values above local ambient values. It is also our goal to provide calibration results that incorporate a characterization of the main isotopologues and accounts for isotopic differences among the primary standards and between the primary standards and measured cylinders through the calibration hierarchy. Doing this will ensure that the transfer of the WMO scale by distributing

calibrated cylinders is not biased by isotopic differences and will provide the users of the distributed standards the information required to properly address isotopic issues when making ambient air measurements.

**3 Two different ways to define isotopic ratios and notation conventions**

In order to estimate the influence of isotopic composition differences on $CO_2$ measurements and to develop a precise

method for calibration transfer that takes isotopic composition into account we first introduce the "mole fraction" notation for isotopic ratios in molecules. The conventional definitions of atomic isotopic ratios (**r**) are:

$$^{13}\mathbf{r} \overset{\text{def}}{=} \frac{^{13}C}{^{12}C} \qquad ^{18}\mathbf{r} \overset{\text{def}}{=} \frac{^{18}O}{^{16}O} \qquad \text{etc.}$$

As used here the symbols $^{13}C$, $^{18}O$, etc. stand for abundances. It will simplify derivations below if we re-define isotopic ratios as abundance ratios relative to all carbon, oxygen, etc., similar to mole fractions in air. We give these re-defined ratios the symbol "R" instead of "**r**".

$$^{13}R \overset{\text{def}}{=} \frac{^{13}C}{^{12}C + ^{13}C} \qquad ^{18}R \overset{\text{def}}{=} \frac{^{18}O}{^{16}O + ^{17}O + ^{18}O} \quad ==> \quad ^{16}R = 1 - \frac{^{17}O + ^{18}O}{^{16}O + ^{17}O + ^{18}O}$$

These definitions lead to the following relationships:

$$^{13}R = \frac{^{13}\mathbf{r}}{1 + ^{13}\mathbf{r}} \qquad ^{13}\mathbf{r} = \frac{^{13}R}{1 - ^{13}R} \tag{2}$$

The equivalents for oxygen are:

$$^{17}R = \frac{^{17}\mathbf{r}}{1 + ^{17}\mathbf{r} + ^{18}\mathbf{r}} \qquad ^{17}\mathbf{r} = \frac{^{17}O}{^{16}O} = \frac{^{17}O/(^{16}O + ^{17}O + ^{18}O)}{^{16}O/(^{16}O + ^{17}O + ^{18}O)} = \frac{^{17}R}{1 - ^{17}R - ^{18}R} \tag{3}$$

and similarly for $^{18}R$ and $^{18}\mathbf{r}$ .





From here on we will abbreviate VPDB and VPDB-CO2 as PDB for simplicity. Using Table 1 and these conventions gives us

$$^{13}R_{PDB} = 0.0111123 \tag{4a}$$

$$^{17}R_{PDB} = 395.11\ 10^{-6} / (1+2088.35\ 10^{-6}+ 395.11\ 10^{-6}) = 394.1\ 10^{-6} \tag{4b}$$

$$^{18}R_{PDB} = 2088.35\ 10^{-6} / (1+ 2088.35\ 10^{-6}- + 395.11\ 10^{-6}) = 2083.2\ 10^{-6} \tag{4c}$$

Isotopic ratio measurements have always been expressed in terms of their (typically) small difference from the standard reference materials, in the so-called delta notation:

$$^{13}\delta \overset{\text{def}}{=} (^{13}\mathbf{r} - {}^{13}\mathbf{r}_{PDB})/{}^{13}\mathbf{r}_{PDB} = {}^{13}\mathbf{r}/{}^{13}\mathbf{r}_{PDB} - 1, \text{ so that } {}^{13}\mathbf{r} - {}^{13}\mathbf{r}_{PDB} = {}^{13}\mathbf{r}_{PDB}\ {}^{13}\delta \tag{5}$$

and similarly for $^{17}\delta$ and $^{18}\delta$

By analogy we define for the ratios R:

$$^{13}\Delta \overset{\text{def}}{=} (^{13}R - {}^{13}R_{PDB})/{}^{13}R_{PDB} = {}^{13}R/{}^{13}R_{PDB} - 1, \text{ so that } {}^{13}R - {}^{13}R_{PDB} = {}^{13}R_{PDB}\ {}^{13}\Delta \tag{6}$$

and similarly for $^{17}\Delta$ and $^{18}\Delta$

In the above (Eqs. (5) and (6)) and the rest of this work we will express $\delta$ and $\Delta$ as small numbers, not in the "permil" (‰) notation, in which every delta value is multiplied by 1000. For example $\delta=0.020$ would normally be

written as 20 permil or 20 ‰.

## 4 Fractional abundances of isotopologues in molecules.

Converting measured $\delta^{13}C$ and $\delta^{18}O$ values into $^{16}O^{13}C^{16}O$ and $^{18}O^{12}C^{16}O$ isotopologue abundances is not straightforward due to the rare $^{17}O^{12}C^{16}O$ and doubly substituted isotopologues. Isotope ratio mass spectrometry

(IRMS) determines $\delta^{13}C$ and $\delta^{18}O$ values by measuring mass 45/44 and mass 46/44 ratios, with appropriate corrections for interfering masses, relative to a standard reference material. These mass ratios can be used with the accepted isotopic ratios of the standard reference materials to approximate the abundance as mole fraction (X) of the three main isotopologues in $CO_2$ using:

$$X(636) \cong {}^{13}R * X_{CO2} \tag{7}$$

$$X(826) \cong 2* {}^{18}R * X_{CO2} \tag{8}$$

$$X(626) \cong X_{CO2} - X(636) - X(826) \tag{9}$$

The oxygen abundance ratio is multiplied by a factor of two in Eq. (8) to convert the isotopic ratios from atomic abundance (i.e. $^{18}O/^{16}O$) into molecular abundance. The approximations in Eqs. (7)-(9) ignore the contribution of the

oxygen isotopes to X(636) and of $^{13}$C to X(826), as well as the portion of the total composed of the rare isotopologues. Depending on the level of uncertainty desired this may or may not be acceptable. As the WMO GAW CCL for $CO_2$, NOAA is obligated to minimize biases in the $CO_2$ calibration scale, and therefore we will correctly account for the apportionment of $CO_2$ through all isotopologues.

We start by assuming a purely statistical distribution of $^{13}$C, $^{18}$O, and other atoms when putting together a molecule starting from atomic abundance ratios as given in Table 1, namely, that the probability of picking a particular isotope is not affected by what is picked before or later. In general the other picks can affect the probability a little (called "clumped" isotopes), so that the thermodynamic abundances are slightly different from the statistical distribution. We will ignore that, and construct a purely statistical baseline distribution. Thus the probability of

picking a $^{13}$C atom for a carbon position is defined as simply $^{13}$R (the abundance ratio of $^{13}$C to total carbon). However, a molecule may have more than one position for C, O, N, etc. For example, suppose there are N chemical positions for a particular atom in a molecule and we want to define the probability of M of those positions being filled with one particular isotope (denoted isotope a). If the locations of the M, as a subset of N, do not matter, as is the case for symmetrical molecules like $CO_2$ and $CH_4$, we could call the N positions equivalent. In that case the

probability is

$$P = \binom{N}{M} * R_a{}^M * R_b{}^{N-M} \tag{10}$$

$R_a$ is the abundance ratio of isotope a, $R_b$ is the abundance ratio of other isotopes ($R_b = 1 - R_a$). The first term in equation 10 is the statistical weight which equals the number of combinations (a statistical term, the order of picking the M does not matter) of M out of N, given as

$$\frac{N!}{M!(N-M)!} \stackrel{\text{def}}{=} \binom{N}{M} \tag{11}$$

N! is the factorial notation, $N! \stackrel{\text{def}}{=} 1 * 2 * 3 * \dots \dots (N-1) * N$, with the special case $0! \stackrel{\text{def}}{=} 1$

Example: there are two equivalent positions for a single $^{18}$O in $CO_2$, namely $^{18}$O$^{12}$C$^{16}$O and $^{16}$O$^{12}$C$^{18}$O, jointly denoted as "826" (one $^{18}$O, one $^{12}$C, one $^{16}$O), so that the statistical weight is

$$\binom{2}{1} = \frac{2!}{1! * (2-1)!} = 2 \, .$$

Or for methane, a single or double substitution of deuterium ($^2$H, or D) for H has respective statistical weights:

$$\binom{4}{1} = \frac{4!}{1! * (4-1)!} = 4 \qquad \binom{4}{2} = \frac{4!}{2! * (4-2)!} = 6$$

It should be noted that whether positions can be considered equivalent depends on the symmetry of the molecule and the measurement method. For example for nitrous oxide the two positions for N in NNO would be equivalent when





mass 45 (one $^{14}$N, one $^{15}$N, one $^{16}$O) is measured in a mass spectrometer but they are not when an optical absorption method is used because the spectrum of $^{14}$N$^{15}$N$^{16}$O is different from $^{15}$N$^{14}$N$^{16}$O. In the latter case we need to keep separate track of the probabilities, denoted below as "P", of these two isotopomers. Isotopomers have the same number of specific isotopes, but they differ in their position in the molecule.

The probability for any particular $CO_2$ isotopologue is the product of the probability of picking the carbon isotope and the probability of picking the oxygen isotopes. Each of these probabilities is determined using Eq. (10). For example, the probability for the $^{18}$O$^{13}$C$^{16}$O isotopologue is the probability of picking one $^{13}$C isotope for one carbon position times the probability of picking one $^{18}$O isotope for one of the two oxygen positions and one $^{16}$O for the other.

The equations below give the probabilities for individual $CO_2$ isotopologues. When the isotopic compositions of the standard reference materials (PDB in Table 1) are filled in we obtain the numbers after the "=>" sign.

$$P(626) = (1-^{13}R)*(1-^{17}R-^{18}R)^2 \qquad => 0.98399421 = 1-0.016005794 \qquad (12)$$

$$P(636) = {}^{13}R*(1-^{17}R-^{18}R)^2 \qquad => 0.011057311 \qquad (13)$$

$$P(826) = (1-^{13}R)*2*^{18}R*(1-^{17}R-^{18}R) \qquad => 0.0041098949 \qquad (14)$$

The sum of the above three major abundances is 0.99916141 = 1- 0.00083859

$$P(726) = (1-^{13}R)*2*^{17}R*(1-^{17}R-^{18}R) \qquad => 0.00077753010 \qquad (15)$$

The sum of the above four major abundances is 0.99993894 = 1- 0.00006106

$$P(836) = {}^{13}R*2*^{18}R*(1-^{17}R-^{18}R) \qquad => 4.618359 \ 10^{-5}$$

and so on, with progressively smaller probabilities. The sum of all probabilities equals 1, which was verified
digitally in double precision. In a population of $CO_2$ molecules, (i.e. a sample or standard cylinder) probabilities determined from the abundance ratios of the population equate to the fractional abundance of each isotopologue.

**5 An expression for potential effects of isotopic mismatches on measurements of $CO_2$**

In this section we derive some practical expressions for errors, and corrections, resulting from isotopic mismatches
if they are ignored, for the case of $CO_2$. Similar considerations apply to other greenhouse gases such as $CH_4$, $N_2O$, etc. The unknown quantity of $CO_2$-in-air that we intend to measure is called "measurand". It can be a real air sample or an intermediate transfer standard. The errors typically depend on the instrument used because an instrument may be sensitive to just one isotopologue, or equally sensitive to all isotopologues, or something in between. Here we





give an example for an instrument that quantifies total $CO_2$ in air, denoted $X_{CO2}$, by measuring only one isotopologue, namely $^{16}O^{12}C^{16}O$. We assume that the instrument is calibrated by a $CO_2$ standard with fractional abundances $^{13}R_{PDB}$ and $^{18}R_{PDB}$ of the two main isotopologues, corresponding to the international PDB reference points for $^{13}C$ and $^{18}O$. In almost all cases deviations of $^{17}R$ from PDB are tightly correlated with deviations of $^{18}R$

from PDB. The deviation of total $CO_2$ from being proportional to $P(626)$ due to inconsistencies of $^{13}R$, $^{17}R$, $^{18}R$ between the measurand and PDB, using Eq. (12), is

$$\Delta P(626) \stackrel{\text{def}}{=} P(626) - P_{PDB}(626) =$$

$$\frac{\partial P(626)}{\partial ^{13}R}(^{13}R - {}^{13}R_{PDB}) + \frac{\partial P(626)}{\partial ^{17}R}(^{17}R - {}^{17}R_{PDB}) + \frac{\partial P(626)}{\partial ^{18}R}(^{18}R - {}^{18}R_{PDB})$$

The above are the first terms of a Taylor expansion around $P_{PDB}(626)$. Inserting the first derivatives and using Eq. (6) gives:

$$\Delta P(626) = -\left(1 - {}^{17}R_{PDB} - {}^{18}R_{PDB}\right)^2 \left(^{13}R_{PDB}{}^{13}\Delta\right) +$$

$$-2\left(1 - {}^{13}R_{PDB}\right)\left(1 - {}^{17}R_{PDB} - {}^{18}R_{PDB}\right)\left(^{17}R_{PDB}{}^{17}\Delta + {}^{18}R_{PDB}{}^{18}\Delta\right)$$

If $^{13}\Delta$ is positive the air to be measured has a higher $^{13}C/^{12}C$ ratio than PDB. Therefore $P(626)$ is slightly lower than it is for PDB, and the relative correction in the mole fraction assigned to the measured air will have to be positive, of opposite sign to the relative error of $P(626)$:

$$\frac{\Delta X_{CO2}}{X_{CO2}} = -\frac{\Delta P(626)}{P(626)} = \frac{^{13}R_{PDB}{}^{13}\Delta}{\left(1 - {}^{13}R_{PDB}\right)} + \frac{2\left(^{17}R_{PDB}{}^{17}\Delta + {}^{18}R_{PDB}{}^{18}\Delta\right)}{\left(1 - {}^{17}R_{PDB} - {}^{18}R_{PDB}\right)}$$

We note here that we could have used a $^{16}O^{13}C^{16}O$ line for quantifying $X_{CO2}$, but an analogous derivation for $\Delta P(636)/P(636)$ shows that it is 90 times more sensitive to isotopic errors or mismatches.

Using Eqs. (2) and (3) gives

$$\frac{\Delta X_{CO2}}{X_{CO2}} = {}^{13}r_{PDB}{}^{13}\Delta + 2\left(^{17}r_{PDB}{}^{17}\Delta + {}^{18}r_{PDB}{}^{18}\Delta\right) \tag{16}$$

Generally, one is not making atmospheric $CO_2$ measurements with standards that have isotopic abundances for C and O exactly like PDB. Because the linear Eq. (16) applies to the measurement of a transfer standard itself as well as to an air sample, we can give an expression for corrections to be made when the standard (subscript "st") has an

isotopic composition different from air but not equal to PDB:

$$\frac{\Delta X_{CO2}}{X_{CO2}} = {}^{13}r_{PDB}\left(^{13}\Delta_{air} - {}^{13}\Delta_{st}\right) + 2\left(^{17}r_{PDB}\left(^{17}\Delta_{air} - {}^{17}\Delta_{st}\right) + {}^{18}r_{PDB}\left(^{18}\Delta_{air} - {}^{18}\Delta_{st}\right)\right) \tag{17}$$

In the Appendix we derive the following very close approximation to Eq. (17) in which the $\Delta$ values have been replaced by the familiar $\delta$ values:





$$\Delta X_{CO2} = X_{CO2} \left[ 0.01111 \left( {}^{13}\delta_{air} - {}^{13}\delta_{st} \right) + 2 * 0.0023 \left( {}^{18}\delta_{air} - {}^{18}\delta_{st} \right) \right] \tag{18}$$

This is an expression for $CO_2$ corrections when only the ${}^{16}O^{12}C^{16}O$ isotopologue is used to measure $X_{CO2}$, and we are using PDB scales. As an example, if we use a standard with $CO_2$ made from natural gas, it could have ${}^{13}\delta_{st} = -0.045$ and ${}^{18}\delta_{st} = -0.017$ on the PDB scales, whereas air has ${}^{13}\delta_{air} \cong -0.008$ and ${}^{18}\delta_{air} \cong 0.000$. Assuming $X_{CO2} = 400$ μmol mol$^{-1}$, then $\Delta X = 0.164 + 0.031 = 0.194$ μmol mol$^{-1}$. ${}^{13}\delta_{air}$ is higher than ${}^{13}\delta_{st}$, so that the ${}^{16}O^{12}C^{16}O$ abundance of the standard is higher than assumed, resulting in the air measurement being too low. Therefore an upward correction is needed for ${}^{13}C$ and likewise for ${}^{18}O$.

## 6 Practical calculations for definition and propagation of the $CO_2$ calibration scale

Equation (18) gives the correction required when only the ${}^{16}O^{12}C^{16}O$ isotopologue is used to determine $X_{CO2}$ as a function of the isotopic differences between the sample and a single standard. However, most $CO_2$ measurements are made vs a suite of standards that may have various isotopic compositions and the isotopic compositions may be a function of $CO_2$ (as is the case for the primary $CO_2$ standards used by the WMO GAW CCL). In this case the calibration curve that defines the response of the analyser may incorporate a systematic error making the idea of a simple "correction" impractical. Equation (18) can best be used to quickly estimate the potential offsets due to sample/standard isotopic differences but is not practical for making corrections when multiple standards are used. Therefore, we must instead use a calibration approach that fully accounts for the isotopic composition of the standards rather than using a post measurement correction.

We have taken the approach of decomposing the total $CO_2$ in the primary standards, as defined by manometric measurements, into individual isotopologue mole fractions based on measured $\delta^{13}C$ and $\delta^{18}O$ values. The $\delta^{13}C$ and $\delta^{18}O$ values are determined by IRMS by the Stable Isotope Laboratory, Institute of Arctic and Alpine Research, University of Colorado, Boulder (INSTAAR) (Trolier et al., 1996). These isotopologue specific mole fractions of the standards are used to calibrate laser based spectroscopic instruments for the three major $CO_2$ isotopologues (${}^{16}O^{12}C^{16}O$, ${}^{16}O^{13}C^{16}O$, and ${}^{18}O^{12}C^{16}O$) individually. The three major isotopologues in unknown cylinders are measured relative to these isotopologue specific calibration curves. The isotopologue mole fractions of the unknowns are then recombined into total $CO_2$ and conventional $\delta^{13}C$, and $\delta^{18}O$ values while properly accounting for the non-measured rare isotopologues.

Suppose we have one instrument measuring each isotopologue (${}^{16}O^{12}C^{16}O$, ${}^{16}O^{13}C^{16}O$, ${}^{18}O^{12}C^{16}O$ and perhaps also ${}^{17}O^{12}C^{16}O$) individually. The response of the instrument for each of the isotopologues needs to be calibrated separately. How often such calibrations need to be repeated depends on the instrument. For this purpose we need to have a series of reference gas standards with well defined total $CO_2$ ($X_{CO2}$) and with known conventional $\delta$-values for the isotopic ratios. Equations (12)-(15) can be used to convert that information to the fractional abundances of





the isotopologues, by first writing them in terms of conventional delta values by using relations (2) and (3) and by writing $\mathbf{r}_{sample}$ as $\mathbf{r}_{PDB}(1+\delta)$ (see Eq. (5)).

$$P(626) = \frac{1}{1+{}^{13}\mathbf{r}_{PDB}(1+{}^{13}\delta)} \quad \frac{1}{\left[1+{}^{17}\mathbf{r}_{PDB}(1+{}^{17}\delta)+{}^{18}\mathbf{r}_{PDB}(1+{}^{18}\delta)\right]^2} \qquad (19)$$

$$P(636) = \frac{{}^{13}\mathbf{r}_{PDB}(1+{}^{13}\delta)}{1+{}^{13}\mathbf{r}_{PDB}(1+{}^{13}\delta)} \quad \frac{1}{\left[1+{}^{17}\mathbf{r}_{PDB}(1+{}^{17}\delta)+{}^{18}\mathbf{r}_{PDB}(1+{}^{18}\delta)\right]^2} \qquad (20)$$

$$P(826) = \frac{1}{1+{}^{13}\mathbf{r}_{PDB}(1+{}^{13}\delta)} \quad \frac{2*{}^{18}\mathbf{r}_{PDB}(1+{}^{18}\delta)}{\left[1+{}^{17}\mathbf{r}_{PDB}(1+{}^{17}\delta)+{}^{18}\mathbf{r}_{PDB}(1+{}^{18}\delta)\right]} \qquad (21)$$

$$P(726) = \frac{1}{1+{}^{13}\mathbf{r}_{PDB}(1+{}^{13}\delta)} \quad \frac{2*{}^{17}\mathbf{r}_{PDB}(1+{}^{17}\delta)}{\left[1+{}^{17}\mathbf{r}_{PDB}(1+{}^{17}\delta)+{}^{18}\mathbf{r}_{PDB}(1+{}^{18}\delta)\right]} \qquad (22)$$

If $\delta^{17}O$ has not been measured, we approximate $\delta^{17}O = 0.528 * \delta^{18}O$ to determine the fractional abundances above.

The fractional abundances (Eqs. (19)-(22)) are converted into mole fractions in dry air by multiplying with the total mole fraction of $CO_2$ in dry air ($X_{CO2}$). The isotopologue mole fractions in air are written as X(626), etc. In other

words, we have X(626) = $X_{CO2}$ * P(626) and similar for all isotopologues.

A series of standards can in this way be used to calibrate the instrument response for each isotopologue individually. With these response functions we can then assign mole fractions in air to the isotopologues of the unknown gas mixtures that are being measured, X(626)$_{unk}$, etc.

Then we need to convert the measured isotopologue abundances of the unknown (X(626)$_{unk}$, X(636)$_{unk}$, and

X(628)$_{unk}$) back to standard delta-notation using Eqs. (19)-(22) as follows:

$$\frac{X(636)_{unk}}{X(626)_{unk}} = \frac{P(636)}{P(626)} = {}^{13}\mathbf{r}_{PDB}(1 + {}^{13}\delta) \quad => \quad {}^{13}\delta = \frac{X(636)_{unk}}{{}^{13}\mathbf{r}_{PDB}*X(626)_{unk}} - 1 \qquad (23)$$

and $$\frac{X(826)_{unk}}{X(626)_{unk}} = \frac{P(826)}{P(626)} = 2 * {}^{18}\mathbf{r}_{PDB}(1 + {}^{18}\delta)\left[1 + {}^{17}\mathbf{r}_{PDB}(1 + {}^{17}\delta) + {}^{18}\mathbf{r}_{PDB}(1 + {}^{18}\delta)\right] \qquad (24)$$

Equation (24) is (weakly) non-linear. We re-arrange it as

$$\frac{X(826)_{unk}}{2 * {}^{18}\mathbf{r}_{PDB} * X(626)_{unk}} = 1 + [\ldots] + {}^{18}\delta (1 + [\ldots])$$

in which we defined $[\ldots] \equiv {}^{17}\mathbf{r}_{PDB}(1 + {}^{17}\delta) + {}^{18}\mathbf{r}_{PDB}(1 + {}^{18}\delta)$

and rearrange it further into:





$$^{18}\delta = \left[\frac{X(826)_{unk}}{2*{}^{18}\mathbf{r}_{PDB}*X(626)_{unk}} - 1\right] - [(1 + {}^{18}\delta)({}^{17}\mathbf{r}_{PDB}(1 + {}^{17}\delta) + {}^{18}\mathbf{r}_{PDB}(1 + {}^{18}\delta))] \tag{25}$$

The first approximation to $^{18}\delta$ is to assume $\delta^{18}O$ and $\delta^{17}O = 0$ on the right hand side, i.e. equal to the standard reference material:

$$^{18}\delta = \left[\frac{X(826)_{unk}}{2 * {}^{18}\mathbf{r}_{PDB} * X(626)_{unk}} - 1\right] - {}^{17}\mathbf{r}_{PDB} - {}^{18}\mathbf{r}_{PDB}$$

Then we substitute this first approximation into Eq. (25), with the assumption of $\delta^{17}O = 0.528 * \delta^{18}O$, and iterate the
solution for $^{18}\delta$ by continuing to substitute it in the right hand side of Eq. (25). The approximation is extremely close to the full solution unless the sample is highly depleted in $\delta^{18}O$.

If $X(726)_{unk}$ has been measured, an equation similar to Eq. (25) applies:

$$^{17}\delta = \left[\frac{X(726)_{unk}}{2*{}^{17}\mathbf{r}_{PDB}*X(626)_{unk}} - 1\right] - [(1 + {}^{17}\delta)({}^{17}\mathbf{r}_{PDB}(1 + {}^{17}\delta) + {}^{18}\mathbf{r}_{PDB}(1 + {}^{18}\delta))] \tag{26}$$

In this case Eqs. (25) and (26) can be iterated together substituting updated values for both $^{18}\delta$ and $^{17}\delta$.

The total $CO_2$ in dry air is given by

$$X_{CO2,unk} = \frac{X(626)_{unk} + X(636)_{unk} + X(826)_{unk} + X(726)_{unk}}{P(626)_{unk} + P(636)_{unk} + P(826)_{unk} + P(726)_{unk}} \tag{27}$$

Dividing by the sum of the fractional abundances P, which would be equal to 0.99993894 if the isotopic ratios are equal to the standard reference materials for carbon and oxygen, would add 0.024 µmol mol$^{-1}$ to $X_{CO2}$, assuming $X_{CO2} \sim 400$ µmol mol$^{-1}$. This small difference accounts for the rare isotopologues with multiple isotopic substitutions
that are not being measured. The correction in Eq. (27) that applies for the unknown will in general be *very* slightly different from 1−0.00006106 (see above, the sum of the four major molecular abundances). We calculate actual P values for the unknown using Eqs. (19)-(22) with the δ-values from Eqs. (23)-(26) and then use those in Eq. (27) instead of the standard reference material values to account for this small discrepancy, but it is not necessary in most cases.

## 7 Analytical methods

NOAA's new $CO_2$ calibration system is based on multiple laser spectroscopic techniques. It uses a combination of cavity ring-down spectroscopy (CRDS, Picarro, Inc. $CO_2/CH_4/H_2O$ analyzer, model number G2301) (O'Keefe and Deacon, 1988; Crosson, 2008), off-axis integrated cavity output spectroscopy (ICOS, Los Gatos Research, Inc.,
carbon dioxide isotope analyzer, CCIA-46-EP, model number 913-0033-0000) (Paul et al., 2001; Baer et al., 2002),



and quantum cascade tunable infrared laser differential absorption spectroscopy (QC-TILDAS, Aerodyne Research, Inc., carbon dioxide isotope analyzer, model QCTILDAS-CS) (Tuzson et al., 2008; McManus et al., 2015).

The CRDS instrument measures a single absorption line from the $^{16}O^{12}C^{16}O$ isotopologue at 1603 nm. For most of the data presented here, the instrument operated in an enhanced $CO_2$ mode where it did not measure $CH_4$ and instead

focussed exclusively on the $CO_2$ absorption line with periodic measurements of $H_2O$ as a diagnostic. However, we have since determined this enhanced $CO_2$ mode does not improve the reproducibility of $CO_2$ measurements. We are currently testing the ability to do $CH_4$ calibrations at the same time as the $CO_2$ calibrations using the standard operating mode of the CRDS.

The ICOS and QC-TILDAS analyzers both measure absorption lines of $^{16}O^{12}C^{16}O$, $^{16}O^{13}C^{16}O$, and $^{18}O^{12}C^{16}O$

isotopologues individually (using lines in the 2300 cm$^{-1}$ $CO_2$ absorption bands). Both analyzers also measure the $^{17}O^{12}C^{16}O$ isotopologue but we cannot independently calibrate this measurement so we assume that $\delta^{17}O$ follows the mass dependent fractionation relative to $\delta^{18}O$. The two analyzers have comparable performance and serve as backups for each other since only one is installed and used at a time. In the following discussion they are designated collectively as the $CO_2$ isotope analyzer. The $^{16}O^{12}C^{16}O$ measurement in the isotope analyzers uses a weak

absorption line to match the measured absorption with the low abundance minor isotopologues. They are therefore not as precise as the measurement on the CRDS. The internal $^{16}O^{12}C^{16}O$ measurement from the isotope analyzer is not used to calculate total $CO_2$ but is used as $X(626)_{unk}$ in the calculation of $\delta^{13}C$ and $\delta^{18}O$ (see Eqs. (23) and (25) in the discussion above). Using this "internal" $X(626)_{unk}$ measurement gives slightly more precise $\delta^{13}C$ and $\delta^{18}O$ results than using the "external" $X(626)_{unk}$ measurement from the CRDS system since it accounts for some instrument bias

common to both the $^{16}O^{12}C^{16}O$ and the $^{16}O^{13}C^{16}O$ and $^{18}O^{12}C^{16}O$ isotopologue measurements. $X(626)_{unk}$ from the CRDS system is used in Eq. (27) to calculate total $CO_2$.

Figure 1 is a plumbing diagram for the $CO_2$ calibration system. The system uses the CRDS analyzer plus one of the $CO_2$ isotope analyzers. All measurements on the system are relative to a reference tank of compressed, unmodified natural air. A 4-port, 2-position switching valve (Valco Instruments Co, Inc. (VICI), model EUDA-24UWE) is used

to send sample/standard gas to one analyzer while the other analyzer simultaneously measures the reference tank. Sample and standard tanks are introduced to the system via two identical sample manifolds composed of 16-port multi-position selection valves (VICI, model EUTA-2CSD16MWE). A 4-port multi-position stream selection valve (VICI, model EUTA-2SD4MWE) is used to select either manifold A, B, or, optionally, for expansion to a third manifold C. A plugged port on the manifold selection valve is used as a safe off port during shutdown.

Sample/standard and reference gas pressures are controlled at 760 ± 1 Torr by two electronic pressure controllers with integrated mass flow meters (MKS Instruments, type 649B electronic pressure controller, model number 649B00813T13C2MR). The analyzers themselves control their internal cell pressures. However, controlling the inlet pressure prevents large inlet pressure swings due to inconsistent cylinder regulator set points and allows the internal pressure control to be more consistent. All three instruments are continuous flow instruments so an idle gas is

provided through a 3-way solenoid valve (Parker, model 009-0143-900) just upstream of the instrument inlet. The





solenoid valve fails to the idle gas during power outages to prevent loss of cylinders. This idle gas is partially dried room air drawn through a Nafion drier (Perma Pure LLC.) for extended system idle time (e.g. on weekends) but is a cylinder of dried ambient natural air (dew point ~ -80 °C) for short idle times during and just prior to actual calibrations. This cylinder ensures that the system downstream of the water traps does not get exposed to elevated

levels of water vapor during short idle times between analyses. Each analyzer has a $H_2O$ trap up stream of the inlet that normalizes any differences in water content among cylinders analyzed. These traps are 3.2 mm OD stainless steel tubing loops immersed in a -78 °C ethanol bath (SP Scientific Inc., MultiCool, model number MC480A). Both analyzers have individual sampling pumps to pull gas through the sample cell at partial vacuum. All tubing in the system is 3.2 mm or 1.6 mm OD stainless steel.

The flow rates are set to 130 - 150 mL min$^{-1}$ by using a critical flow orifice downstream of the isotope analyzer cell or by partially closing the upstream solenoid valve in the CRDS instrument and relying on a stable pressure at the instrument inlet. The analysis sequence starts with a 4 minute flush of the sample/standard regulator (and sample/standard electronic pressure controller) and then alternates reference and sample through the two analyzers for 8 cycles before moving to the next sample or standard. Each measurement cycle is 2.5 minutes of flushing and a

30 second signal average.

## 8 Calibration and system performance

Analyzers are calibrated approximately every two weeks in an offline calibration mode using a suite of 14 secondary standards, covering the range 250 to 600 µmol mol$^{-1}$ total $CO_2$. The system is calibrated routinely to 600 µmol mol$^{-1}$

in expectation of a scale expansion in 2017. Each isotopologue is calibrated independently after decomposing the standard's total $CO_2$ into its component isotopologue mole fractions using the method discussed above. The secondary standards have assigned total $CO_2$ values by calibration against the entire set of primary standards (plus two additional standards that will extend the scale to 600 µmol mol$^{-1}$) in an analogous manner as described here. This is a significant change from our previous NDIR system where subsets of standards were used. It makes the new

calibration system less likely to have $CO_2$ dependent biases. The secondary standard's $\delta^{13}C$ and $\delta^{18}O$ values were assigned by IRMS measurement at INSTAAR. Primary standards also have $\delta^{13}C$ and $\delta^{18}O$ values assigned by INSTAAR, which we use when primary standards are used to calibrate secondary standards. The use of INSTAAR $\delta^{13}C$ and $\delta^{18}O$ assigned values on the secondary standards rather than the values from measurement versus the primary standards, shortens the traceability of the delta measurements to a true IRMS measurement. A comparison

of the INSTAAR assignments with the NOAA measured isotopic values for the secondary standards is discussed below.

The instrument readings are absorption measurements corrected for cell pressure and temperature and converted into nominal mole fraction units. However, we treat them purely as an instrument response in arbitrary units. They could also be a voltage or a current. The responses from the analyzers are subsequently used in an offline calibration of



each instrument. We do not use the internal calibration capabilities of the instruments; this ensures that the measurements are directly traceable to the WMO primary standards and can be reprocessed for future scale revisions. Each standard is measured relative to a reference cylinder to correct for slow drift of the analyzers. For the CRDS and ICOS analyzers the instrument response to each standard is divided by the average instrument response

of the bracketing reference aliquots. For the QC-TILDAS, the difference between the response to the standard and the reference is used. In both cases we term the resulting values "response ratios". The choice of division vs subtraction is made due to the characteristics of the drift in each analyser. For example, the division operation does a better job when there is a slow span drift (perhaps due to variations in cell temperature and pressure) causing relative changes that are proportional to $X_{CO2}$, whereas the difference operation is more appropriate when the majority of the

drift is caused by a uniform shift in the output that does not depend on $X_{CO2}$. Rather than characterize the source of drift in each analyzer we use the reproducibility of target tank measurements to empirically determine which method gives more consistent results between calibration episodes.

The calibration curves are $CO_2$ as a function of response ratios. The CRDS instrument response is linear within the uncertainty of the standards. However, both isotope analyzers are slightly non-linear in their response and are fit

with a quadratic polynomial. Non-linearity in the isotope analyzers may be partially due to incomplete flushing of the sample cell, caused by un-swept dead volumes, as the system switches from reference to standard. Memory of the reference gas (ambient air from Niwot Ridge, ~400 μmol mol$^{-1}$ $CO_2$) in the sample cell influences the standards on the ends of the scale more than those close to the reference gas value potentially leading to a slight non-linear response. Since all standards and all samples are measured against the same reference gas, small memory effects

should cancel out.

Sample measurements are made relative to the same reference tank to account for drift in the analyzers between calibration episodes. The sample response ratios are used with the isotopologue specific calibration curves to determine isotopologue mole fractions for the sample cylinder which are combined into total $CO_2$, $\delta^{13}C$, and $\delta^{18}O$ values using the method discussed above. These values (total $CO_2$, $\delta^{13}C$, and $\delta^{18}O$) are stored in the NOAA database

and are reported to the user via certificates and the web interface. Isotopologue specific mole fractions are not provided, however the equations described in this paper can be used to regenerate them.

Performance of the new calibration system has been evaluated over approximately one year by repeated measurements of target tanks (cylinders repeatedly measured as a diagnostic of system performance). Figure 2 shows the time series of total $CO_2$ measured for 4 target tanks with $CO_2$ ranging from 357 to 456 μmol mol$^{-1}$.

Standard deviations of the measurements are approximately ± 0.007 μmol mol$^{-1}$. Reproducibility of the target tanks close to the reference tank (typically ~ 400 μmol mol$^{-1}$ $CO_2$) are a little better than those farther out on the ends of the calibration range but the difference is small. While one year is not a long enough time series to fully quantify the reproducibility of the system, we estimate it to be ± 0.01 μmol mol$^{-1}$ (68% CI) based on these target tank measurements. This is a significant improvement over the NDIR system where reproducibility is ±0.03 μmol mol$^{-1}$

(68% CI) ("Carbon Dioxide WMO Scale", 2017).





Prior to this new $CO_2$ calibration system, NOAA provided informational isotopic values for tertiary standards delivered to outside organizations by taking discrete samples from cylinders in flasks and having them measured by INSTAAR. This continued during the 6 months period when both calibration systems were run in parallel. Comparisons of these measurements with the isotopic results from the new calibration system are show in Figs. 3

and 4. The top plot in each figure is differences of measured delta values (NOAA – INSTAAR) vs INSTAAR values and the bottom plot in each figure is differences as a function of total $CO_2$ measured by NOAA. There is no systematic bias between the NOAA and INSTAAR measurements for either species, except for highly depleted cylinders ($\delta^{13}C$ or $\delta^{18}O$ less than -20‰, shown by open symbols in both figures) and $\delta^{18}O$ in very high $CO_2$ cylinders (> 490 µmol mol$^{-1}$). The average offset (NOAA – INSTAAR) of non-depleted tanks is   0.0 ± 0.1‰ $\delta^{13}C$

and 0.0 ± 0.2‰ $\delta^{18}O$.  The offset in the highly depleted cylinders most likely occurs as a result of the large extrapolation in the INSTAAR IRMS measurements from the working standard at ambient $\delta^{13}C$ and $\delta^{18}O$. The $\delta^{18}O$ data do show a pronounced "hook" above ~ 490 µmol mol$^{-1}$. This is thought to be due to issues when sampling air from cylinders into flasks and not to the measurements either at INSTAAR or NOAA. A tertiary standard with 497 µmol mol$^{-1}$ $CO_2$ showed excellent agreement when measured directly by both NOAA ($\delta^{18}O$ = -8.92 ± 0.04‰) and

INSTAAR ($\delta^{18}O$ = -8.91 ± 0.06‰). A comparison can also be made using the secondary standards which were calibrated directly by INSTAAR and by NOAA verses the primary standards. The assigned values of the secondary standards (as measured directly by INSTAAR) and the NOAA minus INSTAAR differences are shown in Figs. 5 and 6 for $\delta^{13}C$ and $\delta^{18}O$ respectively. Agreement is very good but there is a loss of precision on the NOAA calibration system near the wings of the $CO_2$ scale. NOAA measurements show some decrease in performance as

total $CO_2$ moves away from the reference cylinder, which is always an ambient $CO_2$ cylinder. However, even on the wings of the range the performance is more than adequate for the purpose of correcting total $CO_2$ for isotopic differences. The reproducibilities of $\delta^{13}C$ (± 0.2‰, 68% CI) and $\delta^{18}O$ (± 0.2‰, 68% CI) are again estimated from target tanks measurements. The uncertainty for the isotope measurements is too large for these results to be used as true $CO_2$ isotope standards but is more than adequate for correcting atmospheric $CO_2$ measurement for standard vs

sample isotope differences.

The new calibration system was run in parallel with the NDIR system from April 2016 through October 2016. Agreement between the two systems near ambient $CO_2$ is good but there are significant offsets between 300 to 360 µmol mol$^{-1}$ and 430 to 500 µmol mol$^{-1}$ (Fig. 7). These offsets can be traced primarily to the effects of calibrating the NDIR system with subsets of the primary standards when transferring the scale to secondary standards. Using

subsets in this way makes the results from the NDIR system sensitive to uncertainty in the assigned values of the individual primary standards. Additional manometric determinations have been made since the assignments were made in 2007. Also the use of the new calibration system for correcting the average manometer values for residuals of a fit to the entire set will help to improve the consistency of the individual assignments and thus reduce the $CO_2$ dependency of the NDIR measurements. These improvements, as well as two additional subtle bias corrections in

the manometer calculations, will be incorporated in an upcoming scale revision (scheduled for mid-2017) (Brad Hall, personal communication). The revised scale should remove most of the $CO_2$ dependent bias between the two analysis systems. Although there may be a component due to gas handling issues on the NDIR system that cannot be



resolved. This is still under investigation and will be addressed in a forthcoming paper discussing the scale revision. After the scale revision all past calibrations of tertiary standards will be revised to the new scale. Calibrating the new system by fitting all primary standards makes the new system very insensitive to the assignment of individual cylinders. Thus results from the new system are more accurate than from the NDIR, however, caution should be

used when evaluating cylinders for drift when comparing historical results from the NDIR system and new measurements from the new calibration system as these systematic system differences could be incorrectly interpreted as drift.

Figure 7 also has results from highly depleted tanks ($\delta^{13}C$ < -20‰) that shows a greater NDIR minus laser difference. This is consistent with the NDIR having reduced sensitivity to the minor isotopologues. Quantifying the

sensitivity of the current NDIR (LI-COR 6252) is difficult due to the $CO_2$ dependent biases and would not be possible for historical NDIR analyzers used on the NDIR $CO_2$ calibration system. Measurements of isotopically depleted cylinders by NOAA via NDIR need to be considered more uncertain due to this unknown isotope sensitivity of NDIR's used for $CO_2$ calibrations.

**9 Conclusions**

We describe here the expected distribution of isotopologues of $CO_2$ based on measured $\delta^{13}C$ and $\delta^{18}O$ and its application in calibrating cylinders for total $CO_2$, $\delta^{13}C$, and $\delta^{18}O$. The distribution accounts for all isotopologues, including rare doubly substituted isotopologues. The methods are applicable to $CO_2$ or any other molecule where isotopologue (or isomer) specific values are required to reach desired precision goals.

The new calibration system provides total $CO_2$ values that are insensitive to isotopic differences between standards and provides to users of the standards a characterization of the isotopic composition of the standards. The isotopic values are not intended for propagating the isotopic standard scales, they are only to be used to make corrections to atmospheric $CO_2$ measurements made by instruments that have selective sensitivities to the isotopologues. Isotopic standards should be calibrated by IRMS measurements.

The performance of the new calibration system improves our ability to propagate the $CO_2$ scale and is expected to lead to improvements in the compatibility of measurement networks provided laboratories maintain tight connection with the CCL. Although the system has not run long enough to fully evaluate the reproducibility of the scale transfer, it is expected to be approximately ± 0.01 μmol mol$^{-1}$ (68% CI). Comparison of the new calibration system with the historical NDIR based system shows significant $CO_2$ dependence in the NDIR measurements. This results

from a combination of errors in the assigned values of the primary standards and the use of small subsets of the primary standards when the scale is transferred to secondary standards. This is under further investigation and we expect to resolve the issue with an upcoming revision to the $CO_2$ in air scale.





**Code availability**

Available upon request.

**Data availability**

Cylinder calibration results presented in this work include those used by laboratories outside of NOAA. We can

5    provide results in anonymous form upon request.





**Appendix**

We will derive expressions for $\Delta$ in terms of conventional $\delta$ values because we currently supply standards to users within the greenhouse gas measurement community with their $\delta$ values as information in addition to the total $X_{CO2}$ calibration.

$$^{13}\Delta = \frac{^{13}R}{^{13}R_{PDB}} - 1 = \frac{^{13}\mathbf{r}}{^{13}\mathbf{r}_{PDB}}\frac{1 + ^{13}\mathbf{r}_{PDB}}{1 + ^{13}\mathbf{r}} - 1 = (1 + ^{13}\delta)(1 + ^{13}\mathbf{r}_{PDB})(1 - ^{13}\mathbf{r} + ^{13}\mathbf{r}^2) - 1$$

Where we have used the first 3 terms of the series expansion $(1+\mathbf{r})^{-1} = 1 - \mathbf{r} + \mathbf{r}^2 - \mathbf{r}^3 + \dots$ and the definitions of $\mathbf{r}$, $R$, $\delta$, and $\Delta$. Expanding,

$$^{13}\Delta = (1 - ^{13}\mathbf{r} + ^{13}\mathbf{r}^2) + ^{13}\mathbf{r}_{PDB}(1 - ^{13}\mathbf{r} + ^{13}\mathbf{r}^2) +$$

$$^{13}\delta(1 - ^{13}\mathbf{r} + ^{13}\mathbf{r}^2) + ^{13}\delta^{13}\mathbf{r}_{PDB}(1 - ^{13}\mathbf{r} + ^{13}\mathbf{r}^2) - 1$$

and rearranging, we get

$$^{13}\Delta = (-^{13}\mathbf{r} + ^{13}\mathbf{r}^2) + (^{13}\mathbf{r}_{PDB} + ^{13}\delta)(1 - ^{13}\mathbf{r} + \underline{^{13}\mathbf{r}^2}) + ^{13}\delta^{13}\mathbf{r}_{PDB}(1 - ^{13}\mathbf{r} + \underline{^{13}\mathbf{r}^2})$$

Rearranging further,

$$^{13}\Delta = ^{13}\delta - (^{13}\mathbf{r} - ^{13}\mathbf{r}_{PDB}) + ^{13}\mathbf{r}(^{13}\mathbf{r} - ^{13}\mathbf{r}_{PDB}) - ^{13}\delta(^{13}\mathbf{r} - ^{13}\mathbf{r}_{PDB}) - ^{13}\delta^{13}\mathbf{r}_{PDB}{}^{13}\mathbf{r}$$

Then, using Eq. (5),

$$^{13}\Delta = ^{13}\delta - ^{13}\mathbf{r}_{PDB}{}^{13}\delta + ^{13}\mathbf{r}^{13}\mathbf{r}_{PDB}{}^{13}\delta - ^{13}\delta^{13}\mathbf{r}_{PDB}{}^{13}\delta - ^{13}\delta^{13}\mathbf{r}_{PDB}{}^{13}\mathbf{r}$$

The third and the last term cancel, and then keeping only the two leading terms, we obtain

$$^{13}\Delta = ^{13}\delta(1 - ^{13}\mathbf{r}_{PDB}) \tag{A1}$$

Equation (A1) is an excellent approximation. Using the values for $^{13}\mathbf{r}_{PDB}$ in Table 1 and assuming that $^{13}\delta = -0.00800$ ($-8.00$ permil, an approximate value for $CO_2$-in-air) we calculate both $^{13}R$ for the air sample and $^{13}R_{PDB}$, and using the definition (Eq. (6)) for $^{13}\Delta$, we obtain $^{13}\Delta = -0.0079102$. Equation (A1) gives us $-0.0079101$.

A very similar derivation holds for $^{17}\Delta$ and $^{18}\Delta$ but it is a bit more complicated because the terms for $^{17}R$ and $^{18}R$ get mixed.

$$^{17}\Delta = \frac{^{17}\mathbf{r}}{^{17}\mathbf{r}_{PDB}}\frac{1 + ^{17}\mathbf{r} + ^{18}\mathbf{r}}{1 + ^{17}\mathbf{r}_{PDB} + ^{18}\mathbf{r}_{PDB}} - 1 = \frac{^{17}\mathbf{r}}{^{17}\mathbf{r}_{PDB}}\frac{1 + ^{78}\mathbf{r}}{1 + ^{78}\mathbf{r}_{PDB}} - 1$$

To keep the notation simpler and stressing the analogy with the derivation for $^{13}\Delta$ we have written in the above $^{78}\mathbf{r} = ^{17}\mathbf{r} + ^{18}\mathbf{r}$ for the air sample and $^{78}\mathbf{r}_{PDB} = ^{17}\mathbf{r}_{PDB} + ^{18}\mathbf{r}_{PDB}$ for the standard.

After keeping only the leading terms we have

$$^{17}\Delta = ^{17}\delta - (^{78}\mathbf{r} - ^{78}\mathbf{r}_{PDB}) = ^{17}\delta - ^{17}\mathbf{r}_{PDB}{}^{17}\delta - ^{18}\mathbf{r}_{PDB}{}^{18}\delta \tag{A2}$$

And similarly for $^{18}O$:



$$^{18}\Delta = {}^{18}\delta - {}^{17}\mathbf{r}_{PDB}{}^{17}\delta - {}^{18}\mathbf{r}_{PDB}{}^{18}\delta \tag{A3}$$

These are the equivalents of Eq. (A1) for $^{13}\Delta$. Because $^{17}\mathbf{r}$ and $^{18}\mathbf{r}$ are significantly smaller than $^{13}\mathbf{r}$, we approximate further $^{17}\Delta \cong {}^{17}\delta$ and $^{18}\Delta \cong {}^{18}\delta$. Since $^{17}\delta$ is not usually measured and also is often very closely related as $^{17}\delta = 0.53 \ ^{18}\delta$, we can write for the oxygen correction terms in Eq. (16) $^{17}\mathbf{r}_{PDB}{}^{17}\delta + {}^{18}\mathbf{r}_{PDB}{}^{18}\delta = (0.53 \ ^{17}\mathbf{r}_{PDB} + {}^{18}\mathbf{r}_{PDB}) \ ^{18}\delta$, and

5     filling in the $\mathbf{r}_{PDB}$ values from Table 1,

$$0.53*395*10^{-6}*{}^{18}\delta + 2088*10^{-6}*{}^{18}\delta = 2297*10^{-6}*{}^{18}\delta \cong 0.0023*{}^{18}\delta$$

Now we return to Eq. (17) in the main text, applicable when the isotopic composition of measured air is different from the standard that is used. We restate it as

$$\Delta X_{tot} = X_{tot} \left[ 0.01111\left({}^{13}\delta_{air} - {}^{13}\delta_{st}\right) + 2*0.0023\left({}^{18}\delta_{air} - {}^{18}\delta_{st}\right) \right] \tag{A4}$$



**Supplemental links**

None

**Author contribution**

P. Tans derived the equations used to apportion total $CO_2$ into component isotopologues and their application. A.

5    Crotwell designed and built the calibration system. K. Thoning wrote software for operating the calibration system and managing the data.

**Competing interests**

The authors declare that they have no conflict of interest.

**Disclaimer**

10    None

**Acknowledgements**

We thank Thomas Mefford for running the calibration system, Duane Kitzis for standards preparation, and Ed Dlugokencky for reviewing this manuscript. In addition, we thank Sylvia Michel, Natalie Cristo, and the other members of the INSTAAR Stable Isotope Laboratory for isotopic measurements.





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





**Tables**

Table 1: Isotopic ratios of international standard reference materials.

| Reference Material | Ratio | Reference |
|---|---|---|
| $(^{18}O/^{16}O)_{VSMOW}$ | 0.0020052 | (Baertschi, 1976) |
| $(^{17}O/^{16}O)_{VSMOW}$[a] | 0.00038672 | (Assonov and Brenninkmeijer, 2003) |
| $(^{18}O/^{16}O)_{VPDB-CO2}$ | 0.002088349 | (Allison et al., 1995) |
| $(^{17}O/^{16}O)_{VPDB-CO2}$[b] | 0.00039511 | (Assonov and Brenninkmeijer, 2003) |
| $(^{13}C/^{12}C)_{VPDB}$[c] | 0.0112372 | (Craig, 1957) |
| $(^{2}H/^{1}H)_{VSMOW}$ | 0.00015576 | (Hagemann et al., 1970) |
| $(^{15}N/^{14}N)_{air-N2}$ | 0.0036765 | (Junk and Svec, 1958; Coplen, et al., 1992) |

[a] In other literature, it is possible to find different $^{17}O/^{16}O$ ratio values for these standard reference materials than those given here. However, for the determination of $^{17}O$ isotopic effects on atmospheric $CO_2$ measurements, differences from the values given in this table are insignificant.

[b] The $^{17}O/^{16}O$ ratio of VPDB-CO2 is derived from $[(^{18}O/^{16}O)_{VPDB-CO2}/(^{18}O/^{16}O)_{VSMOW}]^{0.528} * (^{17}O/^{16}O)_{VSMOW}$

[c] The value for $(^{13}C/^{12}C)_{VPDB}$ was revised to 0.011180 by Zhang et al. (1990). However the value given in this table is still widely used. For the determination of $^{13}C$ isotopic effects on atmospheric $CO_2$ measurements the difference is insignificant.



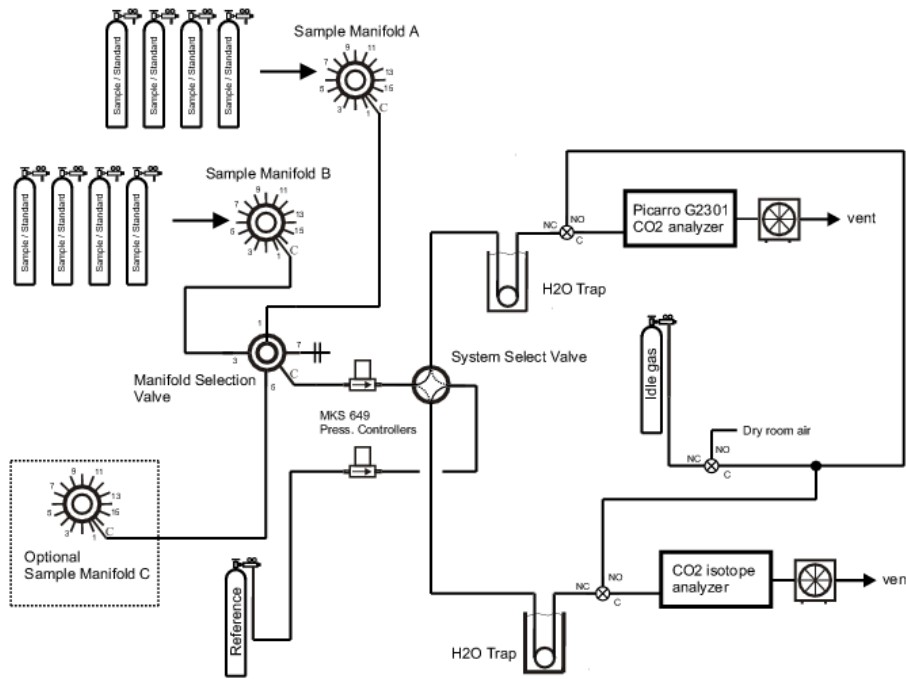

Figure 1: Schematic for the NOAA laser spectroscopic $CO_2$ calibration system. The CRDS analyzer is used with one of the $CO_2$ isotope analyzers which are interchangeable.





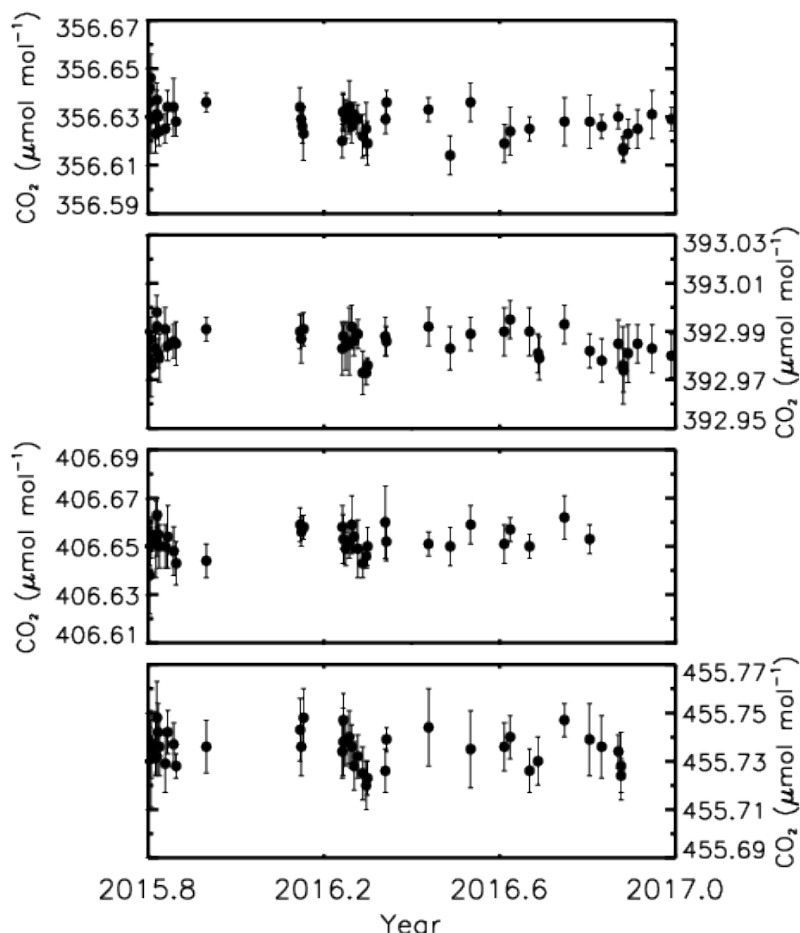

Figure 2: Total CO₂ calibration results for four target tanks measured on the laser spectroscopic CO₂ calibration system over approximately 1 year. The results span multiple gas handling system modifications. Values since April 2016 are on a consistent design. The average standard deviation for the 4 tanks is ± 0.007 µmol mol⁻¹.





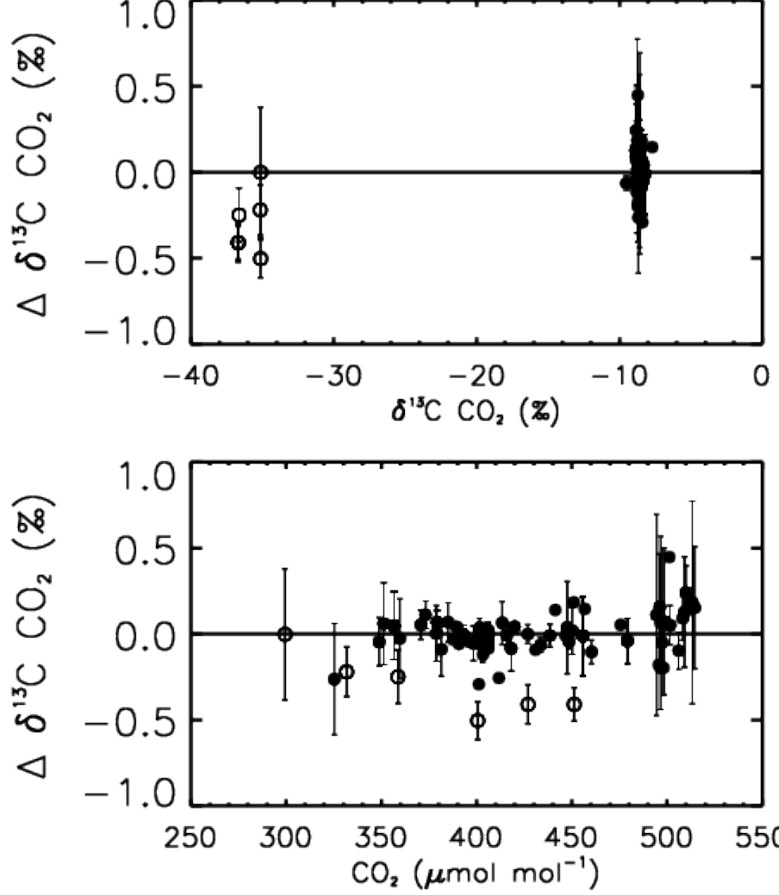

**Figure 3: Discrete samples from tertiary standards were collected in flasks and measured by INSTAAR. The average INSTAAR flask $\delta^{13}C$ result is compared to the average $\delta^{13}C$ tank calibration result on NOAA's laser spectroscopic $CO_2$ calibration system. Top panel is the difference (NOAA – INSTAAR) as a function of the INSTAAR $\delta^{13}C$ value and the bottom panel is the difference vs total $CO_2$. Error bars in both plots are the standard deviation of multiple calibration episodes by NOAA. INSTAAR uncertainties are typically ± 0.03‰ (68% CI) (Trolier et al., 1996) but do not account for problems with the collection of the discrete air sample. Highly depleted cylinders ($\delta^{13}C$ < -20‰) are shown with open circles in each panel.**





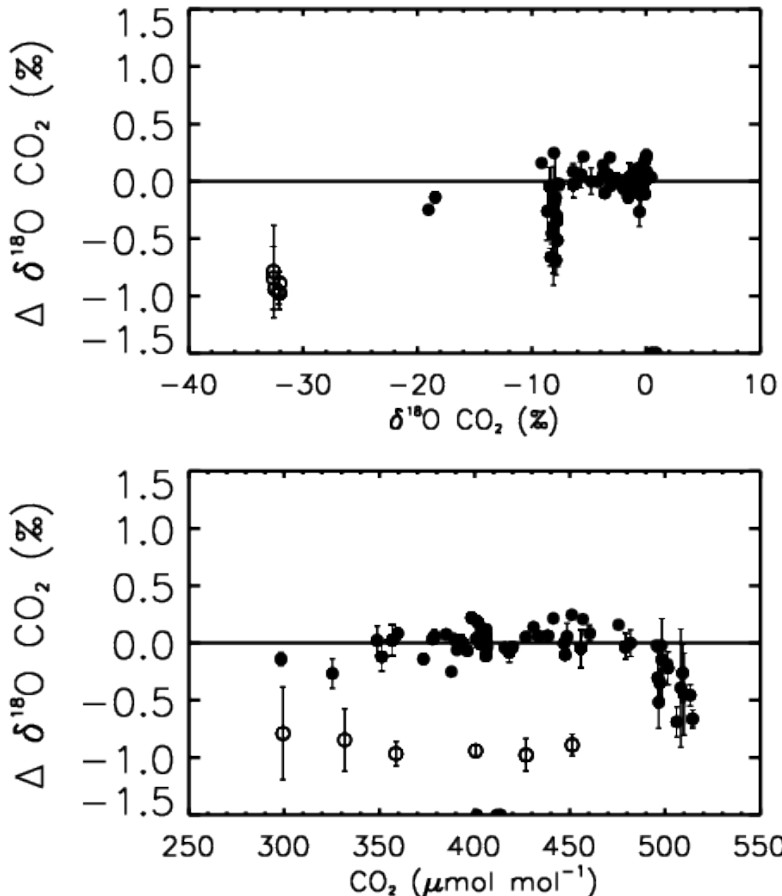

**Figure 4: The same as Figure 3 for δ¹⁸O. INSTAAR uncertainties are typically ± 0.05‰ (68% CI) (Trolier et al., 1996) but again do not account for problems with the collection of the discrete air samples. Differences greater than 1.5‰ are assumed to be caused by problems during discrete sample collection. These results are shown but are not included in the statistics. Highly depleted cylinders (δ¹⁸O < -20‰) are shown with open circles in each panel.**





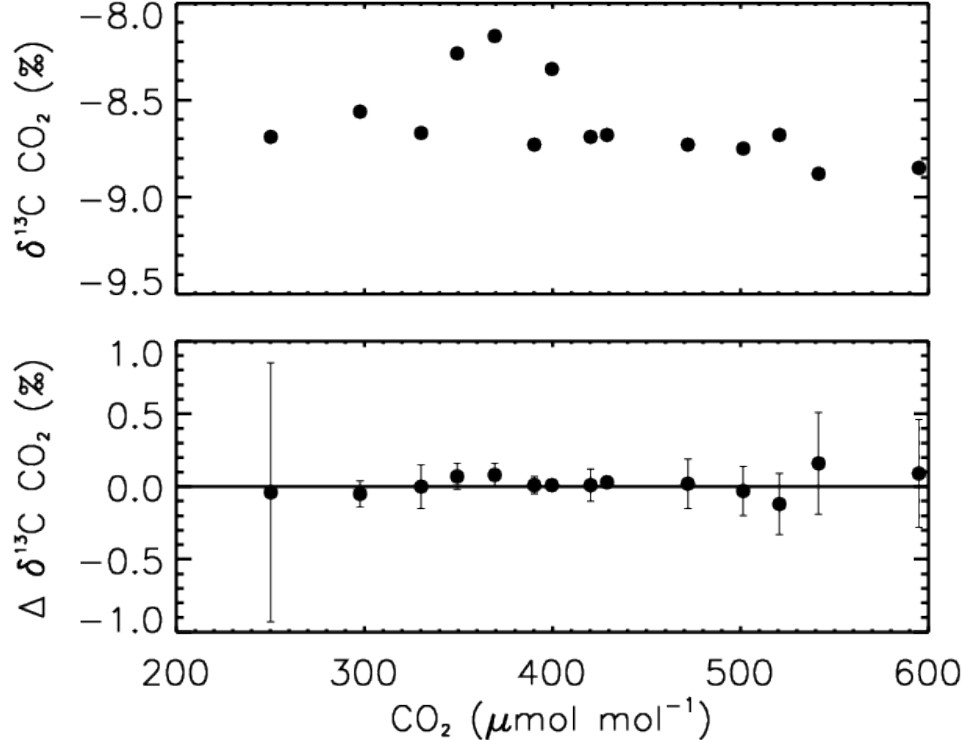

**Figure 5: Secondary standards used to calibrate the laser spectroscopic system have $\delta^{13}C$ and $\delta^{18}O$ values from direct measurement by INSTAAR and they have measured $\delta^{13}C$ and $\delta^{18}O$ from calibration on the laser spectroscopic system against the primary $CO_2$ standards. The top panel shows the INSTAAR $\delta^{13}C$ values as a function of $CO_2$. Uncertainties on the INSTAAR values (less than 0.02‰) are not visible. The bottom panel shows the difference between the NOAA and INSTAAR measurements of the secondary standards (NOAA – INSTAAR) also as a function of $CO_2$. Error bars are the standard deviation of three calibration episodes of the secondary standards vs the primary standards on the NOAA $CO_2$ calibration system.**





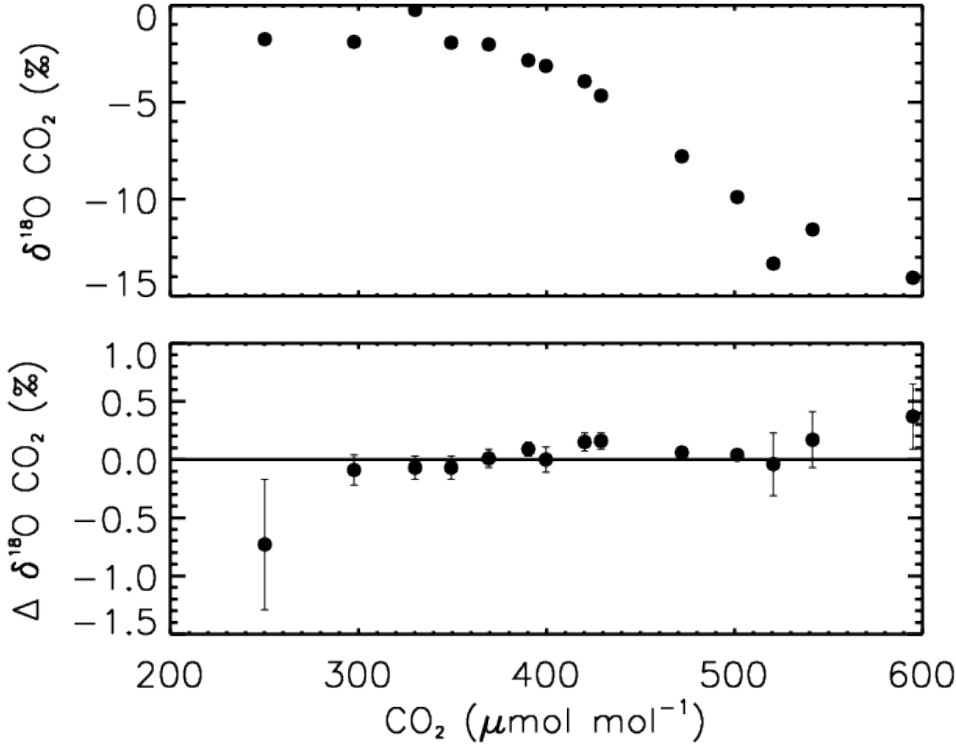

**Figure 6: The same as Figure 5 for $\delta^{18}O$. The $X_{CO_2}$ dependent depletion of $\delta^{18}O$ in cylinders above ambient $CO_2$ results from the depleted $\delta^{18}O$ of the spike gas.**



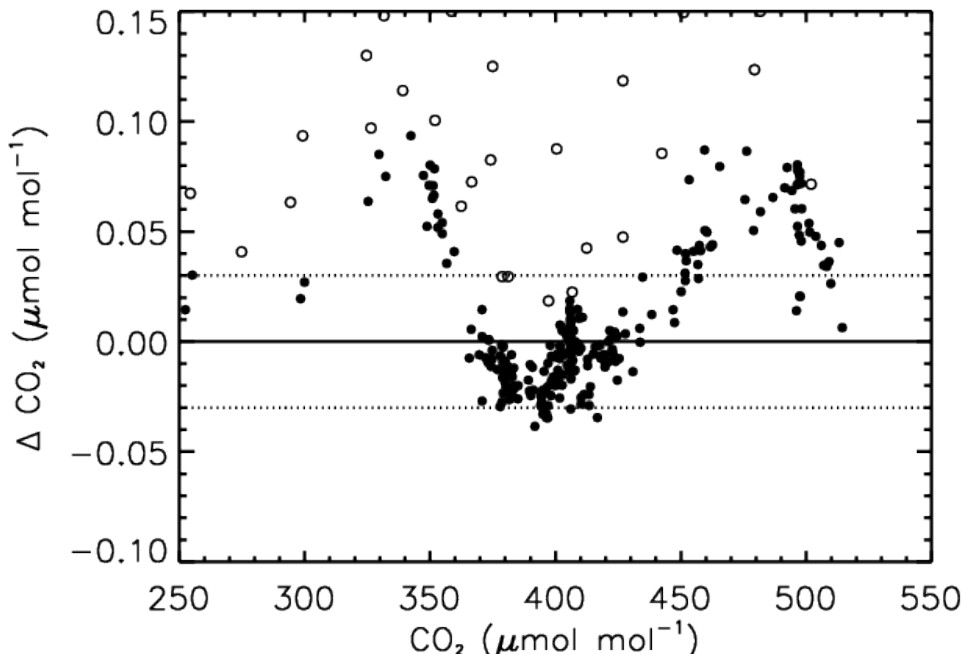

**Figure 7: The laser spectroscopic CO₂ calibration system was run in parallel with the NDIR CO₂ calibration system for approximately 6 months. The differences (average NDIR – average laser spectroscopic system) are plotted as a function of CO₂. Typical reproducibility of the NDIR measurements (±0.03, 68% CI) are shown with dashed lines. Highly depleted cylinders (δ¹³C < -20‰) are shown by open circles. These clearly indicate enhanced offsets due to the NDIR being somewhat sensitive to the isotopic composition differences between the samples and the standards used to calibrate the instrument.**