# Peer review of "Abundances of isotopologues and calibration of $CO_2$ greenhouse gas measurements"

_Atmospheric Measurement Techniques, 2017_

## Referee Comment (RC1) · Anonymous Referee #1 · 18 Mar 2017

Please see attached pdf file.

Please also note the supplement to this comment:
http://www.atmos-meas-tech-discuss.net/amt-2017-34/amt-2017-34-RC1-supplement.pdf

---

## Referee Comment (RC2) · Anonymous Referee #2 · 5 Apr 2017

Recommendation: Publish - minor revisions

General comments:

The manuscript "Abundances of isotopologues and calibration of CO2 greenhouse gas measurements" is well written and reports on a method and a new calibration system to account for differences in isotopic composition between primary CO2 reference standards. This is an important development and essential for addressing biases introduced from measurements sensitive to specific isotopologues. The authors point out that these developments can be applied to other molecules. Application to CH4 and N2O would be of further benefit to users of optical spectroscopy. This work is a valuable contribution and of significant interest to the atmospheric monitoring community. The document defines the state of the art for the CO2 calibration scale and it is

important that this information is in the public domain. I recommend publication subject to the following minor suggestions for revision:

• The section on calibration and system performance refers to measurements of delta13C and delta18O made at INSTAAR on the primary and secondary standards using IRMS. What was the reference used for these measurements and are these traceable to VPDB?

• The term "mole fraction" and "amount of substance fraction" are used interchangeably throughout. One of these terms should be used for consistency. The second sentence of the abstract mol/mol is the unit and should replace mole fraction. This also applies to the fourth paragraph of the introduction.

• I would suggest keeping all y-axis values on the left hand side of the figure and increasing the size of the interval (perhaps 3 y values per chart). It is not clear whether the bars on the data represent standard deviations or uncertainties. Is there any contribution from the change in composition in the $CO_2$ reference standards to the trend observed in figure 2? Is it assumed that the changes in $CO_2$ reference standards is negligible compared to the long term reproducibility of the facility?

• Figures 3 and 4 present the comparability of measurements at INSTAAR and the new calibration system. The offset of non-depleted tanks is attributed to the extrapolation of the calibration at INSTAAR. Is there any data to support this statement?

• Assuming uncertainties are symmetrical, the values presented throughout the manuscript (e.g. $\pm$ 0.007 $\mu$mol/mol in the caption to figure 2) do not require the $\pm$ sign.

• The caption to figures 5 and 7 are missing the term mole fraction (e.g. "The top panel shows the INSTAAR delta13C values as a function of $CO_2$ mole fraction.")

---

## Referee Comment (RC3) · Anonymous Referee #3 · 6 Apr 2017

The paper should be published with revisions. Laser based instruments measuring concentrations of gases in the atmosphere have developed rapidly in recent years, with the ability to measure individual isotopologues. At the same time the standards used for calibrating such instruments need to be adapted accordingly to allow for correct calibration, and the paper describes the progresses made. The paper should be improved by: a) improving terminology on quantities and units b) using conventionally used symbols for quantities in a number of equations c) using internationally accepted conventional values for isotope reference materials d) describing the impact of non-equilibrated CO2 in standards and the potential biases that may arise in isotope ratio measurements as a result e) reduction in the number of equations, with references to already published work f) full description of the traceability and uncertainty of isotope ratio measurements by both IRMS and Optically based techniques. These should also

be propagated through to measurements of mole CO2 mole fractions.

Specific comments: Page 1 line 14: 'units of mole fraction' is not a correct expression; mole fraction is a quantity not a unit. Correct to 'calculate the mole fraction of each component', expressed in units of $\mu$mol/mol Page 1 line 19: same issue as above with the use of 'mole fraction units'. Please correct. Page 2 line 34: correct 'units of mole fraction' also the symbol for mole fraction should be in italics, with normally lower case being used Page 3 line 9: what is calculated is the mole fraction of CO2 in air, not the ratio. Also 'ratio of moles' is not a correct term. Please correct. Page 4 line 10: 'the number of molecules of CO2 per mole of dry air', is not correct – it is a different quantity (which would be expressed in units of 1/mol) from mole fraction (expressed as mol/mol). Restructure the sentence avoiding this part of the phrase. Page 4 line 13: the authors should reference the fact that they are using the shorthand of the spectroscopy community (e.g. reference to HITRAN, see https://www.cfa.harvard.edu/hitran/molecules.html ). Also in this notation the convention is to write isotopologues as 628 and not 826. Please correct. Page 4 line 21: Equation 1 includes the factor 1000. This is not correct, delete the factor 1000. If needed add a phrase that delta values are often expressed in per mil, where the symbol ‰ means 0.001 Page 5 line 16 and subsequently: when quoting ranges these need to be written as -7.0 ‰ to -9.0 ‰ Also there needed to be a space between the number and ‰ i.e -9.0 ‰ and not -9.0‰ Please correct Page 6 line 4: depleted in 13C and not $\delta$13C. Please correct Page 6 Section 3: The authors should use conventional notation in this section, rather than introducing their own. In addition they should differentiate between quantities that are simple ratios and the ones that are fractions. See Santrock (1985) which is referenced in the paper, where the ratio of amounts of substance (abundance as used by authors) of two isotopes is demoted with the symbol R, whilst a fraction has been given the symbol F. In all cases symbols should be in italics following standard practice. Page 6 line 16: the equations should be numbered. The conventional symbol for an isotope ratio is R and not r. Page6 line 19: These are just fractions not 'redefined ratios'. The equations should be numbered Page 7 line 1: Often in papers VPDB-CO2 is shortened to VPDB, when a statement is

included explaining this. PDB is not used as a shorthand for VPDB, because it actually denotes the original PDB scale. Use VPDB if a short had notation is required. Page 7 line 3: (equation 4a) The value used for 13RVPDB is not the one recommended by the IAEA nor the WMO CCL for CO2 isotope ratios. A value of 0.01118 should be used see you reference Brand et al (2010). Similarly the values for 17R and 18R are not the same as for the Brand et al (2010) reference. Internationally accepted conventional values should be used- please correct. Page 7 line 13-15: This sentence is not necessary if equation 1 is corrected. Page 7 line 22: 'approximate the abundance as mole fraction' should be corrected to 'calculate the mole fraction'. Page 7 lines 19 onwards: The ratios measured in IRMS together with the convention already mentioned on Page 4 line 26, can be solved exactly to then calculate atomic isotopic abundances, and with simple probability theory (see Ref 1 in Santrock (1985)) and knowledge of the total CO2 mole fraction calculate the mole fraction of any of the 12 CO2 isotopologues in the gas. This section would be improved by replacing with reference to the Santrock(1985) paper and reference there in. Page 8/9 entire section: Whilst providing a nice description of probability theory, how is this any different from the Santrock paper in describing the distribution of isotopes among molecules at equilibrium is accurately described by a simple probability function, and reference 1 therein? The current text could simply be replaced by a reference. However, what the authors have not discussed and does not seem to be treated in this paper is that these equations are only exact when the gas is in equilibrium. The procedures used for making the WMO standards, especially historically, are likely to lead to a non-equilibrated gas i.e. by mixing two CO2 gases together with different isotopic compositions the resulting mixture does not have the distribution of isotopologues that would be predicted from the average atomic isotopic abundances of the mixture. The effect of this both for the spectroscopic and mass spectrometric methods applied in the paper should be evaluated and commented upon in order to confirm the authors' conclusions. Page 11 lines 21-23: The scale on which INSTAAR is measuring CO2 isotope ratios should be described, as well as the conventional values used for its scale. Is it its own realization of the VPDB or VPDB-LSVEC scale? The

measurement uncertainty of this realization should be described as well as any known bias form the WMO Scale for CO2 in air(JRAS).

Page 11 Entire Section: Several papers have been published describing approaches for calibrating optical system for isotope ratio measurements (Wen et al, Atmos. Meas. Tech. 2013 and Flores et al. Anal. Chem 2017) with the latter including uncertainty estimation of calibration procedures. The authors reference neither, nor do they provide a description of the uncertainty of their calibration or measurements procedures. A reference to previous descriptions of calibration procedures and an assessment of the measurement uncertainty should be added, which would then allow propagation of the uncertainty into mole fraction values. Page 12 and 13: The equations on these two pages are difficult to follow. It is not clear to the reviewer why the sum of all isotopologues is not included in the reported total CO2 mole fraction value. Accurate measurement of the 626 isotopologue, together with its isotope ratios and the assumed distribution of isotopes would allow the mole fractions of all other isotopologues to be calculated and their sum added to the 626 mole fraction to give total CO2. Page 16 line 14: no information on the uncertainty for the standards is given. Please add this. Page 17 line 10 and 11: It would be useful to know if INSTAAR are using a second reference material to control scale contraction effects to substantiate this conclusion. Page 17 lines 22-23. Reproducibility and uncertainty appear to be used as synonyms, which they are not. The author's should differentiate between the reproducibility and uncertainty, and an estimation of the measurement uncertainty would help in this respect. Page 26 Table 1: The currently internationally accepted conventional values for VPDB should be clearly identified in this Table.

---

## Author Comment (AC2) · 26 May 2017

Comments from referee 2 Recommendation: Publish - minor revisions General comments: The manuscript "Abundances of isotopologues and calibration of CO2 greenhouse gas measurements" is well written and reports on a method and a new calibration system to account for differences in isotopic composition between primary CO2 reference standards. This is an important development and essential for addressing biases introduced from measurements sensitive to specific isotopologues. The authors point out that these developments can be applied to other molecules. Application to CH4 and N2O would be of further benefit to users of optical spectroscopy. This work is a valuable contribution and of significant interest to the atmospheric monitoring community. The document defines the state of the art for the CO2 calibration scale and it is important that this information is in the public domain. I recommend publication subject

to the following minor suggestions for revision:

Comment: The section on calibration and system performance refers to measurements of delta13C and delta18O made at INSTAAR on the primary and secondary standards using IRMS. What was the reference used for these measurements and are these traceable to VPDB?

Response: The measurements by INSTAAR are traceable to VPDB. The INSTAAR scales were set using NBS-19 and NBS-20 carbonates and VSMOW, GISP and SLAP waters. Text has been added to clarify. Also, text has been added addressing the effects of the differences between INSTAAR and JRAS.

Comment: The term "mole fraction" and "amount of substance fraction" are used interchangeably throughout. One of these terms should be used for consistency. The second sentence of the abstract mol/mol is the unit and should replace mole fraction. This also applies to the fourth paragraph of the introduction.

Response: We have changed the text to be more consistent with terms. It was our understanding from Schwartz and Warneck (1995) that "amount of substance fraction" was the quantity and "mole fraction" was the unit. It seems this reading was wrong and the IUPAC compendium of chemical terminology (the Gold Book) lists "mole fraction" as a synonym of "amount of substance fraction". We use amount of substance fraction in the first instance with mole fraction in parenthesis and then mole fraction for the rest of the document.

Schwartz, S. E., Warneck, P., Units for use in atmospheric chemistry (IUPAC recommendations 1995). Pure and Appl. Chem., 67, 1377-1406, 1995.

IUPAC. Compendium of Chemical Terminology, 2nd ed. (the "Gold Book"). Compiled by A. D. McNaught and A. Wilkinson. Blackwell Scientific Publications, Oxford (1997). XML on-line corrected version: http://goldbook.iupac.org (2006) created by M. Nic, J. Jirat, B. Kosata; updates compiled by A. Jenkins.

[Figure]

Comment: I would suggest keeping all y-axis values on the left hand side of the figure and increasing the size of the interval (perhaps 3 y values per chart). It is not clear whether the bars on the data represent standard deviations or uncertainties. Is there any contribution from the change in composition in the $CO_2$ reference standards to the trend observed in figure 2? Is it assumed that the changes in $CO_2$ reference standards is negligible compared to the long term reproducibility of the facility?

Response: We find the y-axis labels easier to read when alternating sides so leave them this way. The error bars are the standard deviation of the 8 measurements per calibration episode. Text has been added to clarify. I don't think the apparent trend is significant. Subsequent measurements have shown step changes possibly due to regulators or subtle variations in the stability of the response of the instruments but not a significant trend in the target tanks. We updated the plot with more data collected since initial submission to make it clearer. We also include the standard deviation of the replicate calibration episodes for each tank individually in the caption. The reference tank is not used as a point in the calibration curve. It only corrects for slow drift between calibration episodes. Changing the reference tank does not affect the measurement results. The reference tank is in effect being calibrated while the calibration curve is being determined.

Comment: Figures 3 and 4 present the comparability of measurements at INSTAAR and the new calibration system. The offset of non-depleted tanks is attributed to the extrapolation of the calibration at INSTAAR. Is there any data to support this statement?

Response: The offsets of depleted tanks is roughly consistent with the INSTAAR offset from JRAS described by Wendeberg et al. (2013) and is contributed to scale contraction at INSTAAR. Text has been added to clarify.

Comment: Assuming uncertainties are symmetrical, the values presented throughout the manuscript (e.g. $\pm$ 0.007 $\mu$mol/mol in the caption to figure 2) do not require the $\pm$ sign.

Response: We leave the $\pm$ symbol in to prevent confusion and re-iterate that they are symmetrical ranges.

Comment: The caption to figures 5 and 7 are missing the term mole fraction (e.g. "The top panel shows the INSTAAR delta13C values as a function of CO2 mole fraction.")

Response: We leave the text as is to be succinct.

Please also note the supplement to this comment:
http://www.atmos-meas-tech-discuss.net/amt-2017-34/amt-2017-34-AC2-supplement.pdf
* * *
[Figure]

**Supplement:**

**Abundances of isotopologues and calibration of CO$_2$ greenhouse gas measurements**

Pieter P. Tans[1,3], Andrew M. Crotwell[2,3], and Kirk W. Thoning[1,3]

[1]Global Monitoring Division, Earth System Research Laboratory, National Oceanic and Atmospheric Administration, Boulder, Colorado, 80305, USA.
[2]Cooperative Institute for Research in Environmental Sciences, University of Colorado, Boulder, Colorado, 80309, USA.
[3]Central Calibration Laboratory, World Meteorological Organization Global Atmosphere Watch program (WMO/GAW)

*Correspondence to:* Pieter Tans (Pieter.Tans@noaa.gov)

**Abstract**

We have developed a method to calculate the fractional distribution of CO$_2$ across all of its component isotopologues based on measured $\delta^{13}$C and $\delta^{18}$O values. The fractional distribution can be used with known total CO$_2$ to calculate the amount of substance fraction (mole fraction) abundance of each component isotopologue in air individually, in units of mole fraction. The technique is applicable to any molecule where isotopologue-specific values are desired. We used it with a new CO$_2$ calibration system to account for isotopic differences among the primary CO$_2$ standards that define the WMO X2007 CO$_2$ in air calibration scale and between the primary standards and standards in subsequent levels of the calibration hierarchy. The new calibration system uses multiple laser spectroscopic techniques to measure mole fractions amount of substance fractions (in mole fractions units) of the three major CO$_2$ isotopologues ($^{16}$O$^{12}$C$^{16}$O, $^{16}$O$^{13}$C$^{16}$O, and $^{18}$O$^{12}$C$^{16}$O$^{16}$O$^{12}$C$^{18}$O) individually. The three measured values are then combined into total CO$_2$ (accounting for the rare unmeasured isotopologues), $\delta^{13}$C, and $\delta^{18}$O values. The new calibration system significantly improves our ability to transfer the WMO CO$_2$ calibration scale with low uncertainty through our role as the World Meteorological Organization Global Atmosphere Watch Central Calibration Laboratory for CO$_2$. Our current estimates for reproducibility of the new calibration system are ±-0.01 µmol mol$^{-1}$ CO$_2$, ±-0.2 ‰ $\delta^{13}$C, and ±-0.2 ‰ $\delta^{18}$O, all at 68 % confidence interval (CI).

**1 Introduction**

Long-term atmospheric monitoring of the greenhouse gases relies on a stable calibration scale to be able to quantify small spatial gradients and temporal trends. Small changes in trends and spatial gradients result from realignments in the strengths of emissions ("sources") and removals ("sinks") of the greenhouse gases. Inconsistent scale propagation to atmospheric measurements would give biased results from one monitoring station or network to the next that would be attributed incorrectly to sources/sinks by atmospheric transport models. Preventing biased results from various national monitoring networks enables improved understanding of the carbon cycle and its response to

human intervention and climate change. It has now become even more important as countries have pledged emissions reductions. The capability to independently and transparently verify emission reductions could be helpful for creating trust in the agreements.

The World Meteorological Organization Global Atmosphere Watch (WMO GAW) program facilitates cooperation and data sharing among participating national monitoring programs. Atmospheric data collected over small regional scales is difficult to interpret without global coverage that provides boundary conditions and also insight into influences outside of the region. WMO GAW sets stringent compatibility goals so that measurements from independent laboratories can be combined in scientific studies. This greatly enhances the value of the individual data sets since it allows processes occurring within the region to be better distinguished from processes external to that region. In combining data sets it is imperative that systematic biases between the monitoring networks be small enough that they do not influence scientific interpretation of patterns and strengths of sources and sinks. For $CO_2$, the consensus of the scientific community is that network biases should be below 0.1 $\mu$mol mol$^{-1}$ in the Northern Hemisphere but less than 0.05 $\mu$mol mol$^{-1}$ in the Southern Hemisphere where atmospheric gradients are smaller (WMO, 2016). One initial requirement to accomplishing this network compatibility goal is that measurements are comparable, that is each independent laboratory uses a single common calibration scale. The use of a single calibration scale makes spatial gradients and temporal changes insensitive to large components in the full uncertainty budget of the scale itself. The calibration scale must be maintained indefinitely to ensure that measurements from various organizations are compatible and that measurements over long time scales can be directly compared to infer rates of changes. The WMO GAW has designated a single laboratory as the Central Calibration Laboratory (CCL) whose mission is to maintain a stable reference scale over time and to disseminate it to other organizations with very low uncertainty (WMO, 2016). The WMO GAW program has two ways to help individual laboratories maintain close ties to the WMO scale. One is a "round-robin" comparison where calibrated cylinders are sent from the CCL to individual laboratories. The values assigned by the CCL are unknown to the laboratories, and they measure them as unknowns. At the end all values are compared with the values assigned by the CCL. This occurs once every few years. The second method is the establishment of several World Calibration Centers (WCC). Each of them provides assistance in their own region with general quality control of air measurements and calibrations.

The WMO X2007 $CO_2$ in air calibration scale is maintained and propagated by the National Oceanic and Atmospheric Administration, Earth System Research Laboratory, Global Monitoring Division (NOAA) in its role as the WMO GAW CCL for $CO_2$. The scale is defined by 15 primary standards covering the range 250 – 520 $\mu$mol mol$^{-1}$. The primary standards are modified real air standards made in the early 1990's by filling cylinders with dried ($H_2O < 2$ $\mu$mol mol$^{-1}$) natural air at Niwot Ridge, CO, USA, a remote site at approximately 3040 masl in the Rocky Mountains. It typically is exposed to clean tropospheric air and is only occasionally influenced by local sources. $CO_2$ abundances of the primary standards were adjusted either by scrubbing $CO_2$ from a portion of the natural air using a trap with sodium hydroxide coated silica to lower the $CO_2$ or by spiking with a mixture of $CO_2$ in air (approximately 10%) to raise it. This differs slightly from the current practice of targeting lower than local ambient $CO_2$ by diluting

with ultrapure air, $CO_2$ nominally < 1 µmol mol$^{-1}$ (Scott Marrin Inc., Riverside CA, USA) (Kitzis, 2009).

The assigned values of the primary standards come from repeated (approximately every two years) manometric determinations of the primary standards. The manometer,  described fully in Zhao et al. (1997), essentially measures the $CO_2$  (mole fraction)  by accurate measurement of pressure and temperature (both traceable to SI) of a whole air sample and then of pure $CO_2$ extracted from the whole air sample in fixed volumes. The manometer is enclosed in an oven capable of maintaining a constant temperature (within ±0.01 °C). A 6 L volume borosilicate glass bulb (the large volume) is flushed with the dried whole air sample (dew point < -70 °C) and the pressure and temperature are measured after the large volume temperature equilibrates with the oven. $CO_2$ plus $N_2O$ and trace amounts of $H_2O$ are cryogenically extracted from the whole air sample using two liquid nitrogen cold traps. $CO_2$ and $N_2O$ are then cryogenically distilled from $H_2O$ and transferred to a ~10 mL cylindrical glass vessel (the small volume). Pressure and temperature of the small volume are measured after the oven temperature has stabilized following the transfer. The volume ratio of the small to large volumes, determined by an off-line sequential volume expansion experiment, is used with the measured temperatures and pressures to calculate the ratio of moles $CO_2$ (corrected for the $N_2O$) to total moles of air in the sample using the virial equation of state. Uncertainty of the method is approximately ±0.1 µmol mol$^{-1}$ (68 % CI) at 400 µmol mol$^{-1}$ (Zhao et al., 2006,  Hall, et al., in preparation ).

The subject of this paper is the transfer of the scale to lower level standards and its uncertainty. We do not discuss the total uncertainty of the primary scale itself. It is described in a separate paper (Hall et al., in preparation). The transfer of the scale from primary to secondary standards and hence to tertiary standards (which are used as working standards by NOAA and delivered to other organizations) has been done historically using nondispersive infrared absorption spectroscopy (NDIR). The secondary standards are used to prolong the lifetime of the primary standards. The current primary standards have been in use for nearly 25 years and provide a consistent scale over that time period. All measurements by NOAA and WMO GAW contributing programs are directly traceable to this single set of primary standards through a strict hierarchy of calibration.

The transfer of the scale from primary to secondary standards has typically been done using a subset of 3 or 4 primary standards rather than the entire set of 15 primary standards. This was done because we wanted to perform a local curve fit of the non-linear NDIR response while also minimizing use of the primary standards. The subset of primary standards chosen was a function of the expected $CO_2$  in the secondary standards and was designed to closely bracket the expected values with a small range of $CO_2$ in the primary standards. The relatively large uncertainty of the individual manometric assigned values would potentially introduce significant biases due to the use of subsets of primary standards. To prevent these biases, the individual manometrically assigned values of the primary standards were corrected based on the residuals to a consistency fit of almost all primary standards (usually without the highest and the lowest primary) run on the NDIR. The re-assigned values (average manometer value minus the residual) were assumed to be the best assigned value for the primary standards. This in theory

**Comment [pt1]:** I don't think "absolute" is used in metrology. Instead we can insert that pressure and temperature are traceable to SI.

**Comment [AMC2]:** Pieter, check this wording and change if you think it should be different. This is a correct description. The equation of state, with known pressures, temperatures, and a volume ratio, results in a ratio of number of moles of gas in the large and in the small volume.

**Comment [pt3]:** We need to state this clearly up front. There was some confusion among reviewers about it.

should allow the use of subsets of the primary standards when transferring the scale from primary to secondary. In practice, as will be shown, there are still possible biases due to the grouping of primary standards based on expected $CO_2$ of the secondary standards. Tertiary standards were calibrated similarly against closely spaced subsets of secondary standards that bracketed the expected values of the tertiary standards.

5 New analytical methods developed over the last several years have greatly improved the ability of monitoring stations to measure $CO_2$. These new analytical techniques and improved diligence of monitoring network staff are pushing the uncertainties of measurements lower and improving the network compatibility. Current  reproducibility of standards using the NDIR calibration system is 0.03 μmol mol⁻¹ (68 % CI) ("Carbon Dioxide WMO Scale", 2017). This is a significant component of the targeted 0.1 μmol mol⁻¹ (or 0.05 μmol mol⁻¹ in the

10 Southern Hemisphere) network compatibility goal (WMO, 2016). Improvements in the scale propagation uncertainty would help monitoring programs achieve the compatibility goals. We have therefore undertaken to improve our calibration capabilities and to address key uncertainty components of the scale transfer. These key components are the reproducibility of the scale transfer, the potential for mole fraction dependent biases, and   the potential issues we describe in this paper relating to the isotopic composition of

15 the primary standards and subsequent standards in the calibration hierarchy.

**2 Isotopic influence on $CO_2$ measurement**

The WMO $CO_2$ mole fraction scale is defined as the number of moles of $CO_2$ per mole of dry air, without regard to its isotopic composition. An isotopologue of $CO_2$ has a specific isotopic composition. The five most abundant

20 $CO_2$ isotopologues, in order of abundance, are: $^{16}O^{12}C^{16}O$, $^{16}O^{13}C^{16}O$,  $^{16}O^{12}C^{18}O$,  $^{16}O^{12}C^{17}O$ and  $^{16}O^{13}C^{18}O$ (referred to in equations in this work by the HITRAN (Rothman, 2013) shorthand notations 626, 636, 628, 627, and 638 respectively). For $CO_2$ the two oxygen positions are equivalent due to the symmetry of the molecule so the position of the oxygen isotopes does not matter. The abundance of the radioactive $^{14}C$ relative to $^{12}C$ is ~$10^{-12}$; which is too small to be of significance in this context. Analysts need to take

25 into account differences in the relative sensitivity of their analyzers to different isotopologues (or isotopomers, see below) as well as differences in the isotopic composition of sample and standard gases.

Isotopic composition is typically measured by isotope ratio mass spectroscopy (IRMS) and is reported as the difference in the minor isotope to major isotope ratio (i.e. $^{13}C/^{12}C$) from the ratio of an accepted standard reference material. For example, the $^{13}C$  isotope delta value ($\delta^{13}C$) is defined as:

$$\delta^{13}C = \frac{\left(\tfrac{^{13}C}{^{12}C}\right)_{Sample} - \left(\tfrac{^{13}C}{^{12}C}\right)_{Standard}}{\left(\tfrac{^{13}C}{^{12}C}\right)_{Standard}} * 1000$$

(1)

Where $(^{13}C/^{12}C)_{sample}$ and $(^{13}C/^{12}C)_{standard}$ are the $^{13}C$ to $^{12}C$ isotopic abundance ratios for the sample and the standard reference material respectively. The international accepted scale for $^{13}C$ is the Vienna Pee Dee Bellemnite (VPDB) scale, realized as calcium carbonate. Oxygen isotopic ratios ($^{18}O/^{16}O$ or $^{17}O/^{16}O$) in $CO_2$ are described with a similar isotope delta notation relative to an accepted reference material. Isotope delta values for carbon and oxygen are typically reported in units of per mil (‰) by multiplying Eq. (1) by 1000. For many applications, the $^{17}O$ isotope is not actually measured but is assumed to follow a mass dependent relationship with $^{18}O$ where $\delta^{17}O \approx 0.528 * \delta^{18}O$. This approximation is adequate for the purpose of defining the oxygen isotopic effects on atmospheric $CO_2$ measurements. For more detailed descriptions of this relationship see Santrock et al. (1985), Assonov and Brenninkmeijer (2003), Brand et al. (2010) and references therein. Oxygen isotopes can be related to either Vienna Standard Mean Ocean Water (VSMOW) or to VPDB-$CO_2$, with the latter commonly used in the atmospheric $CO_2$ community. The VPDB-$CO_2$ scale relates to the $CO_2$ gas evolved from the calcium carbonate material itself during the reaction with phosphoric acid and accounts for oxygen fractionation that occurs during the reaction (Swart et al., 1991). In this paper all oxygen isotope values are referenced to the VPDB-$CO_2$ scale unless otherwise noted.

$CO_2$ analysers are not equally sensitive to the isotopologues of $CO_2$. For example, gas chromatography where $CO_2$ is reduced to $CH_4$ and detected with a flame ionization detector (GC-FID) (Weiss, 1981) is equally sensitive to all isotopologues whereas laser based absorption techniques that measure an absorption line from the single major $^{16}O^{12}C^{16}O$ isotopologue are blind to all of the minor isotopologues. NDIR instruments are much more complicated in their response to the various minor isotopologues of $CO_2$. Most NDIR analyzers use an optical band pass filter to limit the wavelengths of light reaching the detectors. These filters often exclude part of the absorption bands of the minor isotopologues (e.g. Tohjima et al. 2009), but are more sensitive to the $^{16}O^{13}C^{16}O$ lines within the pass band because absorption of the much stronger $^{16}O^{12}C^{16}O$ lines is partially saturated. The width and shape of the transmission window of the filter is generally not identical between instruments. Tohjima et al. (2009) found significant differences in the sensitivity to the minor isotopologues between three different LI-COR NDIR analyzers. In addition, Lee et al. (2006) found the response of a Siemens ULTRAMAT 6E NDIR analyzer to be almost completely insensitive to the minor isotopologues.

The range of $\delta^{13}C$ and $\delta^{18}O$ encountered in the background atmosphere (~ -7.0 to -9.0 ‰ $\delta^{13}C$ and 2 to -2 ‰ $\delta^{18}O$) is too small to cause a significant bias on the total $CO_2$ measurements with any of these techniques. At 400 µmol mol$^{-1}$ total $CO_2$, neglecting $\delta^{13}C$ values leads to errors of 0.0044 µmol mol$^{-1}$ per 1 ‰, and neglecting $\delta^{18}O$ values leads to errors of 0.0018 µmol mol$^{-1}$ per 1 ‰. A problem arises however when standards with significantly different isotopic

compositions from the atmosphere are used to calibrate instruments that have partial or no sensitivity to the minor isotopologues. This occurs when standards are made from fossil fuel sourced $CO_2$ (such as  from combustion of oil or natural gas) which results in significant depletion in $^{13}C$ and $^{18}O$ (Andres et al., 2000; Schumacher et al., 2011).

5    In the past we have neglected the dependency of the NDIR response to isotopic composition during scale transfer. The manometer measurement of the primary standards is not sensitive to isotopic composition, all isotopologues are included in the total. However, the primary standards have a range of $\delta^{13}C$ and $\delta^{18}O$ values (-7 ‰ to -18 ‰ $\delta^{13}C$ and 0 to -15 ‰ $\delta^{18}O$) with higher $CO_2$ standards being more depleted due to the use in the early 1990's of a spike gas that was isotopically depleted. This probably introduced a slight bias in the results when the scale was transferred to

10   secondary standards (often with ambient isotopic values) via NDIR measurements. It was assumed that the bias was small relative to the measurement noise in the NDIR analysis.

We intend to provide standards to the atmospheric monitoring community with isotopic values similar to the background atmosphere by using natural air whenever possible. To adjust the $CO_2$ content in the natural air standards, the current practice is to dilute using essentially $CO_2$ free natural air (ultrapure air, Scott

15   Marrin, Inc. Riverside CA, USA) or enrich using high $CO_2$ (10 – 20 %) spike gases with $\delta^{13}C \approx$ -9 ‰ and $\delta^{18}O \approx$ -30 ‰. The $\delta^{13}C$ isotopic composition of the resulting mixture is not significantly different from ambient background air. Currently, urban air highly enriched in $CO_2$ would have $\delta^{13}C$ values  lower than the spiked standards of similar $CO_2$ made by us. However, the WMO scale is designed to track the slow isotopic depletion of background air as the global burden of $CO_2$ increases over the next decades due to burning of fossil fuels rather than

20   approximate the composition of air influenced by local emission sources. We started using the isotopically correct spike gases in November 2011. Background atmospheric $\delta^{18}O$ is not well matched with the current spike gases or the historical spike gases ($\delta^{18}O \approx$ -30 to -40 ‰) and does result in depleted $\delta^{18}O$ values in cylinders that are spiked to targeted values above local ambient values. It is also our goal to provide calibration results that incorporate a characterization of the main

25   isotopologues and accounts for isotopic differences among the primary standards and between the primary standards and measured cylinders through the calibration hierarchy as proposed by Loh et al. (2011). This will ensure that the transfer of the WMO scale by distributing calibrated cylinders is not biased by isotopic differences and will provide users of the distributed cylinders the information required to account for isotopic effects on their own measurement systems.

30

Comment [pt6]: We have already said this

Comment [pt7]: Why don't we state what the value is? We could do that in the previous paragraph

Comment [AMC8]: Agreed, added approximate $\delta^{18}O$ value above

**3 Two different ways to define isotopic ratios and notation conventions**

In order to estimate the influence of isotopic composition differences on $CO_2$ measurements and to develop a precise method for calibration transfer that takes isotopic composition into account we first introduce the "mole fraction" notation for isotopic ratios in molecules. The conventional definitions of atomic isotopic ratios ($r$) are:

$$^{13}r \stackrel{\text{def}}{=} \frac{^{13}C}{^{12}C} \qquad ^{18}r \stackrel{\text{def}}{=} \frac{^{18}O}{^{16}O} \qquad \text{etc.}$$

As used here the symbols $^{13}C$, $^{18}O$, etc. stand for amounts. It will simplify derivations below if we use isotopic ratios as amount ratios relative to all carbon, oxygen, etc., similar to mole fractions in air. We give these  isotope-amount fractions the symbol "$x$" instead of "$r$".

$$^{13}x \stackrel{\text{def}}{=} \frac{^{13}C}{^{12}C + ^{13}C} \qquad ^{18}x \stackrel{\text{def}}{=} \frac{^{18}O}{^{16}O + ^{17}O + ^{18}O} \quad ==> \quad ^{16}x = 1 - \frac{^{17}O + ^{18}O}{^{16}O + ^{17}O + ^{18}O}$$

These definitions lead to the following relationships:

$$^{13}x = \frac{^{13}r}{1 + ^{13}r} \qquad ^{13}r = \frac{^{13}x}{1 - ^{13}x} \tag{2}$$

The equivalents for oxygen are:

$$^{17}x = \frac{^{17}r}{1 + ^{17}r + ^{18}r} \qquad ^{17}r = \frac{^{17}O}{^{16}O} = \frac{^{17}O/(^{16}O + ^{17}O + ^{18}O)}{^{16}O/(^{16}O + ^{17}O + ^{18}O)} = \frac{^{17}x}{1 - ^{17}x - ^{18}x} \tag{3}$$

and similarly for $^{18}x$ and $^{18}r$.

From here on we will abbreviate  VPDB-CO2 as VPDB to keep the notation manageable. Using Table 1 and these conventions gives us

$$^{13}x_{\text{VPDB}} = 0.0110564 \tag{4a}$$

$$^{17}x_{\text{VPDB}} = 395.11 \ 10^{-6} / (1 + 2088.35 \ 10^{-6} + 395.11 \ 10^{-6}) = 394.1 \ 10^{-6} \tag{4b}$$

$$^{18}x_{\text{VPDB}} = 2088.35 \ 10^{-6} / (1 + 2088.35 \ 10^{-6} - + 395.11 \ 10^{-6}) = 2083.2 \ 10^{-6} \tag{4c}$$

Isotopic ratio measurements have always been expressed in terms of their (typically) small difference from the standard reference materials, in the so-called delta notation:

$$^{13}\delta \stackrel{\text{def}}{=} (^{13}r - ^{13}r_{\text{VPDB}})/^{13}r_{\text{VPDB}} = ^{13}r/^{13}r_{\text{VPDB}} - 1, \text{ so that } ^{13}r - ^{13}r_{\text{VPDB}} = ^{13}r_{\text{VPDB}} \ ^{13}\delta \tag{5}$$

and similarly for $^{17}\delta$ and $^{18}\delta$

By analogy we define for the amount fractions $x$ ratios R:

$$^{13}\Delta \overset{\text{def}}{=} (^{13}x - {}^{13}x_{\text{VPDB}})/{}^{13}x_{\text{VPDB}} = {}^{13}x/{}^{13}x_{\text{VPDB}} - 1, \text{ so that } {}^{13}x - {}^{13}x_{\text{VPDB}} = {}^{13}x_{\text{VPDB}} \, {}^{13}\Delta \qquad (6)$$

and similarly for $^{17}\Delta$ and $^{18}\Delta$

In the above (Eqs. (5) and (6)) and the rest of this work we will express $\delta$ and $\Delta$ as small numbers, not in the "permil" (‰) notation, in which every delta value is multiplied by 1000. For example δ=0.020 would normally be written as 20 permil or 20 ‰. To keep the notation economical and the paper more readable we introduced simplified notations such as $^{13}r$ and $^{13}\delta$ instead of $r(^{13}\text{C}/^{12}\text{C})$ and $\delta^{13}\text{C}$ in equations. This produces no ambiguities. In addition, in this paper we need to distinguish between isotope-amount fractions *within* $CO_2$ (denoted "$x$" above in accordance with Coplen (2011)) of isotopes (and isotopologues) from mole fraction in air. We normally denote mole fraction in air by "$x$" or "$X$", but here we use "$y$" (in accordance with notation recommendations for gas mixtures (IUPAC, 2006)) to distinguish mole fraction in air from isotope or isotopologue amount fraction. For example, $x(636)$ is the amount fraction of $^{16}\text{O}^{13}\text{C}^{16}\text{O}$ to all isotopologues of $CO_2$ whereas $y(636)$ is the mole fraction of the $^{16}\text{O}^{13}\text{C}^{16}\text{O}$ isotopologue in air.

**4 Fractional abundances of isotopologues in molecules.**

Converting measured $\delta^{13}\text{C}$ and $\delta^{18}\text{O}$ values into $^{16}\text{O}^{13}\text{C}^{16}\text{O}$ (denoted as 636) and $^{18}\text{O}^{12}\text{C}^{16}\text{O}$ $^{16}\text{O}^{12}\text{C}^{18}\text{O}$ (826) isotopologue abundances is not straightforward due to the rare $^{17}\text{O}^{12}\text{C}^{16}\text{O}$ (726) and doubly substituted isotopologues. Isotope ratio mass spectrometry (IRMS) determines $\delta^{13}\text{C}$ and $\delta^{18}\text{O}$ values by measuring molecular mass 45/44 and mass 46/44 ratios, with appropriate corrections for interfering masses, relative to a standard reference material. These mass ratios can be used with the accepted isotopic ratios of the standard reference materials to *approximate* the abundance as amount mole fraction ($x$ $X$) of the three main isotopologues in $CO_2$ using:

$$x(636) \cong {}^{13}x, \quad x(628) \cong 2 \ast {}^{18}x, \quad x(626) \cong 1 - x(636) - x(628) \qquad (7)$$

The oxygen abundance ratio is multiplied by a factor of two in Eq. (8 7) to convert the amount fractions isotopic ratios from atomic abundance (i.e. $^{18}\text{O}/^{16}\text{O}$) into molecular abundance. The approximations in Eqs. (7) (9) ignore the contribution of the oxygen isotopes to $X$ $x(636)$ and of $^{13}\text{C}$ to $X$ $x(826$ 628), as well as the portion of the total composed of the rare isotopologues. Depending on the level of uncertainty desired this may or may not be acceptable. As the WMO GAW CCL for $CO_2$, NOAA is obligated to minimize biases in the $CO_2$ calibration scale, and therefore we will correctly account for the apportionment of $CO_2$ through all isotopologues. The same technique was developed independently by Flores et al. (2017) for use in calibrating spectroscopic instruments for $\delta^{13}\text{C}$ and $\delta^{18}\text{O}$ measurements. Here our focus is on total $CO_2$ measurements that account for isotopic differences between standards.

We start by assuming a purely statistical distribution of $^{13}$C, $^{18}$O, and other atoms when putting together a molecule starting from atomic amount fractions as given for standard reference materials (Table 1), namely, that the probability of picking a particular isotope is not affected by what is picked before or later. In general the other picks can affect the probability a little (called "clumped" isotopes), so that the thermodynamic abundances are slightly different from the statistical distribution. We will ignore that for now, and construct a purely statistical baseline distribution for the reference. It is important to note that thermodynamic and kinetic fractionation effects are reflected in actual measured delta values and fractionation factors relative to the agreed upon reference material. Thus the probability of picking a $^{13}$C atom for a carbon position is defined as simply $^{13}R_x$. However, a molecule may have more than one position for C, O, N, etc. For example, suppose there are N chemical positions for a particular atom in a molecule and we want to define the probability of M of those positions being filled with one particular isotope (denoted isotope a). If the locations of the M, as a subset of N, do not matter, as is the case for symmetrical molecules like $CO_2$ and $CH_4$, we could call the N positions equivalent. In that case the probability is

$$P = \binom{N}{M} * x_a{}^{M} * x_b{}^{N-M} \tag{8}$$

$x_a$ is the amount fraction of isotope a, $x_b$ is the amount fraction of other isotopes ($x_b = 1 - x_a$). The first term in Eq. (8) is the statistical weight which equals the number of combinations (a statistical term, the order of picking the M does not matter) of M out of N, given as

$$\frac{N!}{M!(N-M)!} \stackrel{\text{def}}{=} \binom{N}{M} \tag{9}$$

N! is the factorial notation, $N! \stackrel{\text{def}}{=} 1 * 2 * 3 * \ldots \ldots (N-1) * N$, with the special case $0! \stackrel{\text{def}}{=} 1$

Example: there are two equivalent positions for a single $^{18}$O in $CO_2$, namely $^{18}O^{12}C^{16}O$ and $^{16}O^{12}C^{18}O$, jointly denoted as "628" (one $^{18}O^{16}O$, one $^{12}C$, one $^{16}O^{18}O$), so that the statistical weight is

$$\binom{2}{1} = \frac{2!}{1! * (2-1)!} = 2 \, .$$

Or for methane, a single or double substitution of deuterium ($^{2}$H) for $^{1}$H has respective statistical weights:

$$\binom{4}{1} = \frac{4!}{1! * (4-1)!} = 4 \qquad \binom{4}{2} = \frac{4!}{2! * (4-2)!} = 6$$

It should be noted that whether positions can be considered equivalent depends on the symmetry of the molecule and the measurement method. For example for nitrous oxide the two positions for N in NNO would be equivalent when mass 45 (one $^{14}$N, one $^{15}$N, one $^{16}$O) is measured in a mass spectrometer but they are not when an optical absorption method is used because the spectrum of $^{14}N^{15}N^{16}O$ is different from $^{15}N^{14}N^{16}O$. In the latter case we need to keep

separate track of the probabilities, denoted below as "P", of these two isotopomers. Isotopomers have the same number of specific isotopes, but they differ in their position in the molecule.

The probability for any particular $CO_2$ isotopologue is the product of the probability of picking the carbon isotope and the probability of picking the oxygen isotopes. Each of these probabilities is determined using Eq. (8). For example, the probability for the  $^{16}O^{13}C^{18}O$ isotopologue is the probability of picking one $^{13}C$ isotope for one carbon position times the probability of picking one $^{18}O$ isotope for one of the two oxygen positions and one $^{16}O$ for the other.

The equations below give the probabilities for individual $CO_2$ isotopologues. When the isotopic compositions of the standard reference materials (VPDB in Table 1) are filled in we obtain the numbers after the "=>" sign.

$$P(626) = (1-^{13}x)*(1-^{17}x-^{18}x)^2 \qquad => 0.98404985 = 1-0.01595015 \qquad (10)$$

$$P(636) = ^{13}x*(1-^{17}x-^{18}x)^2 \qquad => 0.01101688 \qquad (11)$$

$$P(628) = (1-^{13}x)*2*^{18}x*(1-^{17}x-^{18}x) \qquad => \qquad 0.00411273$$
$$(12)$$

The sum of the above three major abundances is $0.99916 = 1- 0.00083$

$$P(627) = (1-^{13}x)*2*^{17}x*(1-^{17}x-^{18}x) \qquad => \qquad 0.00077754$$
$$(13)$$

The sum of the above four major abundances is $0.99993 = 1- 0.00006$

$$P(638) = ^{13}x*2*^{18}x*(1-^{17}x-^{18}x) \qquad => 4.59513 \cdot 10^{-5}$$

and so on, with progressively smaller probabilities. The sum of all probabilities equals 1, which was verified digitally in double precision. This example was for VPDB, but in any population of $CO_2$ molecules, (i.e. in a sample or standard cylinder) probabilities determined from the isotope-amount fractions of the population equate to the  fractional abundance of each isotopologue.

**5 An expression for potential effects of isotopic mismatches on measurements of $CO_2$**

In this section we derive some practical expressions for biases, and corrections, resulting from isotopic mismatches if they are ignored, for the case of $CO_2$. Similar considerations apply to other greenhouse gases such as $CH_4$, $N_2O$, etc. Such corrections can be generally applied to $CO_2$ measurements if desired. The unknown quantity of $CO_2$-in-air that we intend to measure is called "measurand". It can be a real air sample or an intermediate transfer standard.

one isotopologue, or equally sensitive to all isotopologues, or something in between. Here we give an example for an instrument that quantifies the mole fraction of total $CO_2$ in air, denoted $y_{CO2}$, by measuring only one isotopologue, namely $^{16}O^{12}C^{16}O$. We normally denote the species mole fraction or number fraction by "*x*" or "*X*", but here we want to distinguish it from isotope amount fractions by using the symbol "*y*" which is in accordance with notation recommendations for gas mixtures. We assume that the instrument is calibrated by a $CO_2$ standard with amount fractions $^{13}x_{VPDB}$ and $^{18}x_{VPDB}$ of the two main isotopologues, corresponding to the international VPDB reference points for $^{13}C$ and $^{18}O$. In almost all cases deviations of $^{17}x$ from VPDB are tightly correlated with deviations of $^{18}x$ from VPDB. The deviation of total $CO_2$ from being proportional to $P(626)$ due to inconsistencies of $^{13}x$, $^{17}x$, $^{18}x$ between the measurand and VPDB, using Eq. (10), is

$$\Delta P(626) \overset{\text{def}}{=} P(626) - P_{VPDB}(626) =$$

$$\frac{\partial P(626)}{\partial^{13}Rx}\left(^{13}Rx - {}^{13}Rx_{VPDB}\right) + \frac{\partial P(626)}{\partial^{17}Rx}\left(^{17}Rx - {}^{17}Rx_{VPDB}\right) + \frac{\partial P(626)}{\partial^{18}Rx}\left(^{18}Rx - {}^{18}Rx_{VPDB}\right)$$

(14)

The above are the first terms of a Taylor expansion around $P_{VPDB}(626)$. Inserting the first derivatives and using Eq. (6) gives:

$$\Delta P(626) = -\left(1 - {}^{17}Rx_{VPDB} - {}^{18}Rx_{VPDB}\right)^2\left(^{13}Rx_{VPDB}\,{}^{13}\Delta\right) +$$

(15)

$$-2\left(1 - {}^{13}Rx_{VPDB}\right)\left(1 - {}^{17}Rx_{VPDB} - {}^{18}Rx_{VPDB}\right)\left(^{17}Rx_{VPDB}\,{}^{17}\Delta + {}^{18}Rx_{VPDB}\,{}^{18}\Delta\right)$$

If $^{13}\Delta$ is positive the air to be measured has a higher $^{13}C/^{12}C$ ratio than VPDB. Therefore $P(626)$ is slightly lower than it is for VPDB, and the relative correction in the mole fraction assigned to the measured air will have to be positive, of opposite sign to the relative error of $P(626)$:

$$\frac{\Delta y_{CO2}}{y_{CO2}} = -\frac{\Delta P(626)}{P(626)} = \frac{^{13}Rx_{VPDB}\,{}^{13}\Delta}{\left(1 - {}^{13}Rx_{VPDB}\right)} + \frac{2\left(^{17}Rx_{VPDB}\,{}^{17}\Delta + {}^{18}Rx_{VPDB}\,{}^{18}\Delta\right)}{\left(1 - {}^{17}Rx_{VPDB} - {}^{18}Rx_{VPDB}\right)}$$

(16)

We note here that we could have used a $^{16}O^{13}C^{16}O$ line for quantifying $y_{CO2}$, but an analogous derivation for $\Delta P(636)/P(636)$ shows that it is 90 times more sensitive to isotopic errors or mismatches.

Using Eqs. (2) and (3) gives

$$\frac{\Delta y_{CO2}}{y_{CO2}} = {}^{13}r_{VPDB}\,{}^{13}\Delta + 2\left(^{17}r_{VPDB}\,{}^{17}\Delta + {}^{18}r_{VPDB}\,{}^{18}\Delta\right)$$

(17)

Generally, one is not making atmospheric $CO_2$ measurements with standards that have isotopic abundances for C and O exactly like VPDB. Because the linear Eq. (17) applies to the measurement of a transfer standard itself as well as to an air sample, we can give an expression for corrections to be made when the standard (subscript "st") has an isotopic composition different from air but not equal to VPDB:

$$\frac{\Delta \cancel{x}y_{CO2}}{\cancel{x}y_{CO2}} = {}^{13}\boldsymbol{r}_{VPDB}\left({}^{13}\Delta_{air} - {}^{13}\Delta_{st}\right) + 2\left({}^{17}\boldsymbol{r}_{VPDB}\left({}^{17}\Delta_{air} - {}^{17}\Delta_{st}\right) + {}^{18}\boldsymbol{r}_{VPDB}\left({}^{18}\Delta_{air} - {}^{18}\Delta_{st}\right)\right)$$
(18)

In the Appendix we derive the following very close approximation to Eq. (18) in which the $\Delta$ values have been replaced by the familiar $\delta$ values:

$$\Delta \cancel{x}y_{CO2} = \cancel{x}y_{CO2}\left[0.01111\left({}^{13}\delta_{air} - {}^{13}\delta_{st}\right) + 2*0.0023\left({}^{18}\delta_{air} - {}^{18}\delta_{st}\right)\right]$$
(19)

This is an expression for $CO_2$ corrections when only the ${}^{16}O^{12}C^{16}O$ isotopologue is used to measure $\cancel{\boldsymbol{X}_{CO2}}y_{CO2}$, and we are using VPDB scales. As an example, if we use a standard with $CO_2$ made from natural gas, it could have ${}^{13}\delta_{st} = -0.045$ and ${}^{18}\delta_{st} = -0.017$ on the VPDB scales, whereas air has ${}^{13}\delta_{air} \cong -0.008$ and ${}^{18}\delta_{air} \cong 0.000$. Assuming $\cancel{\boldsymbol{X}_{CO2}}y_{CO2} = 400$ µmol mol$^{-1}$, then $\cancel{\Delta X}\Delta y = 0.164 + 0.031 = 0.\cancel{194}195$ µmol mol$^{-1}$. ${}^{13}\delta_{air}$ is higher than ${}^{13}\delta_{st}$, so that the ${}^{16}O^{12}C^{16}O$ abundance of the standard is higher than assumed, resulting in the air measurement being too low. Therefore an upward correction is needed for ${}^{13}C$ and likewise for ${}^{18}O$.

**6 Practical calculations for definition and propagation of the $CO_2$ calibration scale**

Equation (18) and (19) give the correction required when only the ${}^{16}O^{12}C^{16}O$ isotopologue is used to determine $\cancel{\boldsymbol{X}_{CO2}}y_{CO2}$ as a function of the isotopic differences between the sample and a single standard. However, most $CO_2$ measurements are made vs a suite of standards that may have various isotopic compositions and the isotopic compositions may be a function of $CO_2$ (as is the case for the primary $CO_2$ standards used by the WMO GAW CCL). In this case the calibration curve that defines the response of the analyser may incorporate a systematic error making the idea of a simple "correction" impractical. Equation (18) and (19) can  be used to  estimate the potential offsets due to sample/standard isotopic differences but is not practical for making corrections when multiple standards are used. Therefore, we must instead use a calibration approach that fully accounts for the isotopic composition of the standards rather than using a post measurement correction.

~~A great advantage of our method is that it uses multiple standards covering a range of values to create a scale. "Scale contraction" can result from having a single standard reference, and mass spectrometer measurements have suffered from that. We do not have such "contraction" because we have a real scale over the full range of interest instead of a single point so that we can create a response curve for any analyzer. Secondly, having such isotopologue specific response curves over a large range also opens the possibility to make a new determination of the value of$\left({}^{13}C/{}^{12}C\right)_{VPDB}$. One could make for example 400 ppm and 800 ppm CO2 in air mixtures with their isotopic ratios~~

We have taken the approach of decomposing the total $CO_2$ in the primary standards, as defined by manometric measurements, into individual isotopologue mole fractions in air based on measured $\delta^{13}C$ and $\delta^{18}O$ values. The $\delta^{13}C$

5 and $\delta^{18}O$ values are determined by IRMS by the Stable Isotope Laboratory, Institute of Arctic and Alpine Research, University of Colorado, Boulder (INSTAAR) on their own realization of the VPDB scales. The current scales used by INSTAAR were set using NBS-19 and NBS-20 (carbonates) and VSMOW, GISP and SLAP (waters) (Trolier et al., 1996).  These isotopologue specific mole fractions in air of the standards are used to calibrate laser  spectroscopic instruments for the three major $CO_2$ isotopologues ($^{16}O^{12}C^{16}O$, $^{16}O^{13}C^{16}O$, and $^{18}O^{12}C^{16}O$$^{16}O^{12}C^{18}O$)

10 individually. The three major isotopologues in unknown cylinders are measured relative to these isotopologue specific calibration curves. The isotopologue mole fractions in air of the unknowns are then recombined into total $CO_2$ and conventional $\delta^{13}C$, and $\delta^{18}O$ values while properly accounting for the non-measured rare isotopologues.

A great advantage of our method is that it uses multiple standards covering a range of values to create a scale. "Scale contraction" can result from extrapolating from a single standard reference, and mass spectrometer

15 measurements have suffered from that. We do not have such "contraction" because we  calibrate with multiple standards over the full range of interest instead of using a single point  Having such isotopologue specific response curves over a large range also opens the intriguing possibilities of making $CO_2$ isotopic scales that are traceable to SI and improving our understanding of VPDB and its relation with LSVEC. This may be beyond the scope of our laboratory but we offer it as an interesting

20 aside.~~Secondly, having such isotopologue specific response curves over a large range also opens the possibility to make a new determination of the value of ($^{13}C/^{12}C$)$_{VPDB}$. One could make for example 400 ppm and 800 ppm $CO_2$ in air mixtures with their isotopic ratios close to the VPDB values. Then a small and well known amount of pure $^{13}CO_2$ could be added to the 400 ppm standard so that its $^{636}$ isotopologue mole fraction in air ends up close to, say, that of the 800 ppm mixture.added needed to double the amount of 636 that corresponds to VPDB provides a measure of

25 VPDB itself.~~

Suppose we have one or more instruments measuring each isotopologue ($^{16}O^{12}C^{16}O$, $^{16}O^{13}C^{16}O$,  $^{16}O^{12}C^{18}O$ and perhaps also $^{16}O^{12}C^{17}O$) individually. The response of the instrument(s) for each of the isotopologues needs to be calibrated separately. How often such calibrations need to be repeated depends on the

30 instrument. For this purpose we need to have a series of reference gas standards with well defined total $CO_2$ ($\boldsymbol{X_{CO2}}x_{CO_2}$) and with known conventional δ-values for the isotopic ratios. Equations (10)-(13) can be used to convert that information to the fractional abundances of the isotopologues, by first writing them in terms of conventional delta values by using relations (2) and (3) and by writing $r_{sample}$ as $r_{\rho VPDB}(1+\delta)$ (see Eq. (5)).

$$P(626) = \frac{1}{1+^{13}r_{\text{VPDB}}(1+^{13}\delta)} \cdot \frac{1}{\left[1+^{17}r_{\text{VPDB}}(1+^{17}\delta)+^{18}r_{\text{VPDB}}(1+^{18}\delta)\right]^2}$$

(20)

$$P(636) = \frac{^{13}r_{\text{VPDB}}(1+^{13}\delta)}{1+^{13}r_{\text{VPDB}}(1+^{13}\delta)} \cdot \frac{1}{\left[1+^{17}r_{\text{VPDB}}(1+^{17}\delta)+^{18}r_{\text{VPDB}}(1+^{18}\delta)\right]^2}$$

(21)

$$P(628) = \frac{1}{1+^{13}r_{\text{VPDB}}(1+^{13}\delta)} \cdot \frac{2*^{18}r_{\text{VPDB}}(1+^{18}\delta)}{\left[1+^{17}r_{\text{VPDB}}(1+^{17}\delta)+^{18}r_{\text{VPDB}}(1+^{18}\delta)\right]^2}$$

(22)

$$P(627) = \frac{1}{1+^{13}r_{\text{VPDB}}(1+^{13}\delta)} \cdot \frac{2*^{17}r_{\text{VPDB}}(1+^{17}\delta)}{\left[1+^{17}r_{\text{VPDB}}(1+^{17}\delta)+^{18}r_{\text{VPDB}}(1+^{18}\delta)\right]^2}$$

(23)

If $\delta^{17}O$ has not been measured, we approximate $\delta^{17}O = 0.528 * \delta^{18}O$ to determine the fractional abundances above.

The fractional abundances (Eqs. (20)-(23)) are converted into mole fractions in dry air by multiplying with the total mole fraction of $CO_2$ in dry air ($y_{CO2}$). The isotopologue mole fractions in air are written as $y(626)$, etc. In other words, we have $y(626) = y_{CO2} * P(626)$ and similar for all isotopologues.

A series of standards can in this way be used to calibrate the instrument response for each isotopologue individually. With these response functions we can then assign mole fractions in air to the isotopologues of the unknown gas mixtures that are being measured, $y(626)_{\text{unk}}$, etc.

Then we need to convert the measured isotopologue mole fractions of the unknown ($y(626)_{\text{unk}}$, $y(636)_{\text{unk}}$, and $y(628)_{\text{unk}}$) back to standard delta-notation using Eqs. (20)-(23) as follows:

$$\frac{y(636)_{\text{unk}}}{y(626)_{\text{unk}}} = \frac{P(636)}{P(626)} = {}^{13}r_{\text{VPDB}}(1 + {}^{13}\delta) \quad \Rightarrow \quad {}^{13}\delta = \frac{y(636)_{\text{unk}}}{{}^{13}r_{\text{VPDB}}*y(626)_{\text{unk}}} - 1$$

(24)

and $\frac{X(826)_{\text{unk}}}{X(626)_{\text{unk}}} = \frac{P(826)}{P(626)} = 2 * {}^{18}r_{\text{PDB}}(1 + {}^{18}\delta)\left[1 + {}^{17}r_{\text{PDB}}(1 + {}^{17}\delta) + {}^{18}r_{\text{PDB}}(1 + {}^{18}\delta)\right]$ (24)

$$\frac{y(826)_{\text{unk}}}{y(626)_{\text{unk}}} = \frac{P(826)}{P(626)} = 2 * {}^{18}r_{\text{VPDB}}(1 + {}^{18}\delta) \quad \Rightarrow \quad {}^{18}\delta = \frac{y(826)_{\text{unk}}}{2*^{18}r_{\text{VPDB}}*y(626)_{\text{unk}}} - 1$$

(25)

and similarly for $\delta^{17}O$. If $\delta^{17}O$ has not been measured we assume that $\delta^{17}O = 0.528 * \delta^{18}O$. Equation (24) is (weakly) non-linear. We re-arrange it as

$$\frac{X(826)_{unk}}{2 * {}^{18}r_{PDB} * X(626)_{unk}} = 1 + [\dots] + {}^{18}\delta \ (1 + [\dots])$$

in which we defined $[\dots] = {}^{17}r_{PDB}(1 + {}^{17}\delta) + {}^{18}r_{PDB}(1 + {}^{18}\delta)$

and rearrange it further into:

$${}^{18}\delta = \left[\frac{X(826)_{unk}}{2*{}^{18}r_{PDB}*X(626)_{unk}} - 1\right] - [(1 + {}^{18}\delta)({}^{17}r_{PDB}(1 + {}^{17}\delta) + {}^{18}r_{PDB}(1 + {}^{18}\delta))] \qquad (25)$$

The first approximation to ${}^{18}\delta$ is to assume $\delta^{18}O$ and $\delta^{17}O = 0$ on the right hand side, i.e. equal to the standard reference material:

$${}^{18}\delta = \left[\frac{X(826)_{unk}}{2 * {}^{18}r_{PDB} * X(626)_{unk}} - 1\right] - {}^{17}r_{PDB} - {}^{18}r_{PDB}$$

Then we substitute this first approximation into Eq. (25), with the assumption of $\delta^{17}O = 0.528 * \delta^{18}O$, and iterate the solution for ${}^{18}\delta$ by continuing to substitute it in the right hand side of Eq. (25). The approximation is extremely close to the full solution unless the sample is highly depleted in $\delta^{18}O$.

If $X(726)_{unk}$ has been measured, an equation similar to Eq. (25) applies:

$${}^{17}\delta = \left[\frac{X(726)_{unk}}{2*{}^{17}r_{PDB}*X(626)_{unk}} - 1\right] - [(1 + {}^{17}\delta)({}^{17}r_{PDB}(1 + {}^{17}\delta) + {}^{18}r_{PDB}(1 + {}^{18}\delta))] \qquad (26)$$

In this case Eqs. (25) and (26) can be iterated together substituting updated values for both ${}^{18}\delta$ and ${}^{17}\delta$.

The total $CO_2$ in dry air is given by

$$\boldsymbol{Xy}_{CO2,unk} = \frac{\boldsymbol{X}y(626)_{unk} + \boldsymbol{X}y(636)_{unk} + \boldsymbol{X}y(826628)_{unk} + \boldsymbol{X}y(726627)_{unk}}{P(626)_{unk} + P(636)_{unk} + P(826628)_{unk} + P(726627)_{unk}}$$

$(2\underline{6}\underline{7})$

Dividing by the sum of the probabilities ( P) corrects the sum of the measured isotopologues for the unmeasured rare isotopologues The sum of the probabilities in Eq (26) would  equal  0. 99993922 if the isotopic ratios are equal to the standard reference materials for carbon and oxygen. This would add 0.024 μmol mol⁻¹ to the sum of the measured isotopologues, assuming $\boldsymbol{X}_{CO2}$ $\boldsymbol{y}_{CO2}$ ~ 400 μmol mol⁻¹.  The correction in Eq. (2\underline{6}\underline{7}) that applies for actual  
[revised manuscript text omitted]

As mentioned in section 6, INSTAAR $\delta^{13}C$ and $\delta^{18}O$ measurements are relative to their own realization of the VPBD scales rather than on the WMO GAW scale for isotopic measurements of $CO_2$ (Jena Reference Air Set (JRAS-06) maintained by the Max Planck Institute for Biogeochemistry, Jena Germany) (Wendeberg et al., 2013). INSTAAR has scale contraction issues relative to JRAS. The relationships between INSTAAR and JRAS published by Wendeberg et al. (2013) indicate that while the offsets are significant for isotopic studies, the use of the INSTAAR realization for accounting for isotopic differences when determining total $CO_2$ will not add significant bias. When we use primary standards to calibrate secondary standards, the apportionment of the total $CO_2$ into component isotopologues will be slightly off. However, this is partially corrected when we recombine the resulting measured isotopologue mole fractions of the secondary standards into total $CO_2$. Using approximate JRAS values for our primary standards based on the Wendeberg et al. (2013) relationships, we see changes in the apportionment of the $^{16}O^{12}C^{16}O$ isotopologues on the order of 0.000 to 0.004 μmol mol$^{-1}$ with corresponding but opposite sign changes in the other isotopologues.

The instrument readings are absorption measurements corrected for cell pressure and temperature and converted into nominal mole fraction units. However, we treat them purely as an instrument response in arbitrary units. They could also be a voltage or a current. The responses from the analyzers are subsequently used in an offline calibration of each instrument. We do not use the internal calibration capabilities of the instruments; this ensures that the measurements are directly traceable to the WMO primary standards and can be reprocessed for future scale revisions. Each standard is measured relative to a reference cylinder to correct for slow drift of the analyzers. For the CRDS and ICOS analyzers the instrument response to each standard is divided by the average instrument response of the bracketing reference aliquots. For the QC-TILDAS, the difference between the response to the standard and the reference is used. In both cases we term the resulting values "response ratios". The choice of division vs subtraction is made due to the characteristics of the drift in each analyser. For example, the division operation does a better job when there is a slow span drift (perhaps due to variations in cell temperature and pressure) causing relative changes that are proportional to $X_{CO2}$ $CO_2$, whereas the difference operation is more appropriate when the majority of the drift is caused by a uniform shift in the output that does not depend on $X_{CO2}$ is not proportional to $CO_2$. Rather

than characterize the source of drift in each analyzer we use the reproducibility of target tank measurements to empirically determine which method gives more consistent results between calibration episodes.

The calibration curves are $CO_2$ isotopologue mole fractions as a function of response ratios. The CRDS instrument response is linear within the uncertainty of the standards (typical uncertainty of the primary standards is ±0.1 μmol mol$^{-1}$ 68 % CI). However, both isotope analyzers are slightly non-linear in their response and are fit with a quadratic polynomial. Non-linearity in the isotope analyzers may be partially due to incomplete flushing of the sample cell, caused by un-swept dead volumes, as the system switches from reference to standard. Memory of theResidual reference gas (ambient air from Niwot Ridge, ~400 μmol mol$^{-1}$ $CO_2$) in the sample cell influences the standards on the ends of the scale more than those close to the reference gas value potentially leading to a slight non-linear response. The difference in $^{16}O^{13}C^{16}O$ calibration curve residuals at 600 μmol/mol using a quadratic fit (0.0005 μmol/mol) and a linear fit (0.003 μmol/mol) indicate the memory effect is small in terms of total $CO_2$. Since all standards and all samples are treated identically and measured against the same reference gas, small memory effects should cancel out. Longer flushing times would reduce the memory effect but would decrease the lifetime of the standards.

Sample measurements are made relative to the same reference tank to account for drift in the analyzers between calibration episodes. The sample response ratios are used with the isotopologue specific calibration curves to determine isotopologue mole fractions for the sample cylinder which are combined into total $CO_2$, $\delta^{13}C$, and $\delta^{18}O$ values using the method discussed above. These values (total $CO_2$, $\delta^{13}C$, and $\delta^{18}O$) are stored in the NOAA database and are reported to the user via certificates and the web interface. Isotopologue specific mole fractions are not provided, however the equations described in this paper can be used to regenerate them.

Performance of the new calibration system has been evaluated over approximately one year by repeated measurements of target tanks (cylinders repeatedly measured as a diagnostic of system performance). Figure 2 shows the time series of total $CO_2$ measured for 4 target tanks with $CO_2$ ranging from 357 to 456 μmol mol$^{-1}$. Standard deviations of the measurements are approximately ±0.007 μmol mol$^{-1}$. Reproducibility of the target tanks close to the reference tank (typically ~ 400 μmol mol$^{-1}$ $CO_2$) are a little better than those farther out on the ends of the calibration range but the difference is small. While one year is not a long enough time series to fully quantify the reproducibility of the system, we estimate it to be ±0.01 μmol mol$^{-1}$ (68 % CI) based on these target tank measurements. This is a significant improvement over the NDIR system where reproducibility is ±0.03 μmol mol$^{-1}$ (68 % CI) ("Carbon Dioxide WMO Scale", 2017).

Prior to this new $CO_2$ calibration system, NOAA provided informational isotopic values for tertiary standards delivered to outside organizations by taking discrete samples from cylinders in flasks and having them measured by INSTAAR. This continued during the 6 months period when both calibration systems were run in parallel. Comparisons of these measurements with the isotopic results from the new calibration system are show in Figs. 3 and 4. The top plot in each figure is differences of measured delta values (NOAA – INSTAAR) vs INSTAAR values and the bottom plot in each figure is differences as a function of total $CO_2$ measured by NOAA. There is no

systematic bias between the NOAA and INSTAAR measurements for either species, except for highly depleted cylinders ($\delta^{13}$C or $\delta^{18}$O less than -20 ‰, shown by open symbols in both figures) and $\delta^{18}$O in very high $CO_2$ cylinders (> 490 µmol mol$^{-1}$). The average offset (NOAA – INSTAAR) of non-depleted tanks is   0.0 ± 0.1 ‰ $\delta^{13}$C and 0.0 ± 0.2 ‰ $\delta^{18}$O.  The offset in the highly depleted cylinders most likely occurs as a result of the large extrapolation in the INSTAAR IRMS measurements from the working standard at ambient $\delta^{13}$C and $\delta^{18}$O. These offsets are roughly consistent with the INSTAAR JRAS offsets (Wendeberg et al., 2013) which are attributed to scale contraction issues at INSTAAR. The secondary standards used to routinely calibrate the NOAA system have isotopic assignments made by direct measurement by INSTAAR and are all relatively close to ambient (see figures 5 and 6) where the INSTAAR scale contraction is very small. By using these standards and calibrating our measurements in mole fraction space we are not sensitive to the scale contraction issues in the INSTAAR measurement of depleted tanks. The $\delta^{18}$O data do show a pronounced "hook" above ~ 490 µmol mol$^{-1}$. This is thought to be due to issues when sampling air from cylinders into flasks and not to the measurements either at INSTAAR or NOAA. A tertiary standard with 497 µmol mol$^{-1}$ $CO_2$ showed excellent agreement when measured directly by both NOAA ($\delta^{18}$O = -8.92 ± 0.04 ‰) and  INSTAAR ($\delta^{18}$O = -8.94 ± 0.1 ‰). A comparison can also be made using the secondary standards which were calibrated directly by INSTAAR and by NOAA verses the primary standards. The assigned values of the secondary standards (as measured directly by INSTAAR) and the NOAA minus INSTAAR differences are shown in Figs. 5 and 6 for $\delta^{13}$C and $\delta^{18}$O respectively. Agreement is very good but there is a loss of precision on the NOAA calibration system near the wings of the $CO_2$ scale. NOAA measurements show some decrease in performance as total $CO_2$ moves away from the reference cylinder, which is always an ambient $CO_2$ cylinder. However, even on the wings of the range the performance is more than adequate for the purpose of correcting total $CO_2$ for isotopic differences. The reproducibility of $\delta^{13}$C (±-0.2 ‰, 68 % CI) and $\delta^{18}$O (±-0.2 ‰, 68 % CI) measurements are again estimated from target tanks measurements. The uncertainty of the $\delta^{13}$C and $\delta^{18}$O measurements is dependent on the uncertainty of the total $CO_2$ values of the standards in addition to the reproducibility of the measurement system (Flores et al., 2017). This will be treated in an upcoming publication describing the $CO_2$ scale revision (Hall et al., in preparation). The uncertainties of our measurement results for $\delta^{13}$C and $\delta^{18}$O are more than adequate for correcting atmospheric $CO_2$ measurements for standard vs sample isotopic differences. However, we caution against using them as standards for high precision $CO_2$ isotopic measurements.

[revised manuscript text omitted]

**Appendix**

We will derive expressions for $\Delta$ in terms of conventional $\delta$ values because we currently supply standards to users within the greenhouse gas measurement community with their $\delta$ values as information in addition to the total $x_{CO2}$ $x_{CO2}$ calibration.

$$^{13}\Delta = \frac{^{13}Rx}{^{13}Rx_{\text{VPDB}}} - 1 = \frac{^{13}r}{^{13}r_{\text{VPDB}}}\frac{1 + {}^{13}r_{\text{VPDB}}}{1 + {}^{13}r} - 1 = (1 + {}^{13}\delta)(1 + {}^{13}r_{\text{VPDB}})(1 - {}^{13}r + {}^{13}r^2) - 1$$

Where we have used the first 3 terms of the series expansion $(1+r)^{-1} = 1 - r + r^2 - r^3 + \dots$ and the definitions of $r$, $Rx$, $\delta$, and $\Delta$. Expanding,

$$^{13}\Delta = (1 - {}^{13}r + {}^{13}r^2) + {}^{13}r_{\text{VPDB}}(1 - {}^{13}r + {}^{13}r^2) +$$
$$^{13}\delta(1 - {}^{13}r + {}^{13}r^2) + {}^{13}\delta{}^{13}r_{\text{VPDB}}(1 - {}^{13}r + {}^{13}r^2) - 1$$

and rearranging, we get

$$^{13}\Delta = (-{}^{13}r + {}^{13}r^2) + ({}^{13}r_{\text{VPDB}} + {}^{13}\delta)(1 - {}^{13}r + {}^{13}r^2) + {}^{13}\delta{}^{13}r_{\text{VPDB}}(1 - {}^{13}r + {}^{13}r^2)$$

Rearranging further,

$$^{13}\Delta = {}^{13}\delta - ({}^{13}r - {}^{13}r_{\text{VPDB}}) + {}^{13}r({}^{13}r - {}^{13}r_{\text{VPDB}}) - {}^{13}\delta({}^{13}r - {}^{13}r_{\text{VPDB}}) - {}^{13}\delta{}^{13}r_{\text{VPDB}}{}^{13}r$$

Then, using Eq. (5),

$$^{13}\Delta = {}^{13}\delta - {}^{13}r_{\text{VPDB}}{}^{13}\delta + {}^{13}r{}^{13}r_{\text{VPDB}}{}^{13}\delta - {}^{13}\delta{}^{13}r_{\text{VPDB}}{}^{13}\delta - {}^{13}\delta{}^{13}r_{\text{VPDB}}{}^{13}r$$

The third and the last term cancel, and then keeping only the two leading terms, we obtain

$$^{13}\Delta = {}^{13}\delta(1 - {}^{13}r_{\text{VPDB}}) \tag{A1}$$

Equation (A1) is an excellent approximation. Using the values for $^{13}r_{\text{VPDB}}$ in Table 1 and assuming that $^{13}\delta = -0.00800$ ($-8.00$ permil, an approximate value for $CO_2$-in-air) we calculate both $^{13}R$ $^{13}x$ for the air sample and $^{13}R_{\text{PDB}}$ $^{13}x_{\text{VPDB}}$, and using the definition (Eq. (6)) for $^{13}\Delta$, we obtain $^{13}\Delta = -0.0079120$ $2$. Equation (A1) gives us $-0.0079106$ $1$.

A very similar derivation holds for $^{17}\Delta$ and $^{18}\Delta$ but it is a bit more complicated because the terms for $^{17}R$ $^{17}x$ and $^{18}R$ $^{18}x$ get mixed.

$$^{17}\Delta = \frac{^{17}r}{^{17}r_{\text{VPDB}}}\frac{1 + {}^{17}r + {}^{18}r}{1 + {}^{17}r_{\text{VPDB}} + {}^{18}r_{\text{VPDB}}} - 1 = \frac{^{17}r}{^{17}r_{\text{VPDB}}}\frac{1 + {}^{78}r}{1 + {}^{78}r_{\text{VPDB}}} - 1$$

To keep the notation simpler and stressing the analogy with the derivation for $^{13}\Delta$ we have written in the above $^{78}r = {}^{17}r + {}^{18}r$ for the air sample and $^{78}r_{\text{PVPDB}} = {}^{17}r_{\text{PVPDB}} + {}^{18}r_{\text{PVPDB}}$ for the standard.

After keeping only the leading terms we have

$$^{17}\Delta = {}^{17}\delta - ({}^{78}r - {}^{78}r_{\text{VPDB}}) = {}^{17}\delta - {}^{17}r_{\text{VPDB}}{}^{17}\delta - {}^{18}r_{\text{VPDB}}{}^{18}\delta$$
$$\tag{A2}$$

And similarly for $^{18}O$:

$$^{18}\Delta = {}^{18}\delta - {}^{17}r_{\text{VPDB}}{}^{17}\delta - {}^{18}r_{\text{VPDB}}{}^{18}\delta \qquad\qquad —(A3)$$

These are the equivalents of Eq. (A1) for $^{13}\Delta$. Because $^{17}r$ and $^{18}r$ are significantly smaller than $^{13}r$, we approximate further $^{17}\Delta \cong {}^{17}\delta$ and $^{18}\Delta \cong {}^{18}\delta$. Since $^{17}\delta$ is not usually measured and also is often very closely related as $^{17}\delta = 0.53$

5   $^{18}\delta$, we can write for the oxygen correction terms in Eq. (16) $\frac{{}^{17}r_{\text{PDB}}}{{}^{17}r_{\text{VPDB}}}{}^{17}\delta + \frac{{}^{18}r_{\text{PDB}}}{{}^{18}r_{\text{VPDB}}}{}^{18}\delta = (0.53$ $\frac{{}^{17}r_{\text{PDB}}}{{}^{17}r_{\text{VPDB}}} + \frac{{}^{18}r_{\text{PDB}}}{{}^{18}r_{\text{VPDB}}}) {}^{18}\delta$, and filling in the $r_{\text{PVPDB}}$ values from Table 1,

$0.53*395*10^{-6}*{}^{18}\delta + 2088*10^{-6}*{}^{18}\delta = 2297*10^{-6}*{}^{18}\delta \cong 0.0023*{}^{18}\delta$

Now we return to Eq. (18) in the main text, applicable when the isotopic composition of measured air is different from the standard that is used. We restate it as

10   $\Delta \cancel{X} y_{\text{totCO2}} = \cancel{X} y_{\text{totCO2}} \left[ 0.0110\underline{6}\cancel{11}\left({}^{13}\delta_{\text{air}} - {}^{13}\delta_{\text{st}}\right) + 2*0.0023\left({}^{18}\delta_{\text{air}} - {}^{18}\delta_{\text{st}}\right) \right]$
(A4)

**Supplemental links**

None

**Author contribution**

P. Tans derived the equations used to apportion total $CO_2$ into component isotopologues and their application. A. Crotwell designed and built the calibration system. K. Thoning wrote software for operating the calibration system and managing the data.

**Competing interests**

The authors declare that they have no conflict of interest.

**Disclaimer**

None

**Acknowledgements**

We thank Thomas Mefford for running the calibration system, Duane Kitzis for standards preparation, and Ed Dlugokencky and three anonymous reviewers for reviewing this manuscript. In addition, we thank Sylvia Michel, Natalie Cristo, and the other members of the INSTAAR Stable Isotope Laboratory for isotopic measurements.

**Tables**

Table 1: Isotopic ratios of international standard reference materials.

| Reference Material | Ratio | Reference |
|---|---|---|
| $(^{18}O/^{16}O)_{VSMOW}$ | 0.0020052 | (Baertschi, 1976) |
| $(^{17}O/^{16}O)_{VSMOW}$[a] | 0.00038672 | (Assonov and Brenninkmeijer, 2003) |
| $(^{18}O/^{16}O)_{VPDB-CO2}$ | 0.0020883 | (Allison et al., 1995) |
| $(^{17}O/^{16}O)_{VPDB-CO2}$[b] | 0.00039511 | (Assonov and Brenninkmeijer, 2003) |
| $(^{13}C/^{12}C)_{VPDB}$[c] | 0.011180 | (Zhang, 1990) |
| $(^{2}H/^{1}H)_{VSMOW}$ | 0.00015576 | (Hagemann et al., 1970) |
| $(^{15}N/^{14}N)_{air-N2}$ | 0.0036765 | (Junk and Svec, 1958; Coplen, et al., 1992) |

[a] In other literature, it is possible to find different $^{17}O/^{16}O$ ratio values for these standard reference materials than those given here. However, for the determination of $^{17}O$ isotopic effects on atmospheric $CO_2$ measurements, differences from the values given in this table are insignificant.

[b] This $^{17}O/^{16}O$ ratio of VPDB-CO2 is consistent with $[(^{17}O/^{16}O)_{VPDB-CO2}/((^{17}O/^{16}O)_{VSMOW}] =$  $[(^{18}O/^{16}O)_{VPDB-CO2} /(^{18}O/^{16}O)_{VSMOW}]^{0.528}$

[c] We used the revised value of  Zhang et al. (1990).  For the determination of $^{13}C$ isotopic effects on atmospheric $CO_2$ measurements the difference between this value and the original value 0.0112372 (Craig, 1957) is insignificant.

**Figure captions**

Figure 1: Schematic for the NOAA laser spectroscopic $CO_2$ calibration system. The CRDS analyzer is used with one of the $CO_2$ isotope analyzers which are interchangeable.

Figure 2: Total $CO_2$ calibration results for four target tanks measured on the laser spectroscopic $CO_2$ calibration system over approximately 1 year. Error bars are the standard deviation of 8 measurements per calibration episode. The results span multiple gas handling system modifications. Values since April 2016 are on a consistent design. Average and standard deviations of the four target tanks results are: A) CC71624 356.628 ± 0.007 μmol mol$^{-1}$, B) CB11127 392.985 ± 0.006 μmol mol$^{-1}$, C) CA05008 406.652 ±0.006 μmol mol$^{-1}$, and D) CB10826 455.734 ± 0.008 μmol mol$^{-1}$.

Figure 3: Discrete samples from tertiary standards were collected in flasks and measured by INSTAAR. The average INSTAAR flask $\delta^{13}C$ result is compared to the average $\delta^{13}C$ tank calibration result on NOAA's laser spectroscopic $CO_2$ calibration system. Top panel is the difference (NOAA – INSTAAR) as a function of the INSTAAR $\delta^{13}C$ value and the bottom panel is the difference vs total $CO_2$. Error bars in both plots are the standard deviation of multiple calibration episodes by NOAA.  INSTAAR uncertainties are typically ±0.03 ‰ (68 % CI) (Trolier et al., 1996) but do not account for problems with the collection of the discrete air sample. Highly depleted cylinders ($\delta^{13}C$ < -20 ‰) are shown with open circles in each panel.

Figure 4: The same as Figure 3 for $\delta^{18}O$.  INSTAAR uncertainties are typically ±0.05 ‰ (68 % CI) (Trolier et al., 1996) but again do not account for problems with the collection of the discrete air samples. Differences greater than 1.5 ‰ are assumed to be caused by problems during discrete sample collection. These results are shown but are not included in the statistics. Highly depleted cylinders ($\delta^{18}O$ < -20 ‰) are shown with open circles in each panel.

Figure 5: Secondary standards used to calibrate the laser spectroscopic system have $\delta^{13}C$ and $\delta^{18}O$ values from direct measurement by INSTAAR and they have measured $\delta^{13}C$ and $\delta^{18}O$ from calibration on the laser spectroscopic system against the primary $CO_2$ standards. The top panel shows the INSTAAR $\delta^{13}C$ values as a function of $CO_2$. Uncertainties on the INSTAAR values (less than 0.02 ‰) are not visible. The bottom panel shows the difference between the NOAA and INSTAAR measurements of the secondary standards (NOAA – INSTAAR) also as a function of $CO_2$. Error bars are the standard deviation of  four calibration episodes of the secondary standards vs the primary standards on the NOAA $CO_2$ calibration system.

Figure 6: The same as Figure 5 for $\delta^{18}O$. The  $CO_2$ dependent depletion of $\delta^{18}O$ in cylinders above ambient $CO_2$ results from the depleted $\delta^{18}O$ of the spike gas.

Figure 7: The laser spectroscopic $CO_2$ calibration system was run in parallel with the NDIR $CO_2$ calibration system for approximately 6 months. The differences (average NDIR – average laser spectroscopic system) are plotted as a function of $CO_2$. Typical reproducibility of the NDIR measurements (±0.03, 68 % CI) are shown with dashed lines. Highly depleted cylinders ($\delta^{13}C$ < -20 ‰) are shown by open circles. These clearly indicate enhanced offsets due to the NDIR being somewhat sensitive to the isotopic composition differences between the samples and the standards used to calibrate the instrument.

---

## Author Response (AR1)

Tans-AMT-2017-34 Response to reviewers.

Manuscript with track changes below response to reviewers

Comments from referee 1

The manuscript of Pieter P. Tans et al. with the title "Abundances of isotopologues and calibration of CO2 greenhouse gas measurements" is reporting the development of a method to calculate amount of substance fractions for individual CO2 isotopologues, based on XCO2, ☐13C and ☐18O values. The new method is applied in combination with a new CO2 calibration system, consisting of three laser spectrometer with different technology and sensitivity for isotopologues: CRDS (16O12C16O), OA-ICOS and QC-TILDAS (16O12C16O, 16O13C16O, 18O12C16O, 17O12C16O), to account for isotopic differences among standards. The topic is very timely and of very high interest for a large number of readers of Atmospheric Measurement Techniques involved in atmospheric monitoring of greenhouse gases (especially CO2) and their isotopic composition. The manuscript is a fundamental conceptual and technical description on the WMO CO2 calibration scale and therefore a basic document to define the state of the art. As mentioned by the authors, the developed technique can be applied to other molecules, where isotopologues-specific values are desired or isotopic differences affect analytical techniques. Furthermore, the technique to calculate fractional distribution of isotopologues of CO2 (and other target substances, e.g. CH4, N2O) will be of great benefit for users of optical isotope laser spectroscopy (e.g. Edgar Flores et al., Analytical Chemistry (2017), DOI: 10.1021/acs.analchem.6b05063). The manuscript is very carefully written, concise and clear, and it can be published with very minor revisions. The authors might consider a small number of suggestions to further increase the readability / impact of their work.

Comment:    Page 12 Line 5: Eq. 21 + 22: The two formulas for P(826) and P(726) are incorrect as the second denominator is supposed to be squared. Although the mathematical equations are clearly presented, it would be very helpful for the reader to have a sample calculation, for example as a supplementary file, on how to calculate mole fractions of isotopologues (X(626) etc.) from $\delta^{13}C$ , $\delta^{18}O$  and XCO2. Implementing the equations of the presented manuscript (Eq. 12 – 15 or 19 – 22) on an example from Flores et al. (Analytical Chemistry (2017), DOI:10.1021/acs.analchem.6b05063, e.g. Table 1 mixture 1) results in different mole fractions of isotopologues than given by Flores – please state on the differences in the calculations, and cite the work of Flores at al. in your manuscript.

Response:    We agree with the comment and thank the reviewer for finding the error in the equations. The manuscript has been updated to reflect the correct equations for P(826) and P(726) as well as subsequent use of these terms. The corrected equations agree with those of Flores et al. The use of the correct equations does not significantly change the results for total $CO_2$, $\delta^{13}C$, or $\delta^{18}O$ because the same mistake was made when deconstructing the total $CO_2$ of the standards into component isotopologue mole fractions and when reconstructing total $CO_2$ from the measured isotopologue mole fractions of the samples. To a large part cancelling out. The

differences in total $CO_2$ mole fractions were less than 0.001 ppm.  Calculated $\delta^{18}O$ values did change by a few hundredths of one per mil but this is less than our analytical uncertainty.

Comment:        Page 2 Line 17-20: The role of the world calibration centres (WCCs) to independently verify the implementation of the calibration scales at laboratories of monitoring stations could be mentioned here or elsewhere in the text.

Response:       Text has been added to mention the role of the WCC's in helping institutions remain closely tied to the WMO scales.  Also mentioned is the use of "round robin" experiments for this.

Comment:        Page 3 Line 11:  Results of most recent key comparisons (CCQM) with national metrological institutes (NMIs) could be included here.
Response:       The WMO $CO_2$ scale is being revised to account for biases in the calculations of $CO_2$ mole fraction measured by the manometer and incorporate the increased knowledge of the primary standard values gained through additional manometer measurements since 2007. These biases and the scale revision are being described in an additional paper that is currently in preparation. We allude to that on page 17 line 35 in the current manuscript. The authors would prefer to discuss the results of the comparisons with independent scales in this upcoming paper that is dedicated to the revision of the scale since any bias in the manometer measurements and calculations have a direct impact on these comparisons. We hope to have this publication submitted in summer 2017.

Comment:        Page 4 Line 20 (Eq. 1): Multiplication with " x 1000" is frequently used, but should be avoided according to Tyler Coplen, RCM (2011), DOI: 10.1002/rcm.5129.

Response:       The equation and text has been corrected. We don't like the x1000 convention, but had included it because many people use it.

Comment:        Page 5 Line 32: Scott Marrin Inc. offers "ultrapure air" and "cryogenic ultrapure" air but not "ultra high purity air" please specify accordingly.

Response:       The manuscript has been changed to reflect that NOAA uses "ultrapure air" from Scott Marrin.

Comment:        Page 6 Line 11: The sub-sentence "… to properly address isotopic issues when…" is colloquial and might be rephrased.

Response:       The sentence has been rephrased.

Comment:     Page 9 Line 24: Is error the correct wording in this context, or should it be bias? Please check.

Response:     Bias is the correct term.  The text has been changed.

Comment:     Page 11 Line 6: Replace 0.194 with 0.195.

Response:     The typo has been fixed.

Comment:     Page 14 Line 1-2: Please specify the wavelength region used for the analysis of $CO_2$ isotopologues as different possibilities exist and this might vary from instrument to instrument. Was there any additional temperature stabilization implemented for the optics/electronics of the QC-TILDAS or is the laboratory air-conditioned?

Response:     Text has been changed to indicate both analyzers use lines at approximately 2309 $cm^{-1}$. Both $CO_2$ isotope analyzers are temperature stabilized by the manufacturer, we did not add any additional control since our laboratories are well controlled.

Comment:     Page 14 Line 11: What is the reason that $\delta^{17}O$-$CO_2$ cannot be calibrated independently, no IRMS measurements?

Response:     The INSTAAR stable isotope laboratory does not have the ability to measure $\delta^{17}O$ so our standards have not been calibrated for it. We've added text to make this clear. We therefore have to assume $\delta^{17}O$ follows the mass dependent relationship to $\delta^{18}O$. Deviations from this relationship would be small and insignificant when used to assign a total $CO_2$ value to a sample.

Comment:     Page 14 Line 35 – Page 15 Line 1: The sentence "The solenoid valve fails to the idle gas …" is unclear and might be rephrased.

Response:     This sentence is unnecessary so has been removed to prevent confusion.

Comment:     Page 16 Line 13-20: The memory effect might cancel out, but it will add to the uncertainty of the isotope analyzers. Has this been quantified? Are improvements possible, such as longer flushing times, or optimization of the flow scheme?

Response:     We agree, the memory effect will increase the uncertainty near the ends of the scale.  However, the identical treatment of standards and samples should minimize the effect. The residuals of the calibration curve for the 636 isotopologue using a quadratic fit are typically ±0.0005 μmol/mol.  With a linear fit the largest residuals are 0.003 μmol/mol at the ends of the

scale. The difference in these residuals indicates the approximate magnitude of the memory effect. In terms of total $CO_2$, assuming the identical treatment corrects for some of the effect, the memory effect is not significant. It would however be important for high precision isotopic measurements. The problem could be resolved by using longer flushing times. However, we feel the extra gas usage, resulting in shorter lifetimes for the standards, is not warranted. The text has been changed to make this clearer.

Comment: Page 18 Line 24: The sentence "Isotopic standards should be calibrated by IRMS measurements" could be valid for the given example but is not a correct general statement, please specify the sentence.

Response: Agreed. We want to make the point that the isotopic measurement results provided by the CCL are not intended to propagate the VPDB scales for scientific studies that require high precision isotopic values. We want to encourage users to continue to have isotopic standards calibrated by techniques and facilities with higher precision than what our system can deliver. Our focus is on making total $CO_2$ measurements that are not influenced by isotopic differences rather than on the isotopic values themselves. The sentence has been reworded to clarify.

Comments from referee 2

Recommendation: Publish - minor revisions

General comments:

The manuscript "Abundances of isotopologues and calibration of CO2 greenhouse gas measurements" is well written and reports on a method and a new calibration system to account for differences in isotopic composition between primary CO2 reference standards. This is an important development and essential for addressing biases introduced from measurements sensitive to specific isotopologues. The authors point out that these developments can be applied to other molecules. Application to $CH_4$ and N2O would be of further benefit to users of optical spectroscopy. This work is a valuable contribution and of significant interest to the atmospheric monitoring community. The document defines the state of the art for the CO2 calibration scale and it is important that this information is in the public domain. I recommend publication subject to the following minor suggestions for revision:

Comment: The section on calibration and system performance refers to measurements of delta13C and delta18O made at INSTAAR on the primary and secondary standards using IRMS. What was the reference used for these measurements and are these traceable to VPDB?

Response: The measurements by INSTAAR are traceable to VPDB. The INSTAAR scales were set using NBS-19 and NBS-20 carbonates and VSMOW, GISP and SLAP waters. Text has been added to clarify. Also, text has been added addressing the effects of the differences between INSTAAR and JRAS.

Comment:    The term "mole fraction" and "amount of substance fraction" are used interchangeably throughout. One of these terms should be used for consistency. The second sentence of the abstract mol/mol is the unit and should replace mole fraction. This also applies to the fourth paragraph of the introduction.

Response:    We have changed the text to be more consistent with terms.  It was our understanding from Schwartz and Warneck (1995) that "amount of substance fraction" was the quantity and "mole fraction" was the unit. It seems this reading was wrong and the IUPAC compendium of chemical terminology (the Gold Book) lists "mole fraction" as a synonym of "amount of substance fraction".  We use amount of substance fraction in the first instance with mole fraction in parenthesis and then mole fraction for the rest of the document.

Schwartz, S. E., Warneck, P., Units for use in atmospheric chemistry (IUPAC recommendations 1995). Pure and Appl. Chem., 67, 1377-1406, 1995.

IUPAC. Compendium of Chemical Terminology, 2nd ed. (the "Gold Book"). Compiled by A. D. McNaught and A. Wilkinson. Blackwell Scientific Publications, Oxford (1997). XML on-line corrected version: http://goldbook.iupac.org (2006) created by M. Nic, J. Jirat, B. Kosata; updates compiled by A. Jenkins.

Comment:    I would suggest keeping all y-axis values on the left hand side of the figure and increasing the size of the interval (perhaps 3 y values per chart). It is not clear whether the bars on the data represent standard deviations or uncertainties. Is there any contribution from the change in composition in the $CO_2$ reference standards to the trend observed in figure 2? Is it assumed that the changes in $CO_2$ reference standards is negligible compared to the long term reproducibility of the facility?

Response:    We find the y-axis labels easier to read when alternating sides so leave them this way. The error bars are the standard deviation of the 8 measurements per calibration episode. Text has been added to clarify.
I don't think the apparent trend is significant. Subsequent measurements have shown step changes possibly due to regulators or subtle variations in the stability of the response of the instruments but not a significant trend in the target tanks. We updated the plot with more data collected since initial submission to make it clearer. We also include the standard deviation of the replicate calibration episodes for each tank individually in the caption.
The reference tank is not used as a point in the calibration curve.  It only corrects for slow drift between calibration episodes.  Changing the reference tank does not affect the measurement results. The reference tank is in effect being calibrated while the calibration curve is being determined.

Comment:    Figures 3 and 4 present the comparability of measurements at INSTAAR and the new calibration system. The offset of non-depleted tanks is attributed to the extrapolation of the calibration at INSTAAR. Is there any data to support this statement?

Response:     The offsets of depleted tanks is roughly consistent with the INSTAAR offset from JRAS described by Wendeberg et al. (2013) and is contributed to scale contraction at INSTAAR. Text has been added to clarify.

5   Comment:     Assuming uncertainties are symmetrical, the values presented throughout the manuscript (e.g. ± 0.007 μmol/mol in the caption to figure 2) do not require the ± sign.

Response:     We leave the ± symbol in to prevent confusion and re-iterate that they are symmetrical ranges.

Comment:     The caption to figures 5 and 7 are missing the term mole fraction (e.g. "The top panel shows the INSTAAR delta13C values as a function of CO2 mole fraction.")

15   Response:     We leave the text as is to be succinct.

Comments from referee 3

20   The paper should be published with revisions. Laser based instruments measuring concentrations of gases in the atmosphere have developed rapidly in recent years, with the ability to measure individual isotopologues. At the same time the standards used for calibrating such instruments need to be adapted accordingly to allow for correct calibration, and the paper describes the progresses made. The paper should be improved by: a) improving terminology on quantities and

25   units b) using conventionally used symbols for quantities in a number of equations c) using internationally accepted conventional values for isotope reference materials d) describing the impact of nonequilibrated CO2 in standards and the potential biases that may arise in isotope ratio measurements as a result e) reduction in the number of equations, with references to already published work f) full description of the traceability and uncertainty of isotope ratio

30   measurements by both IRMS and Optically based techniques. These should also be propagated through to measurements of mole CO2 mole fractions.

Comment:     Page 1 line 14: 'units of mole fraction' is not a correct expression; mole fraction

35   is a quantity not a unit. Correct to 'calculate the mole fraction of each component', expressed in units of μmol/mol

Response:     See response to reviewer 2 on this issue.

Comment:     Page 1 line 19: same issue as above with the use of 'mole fraction units'. Please correct.

Response:     See response to reviewer 2 on this issue.

Comment:        Page 2 line 34: correct 'units of mole fraction' also the symbol for mole fraction should be in italics, with normally lower case being used

Response:        See response to reviewer 2 on this issue.  Also, all symbols have been put into italics.  See comments below regarding use of symbols consistent with Coplen (2011) and the IUPAC gold book.

Comment:        Page 3 line 9: what is calculated is the mole fraction of CO2 in air, not the ratio. Also 'ratio of moles' is not a correct term. Please correct.

Response:        Mole fraction is the ratio of moles of $CO_2$ to moles of air.  We leave the text as is to clearly state that we calculate ratios not absolute moles on the manometer.  The absolute volumes in the manometer are not nearly as well known as the volume ratios. In all calculations we use the volume ratio rather than the actual volumes.

Comment:        Page 4 line 10: 'the number of molecules of CO2 per mole of dry air', is not correct – it is a different quantity (which would be expressed in units of 1/mol) from mole fraction (expressed as mol/mol). Restructure the sentence avoiding this part of the phrase.

Response:        This typo has been fixed, should have been "number of moles of $CO_2$ per mole of dry air".

Comment:        Page 4 line 13: the authors should reference the fact that they are using the shorthand of the spectroscopy community (e.g. reference to HITRAN, see https://www.cfa.harvard.edu/hitran/molecules.html). Also in this notation the convention is to write isotopologues as 628 and not 826. Please correct.

Response:        Reference to HITRAN added and the appropriate corrections to the shorthand notation have been made.

Comment:        Page 4 line 21: Equation 1 includes the factor 1000. This is not correct, delete the factor 1000. If needed add a phrase that delta values are often expressed in per mil, where the symbol ‰ means 0.001

Response:        The equation and text has been corrected.

Comment:        Page 5 line 16 and subsequently: when quoting ranges these need to be written as -7.0 ‰ to -9.0 ‰  Also there needed to be a space between the number and ‰ i.e -9.0 ‰ and not -9.0‰ Please correct

Response:        Spaces have been added between values and "‰" symbols.  Previous published manuscripts in AMT have expressed isotopic ranges as -7.0 to -9.0 ‰ so we leave these as is to

be consistent with other AMT manuscripts unless the editors have a preference. See Griffith et al. 2012 as example.

5    Comment:    Page 6 line 4: depleted in 13C and not _13C. Please correct

Response:    Text has been corrected.

10   Comment:    Page 6 Section 3: The authors should use conventional notation in this section, rather than introducing their own. In addition they should differentiate between quantities that are simple ratios and the ones that are fractions. See Santrock (1985) which is referenced in the paper, where the ratio of amounts of substance (abundance as used by authors) of two isotopes is demoted with the symbol R, whilst a fraction has been given the symbol F. In all cases symbols
15   should be in italics following standard practice.

Response:    When we developed our methods we were not aware of the Santrock paper. While responding to the reviewer's comments we consulted IUPAC guidelines for notation (Coplen reference), but the Santrock paper does not follow them. The IUPAC recommendations have "$R$"
20   for the isotope-number ratio and "$r$" for the isotope-amount ratio relative to the most abundant isotope, even though numerically $R=r$. Similarly, they have "$X$" and "$x$" (again, $X=x$ numerically) for the isotope-number fraction and isotope-amount fraction. Because we prefer to think in terms of amounts with the unit of moles we changed our notation to $\boldsymbol{r}$ and $\boldsymbol{x}$ for the isotopic ratio and the isotopic amount fraction respectively, and we use bold-face type for easier
25   readability. IUPAC recommends the same "$\boldsymbol{x}$" symbol for general amount of substance fractions, and "$\boldsymbol{y}$" specifically for gases. We will call these "mole fractions". We will use the "$\boldsymbol{y}$" in this paper for the $CO_2$ mole fraction in air because we like to distinguish total $CO_2$ from isotopologue-amount fractions.

     Comment:    Page 6 line 16: the equations should be numbered. The conventional symbol for an isotope ratio is R and not r.

Response:    "r" is correct. See above.

     Comment:    Page 6 line 19: These are just fractions not 'redefined ratios'. The equations should be numbered

40   Response:    We have changed the text to read "we use" rather than "we re-define".

     Comment:    Page 7 line 1: Often in papers VPDB-CO2 is shortened to VPDB, when a statement is included explaining this. PDB is not used as a shorthand for VPDB, because it
45   actually denotes the original PDB scale. Use VPDB if a short had notation is required.

Response: Shorthand "PDB" has been changed to "VPDB".

Comment: Page 7 line 3: (equation 4a) The value used for 13RVPDB is not the one recommended by the IAEA nor the WMO CCL for CO2 isotope ratios. A value of 0.01118 should be used see you reference Brand et al (2010). Similarly the values for 17R and 18R are not the same as for the Brand et al (2010) reference. Internationally accepted conventional values should be used- please correct.

Response: We now use $(^{13}C/^{12}C)_{VPDB} = 0.011180$ but we did not use $(^{17}O/^{16}O)_{VPDB-CO2} = 0.0003931$ from Brand et al. because it is inconsistent with $[(^{17}O/^{16}O)_{VPDB-CO2}/((^{17}O/^{16}O)_{VSMOW}] = [(^{18}O/^{16}O)_{VPDB-CO2}/(^{18}O/^{16}O)_{VSMOW}]^{0.528}$

Comment: Page 7 line 13-15: This sentence is not necessary if equation 1 is corrected.

Response: We leave this sentence in because many people assume all "δ" values have been multiplied by 1000 to read as permil. We want to be explicit that the δ values in the equations are in small numbers.

Comment: Page 7 line 22: 'approximate the abundance as mole fraction' should be corrected to 'calculate the mole fraction'.

Response: We prefer 'approximate' to re-iterate that the equations are not exact.

Comment: Page 7 lines 19 onwards: The ratios measured in IRMS together with the convention already mentioned on Page 4 line 26, can be solved exactly to then calculate atomic isotopic abundances, and with simple probability theory (see Ref 1 in Santrock (1985)) and knowledge of the total $CO_2$ mole fraction calculate the mole fraction of any of the 12 CO2 isotopologues in the gas.
This section would be improved by replacing with reference to the Santrock(1985) paper and reference there in.

Response: We do not follow the reviewer's recommendation because our paper would be harder to read if one has to go to another paper to follow ours, and furthermore because Santrock is focused on a different issue, namely $^{17}O$ corrections for mass spectrometers.

Comment: Page 8/9 entire section: Whilst providing a nice description of probability theory, how is this any different from the Santrock paper in describing the distribution of isotopes among molecules at equilibrium is accurately described by a simple probability function, and reference 1 therein? The current text could simply be replaced by a reference. However, what the authors have not discussed and does not seem to be treated in this paper is that these equations are only exact when the gas is in equilibrium. The procedures used for making the WMO standards, especially historically, are likely to lead to a non-equilibrated gas i.e. by mixing two

CO2 gases together with different isotopic compositions the resulting mixture does not have the distribution of isotopologues that would be predicted from the average atomic isotopic abundances of the mixture. The effect of this both for the spectroscopic and mass spectrometric methods applied in the paper should be evaluated and commented upon in order to confirm the authors' conclusions.

Response:    Our description on p. 8-9 is more general than Santrock's. If the reviewer refers to thermodynamic equilibrium, it plays no role in our treatment. CO2 as well as its  isotopic ratios are not in equilibrium in the atmosphere because the atmosphere is imperfectly mixed and sources/sinks of CO2 occur in different places and times and come with different isotopic ratios, including pure isotopic exchange not associated that any net source/sink of total CO2 (Tans et al, 1993). Even inside reference gas standards there is no full equilibrium. So what? Approximate thermodynamic calculations have been carried out to estimate fractionation factors between different species and phases (liquid, gas, solid) when isotopes are exchanged. Those fractionation factors could be kinetic or equilibrium factors. We take those fractionation factors as given, derived in most cases empirically from measurements. We measure what **is** in the atmosphere relative to what **is** in the standards, and differences of isotopic ratios are expressed in the standard delta notation, with uncertainties. That is as exact as it gets.

Comment:    Page 11 lines 21-23: The scale on which INSTAAR is measuring CO2 isotope ratios should be described, as well as the conventional values used for its scale. Is it its own realization of the VPDB or VPDB-LSVEC scale? The measurement uncertainty of this realization should be described as well as any known bias form the WMO Scale for CO2 in air (JRAS).

Response:    A great advantage of our method is that it uses multiple standards covering a range of values to create a scale. "Scale contraction" can result from having a single standard reference, and mass spectrometer measurements have suffered from that. We do not have such "contraction" because we have a real scale over a range of interest instead of a single point so that we can create for any analyzer a response curve. We have added this to section 6.
The INSTAAR offset from JRAS has been included in the discussion.  The offsets are attributed to scale contraction at INSTAAR (Wendeberg et al. 2013 and personal communication with Slyvia Michel) The differences are not significant for the total $CO_2$ calculations and our calibration strategy is largely immune to scale contraction at INSTAAR since all of our secondary standards are close to ambient isotopic composition where the effect is minimal.

Comment:    Page 11 Entire Section: Several papers have been published describing approaches for calibrating optical system for isotope ratio measurements (Wen et al, Atmos. Meas. Tech. 2013 and Flores et al. Anal. Chem 2017) with the latter including uncertainty estimation of calibration procedures. The authors reference neither, nor do they provide a description of the uncertainty of their calibration or measurements procedures. A reference to previous descriptions of calibration procedures and an assessment of the measurement uncertainty should be added, which would then allow propagation of the uncertainty into mole fraction values.

Response:      We only became aware of Flores when the first review of our paper came in. We
were not aware of the Wen et al paper.

Comment:      Page 12 and 13: The equations on these two pages are difficult to follow. It is not
clear to the reviewer why the sum of all isotopologues is not included in the reported total CO2
mole fraction value. Accurate measurement of the 626 isotopologue, together with its isotope
ratios and the assumed distribution of isotopes would allow the mole fractions of all other
10   isotopologues to be calculated and their sum added to the 626 mole fraction to give total CO2.

Response:      All isotopologues are included in the total $CO_2$.  Dividing the sum of the
measured isotopologues (x626, x636, and x628, plus x627 calculated using the $\delta^{17}O$ to $\delta^{18}O$
relationship) by the sum of the probabilities for the 4 major isotopologues corrects the sum for
15   the unmeasured rare isotopologues. The probabilities used in the equation are determined for the
unknown sample based on it's measured isotopologue mole fractions. Generally they are very
slightly different than the sum of the probabilities assuming VPDB values but we calculate them
anyway. Text has been changed to clarify.

Comment:      Page 16 line 14: no information on the uncertainty for the standards is given.
Please add this.

Response:      The uncertainty of the manometric method used to assign the primary standards
25   was given on page 3.  It has been re-stated in this section to clarify.

Comment:      Page 17 line 10 and 11: It would be useful to know if INSTAAR are using a
second reference material to control scale contraction effects to substantiate this conclusion.
30
Response:      INSTAAR is not using a second reference material to control scale contraction.
The published INSTAAR vs JRAS offsets has been referenced as evidence that scale contraction
at INSTAAR is the cause of the offsets seen between NOAA and INSTAAR when measuring
highly depleted tanks.
35
Comment:      Page 17 lines 22-23. Reproducibility and uncertainty appear to be used as
synonyms, which they are not. The author's should differentiate between the reproducibility and
uncertainty, and an estimation of the measurement uncertainty would help in this respect.

40   Response:      This was unintentional word choice.  Text has been added to clarify that
reproducibility is one component of uncertainty.  Even without a full uncertainty budget the
reproducibility estimate shows the $\delta^{13}C$ and $\delta^{18}O$ calibrations are not applicable as standards for
high precision $CO_2$ isotopic measurements. Full uncertainty calculations for the $CO_2$ scale will
be in an upcoming publications describing the revision of the WMO $CO_2$ scale since the largest
45   terms in the uncertainty budget is related to the manometer measurements.

Comment:     Page 26 Table 1: The currently internationally accepted conventional values for VPDB should be clearly identified in this Table.

Response:     See response to previous comment on values for reference materials.

**Abundances of isotopologues and calibration of $CO_2$ greenhouse gas measurements**

Pieter P. Tans[1,3], Andrew M. Crotwell[2,3], and Kirk W. Thoning[1,3]

[1]Global Monitoring Division, Earth System Research Laboratory, National Oceanic and Atmospheric Administration, Boulder, Colorado, 80305, USA.
[2]Cooperative Institute for Research in Environmental Sciences, University of Colorado, Boulder, Colorado, 80309, USA.
[3]Central Calibration Laboratory, World Meteorological Organization Global Atmosphere Watch program (WMO/GAW)

*Correspondence to:* Pieter Tans (Pieter.Tans@noaa.gov)

**Abstract**

We have developed a method to calculate the fractional distribution of $CO_2$ across all of its component isotopologues based on measured $\delta^{13}C$ and $\delta^{18}O$ values. The fractional distribution can be used with known total $CO_2$ to calculate the amount of substance fraction (mole fraction)  of each component isotopologue in air individually. The technique is applicable to any molecule where isotopologue-specific values are desired. We used it with a new $CO_2$ calibration system to account for isotopic differences among the primary $CO_2$ standards that define the WMO X2007 $CO_2$ in air calibration scale and between the primary standards and standards in subsequent levels of the calibration hierarchy. The new calibration system uses multiple laser spectroscopic techniques to measure mole fractions  of the three major $CO_2$ isotopologues ($^{16}O^{12}C^{16}O$, $^{16}O^{13}C^{16}O$, and  $^{16}O^{12}C^{18}O$) individually. The three measured values are then combined into total $CO_2$ (accounting for the rare unmeasured isotopologues), $\delta^{13}C$, and $\delta^{18}O$ values. The new calibration system significantly improves our ability to transfer the WMO $CO_2$ calibration scale with low uncertainty through our role as the World Meteorological Organization Global Atmosphere Watch Central Calibration Laboratory for $CO_2$. Our current estimates for reproducibility of the new calibration system are $\pm$-0.01 $\mu$mol mol$^{-1}$ $CO_2$, $\pm$-0.2 ‰ $\delta^{13}C$, and $\pm$-0.2 ‰ $\delta^{18}O$, all at 68 % confidence interval (CI).

**1 Introduction**

Long-term atmospheric monitoring of the greenhouse gases relies on a stable calibration scale to be able to quantify small spatial gradients and temporal trends. Small changes in trends and spatial gradients result from realignments in the strengths of emissions ("sources") and removals ("sinks") of the greenhouse gases. Inconsistent scale propagation to atmospheric measurements would give biased results from one monitoring station or network to the next that would be attributed incorrectly to sources/sinks by atmospheric transport models. Preventing biased results from various national monitoring networks enables improved understanding of the carbon cycle and its response to human intervention and climate change. It has now become even more important as countries have pledged

[revised manuscript text omitted]

The subject of this paper is the transfer of the scale to lower level standards and its uncertainty. We do not discuss the total uncertainty of the primary scale itself. It is described in a separate paper (Hall et al., in preparation). The

20 transfer of the scale from primary to secondary standards and hence to tertiary standards (which are used as working standards by NOAA and delivered to other organizations) has been done historically using nondispersive infrared absorption spectroscopy (NDIR). The secondary standards are used to prolong the lifetime of the primary standards. The current primary standards have been in use for nearly 25 years and provide a consistent scale over that time period. All measurements by NOAA and WMO GAW contributing programs are directly traceable to this single set

25 of primary standards through a strict hierarchy of calibration.

The transfer of the scale from primary to secondary standards has typically been done using a subset of 3 or 4 primary standards rather than the entire set of 15 primary standards. This was done because we wanted to perform a local curve fit of the non-linear NDIR response while also minimizing use of the primary standards. The subset of primary standards chosen was a function of the expected $CO_2$  in the secondary standards and was

30 designed to closely bracket the expected values with a small range of $CO_2$ in the primary standards. The relatively large uncertainty of the individual manometric assigned values would potentially introduce significant biases due to the use of subsets of primary standards. To prevent these biases, the individual manometrically assigned values of the primary standards were corrected based on the residuals to a consistency fit of almost all primary standards (usually without the highest and the lowest primary) run on the NDIR. The re-assigned values (average manometer

35 value minus the residual) were assumed to be the best assigned value for the primary standards. This in theory

**Comment [pt1]:** I don't think "absolute" is used in metrology. Instead we can insert that pressure and temperature are traceable to SI.

**Comment [AMC2]:** Pieter, check this wording and change if you think it should be different. This is a correct description. The equation of state, with known pressures, temperatures, and a volume ratio, results in a ratio of number of moles of gas in the large and in the small volume.

**Comment [pt3]:** We need to state this clearly up front. There was some confusion among reviewers about it.

should allow the use of subsets of the primary standards when transferring the scale from primary to secondary. In practice, as will be shown, there are still possible biases due to the grouping of primary standards based on expected $CO_2$ of the secondary standards. Tertiary standards were calibrated similarly against closely spaced subsets of secondary standards that bracketed the expected values of the tertiary standards.

5      New analytical methods developed over the last several years have greatly improved the ability of monitoring stations to measure $CO_2$. These new analytical techniques and improved diligence of monitoring network staff are pushing the uncertainties of measurements lower and improving the network compatibility. Current  reproducibility of standards using the NDIR calibration system is 0.03 μmol mol$^{-1}$ (68_% CI) ("Carbon Dioxide WMO Scale", 2017). This is a significant component of the targeted 0.1 μmol mol$^{-1}$ (or 0.05 μmol mol$^{-1}$ in the

10     Southern Hemisphere) network compatibility goal (WMO, 2016). Improvements in the scale propagation uncertainty would help monitoring programs achieve the compatibility goals. We have therefore undertaken to improve our calibration capabilities and to address key uncertainty components of the scale transfer. These key components are the reproducibility of the scale transfer, the potential for mole fraction dependent biases, and  the potential issues we describe in this paper relating to the isotopic composition of

15     the primary standards and subsequent standards in the calibration hierarchy.

**2 Isotopic influence on $CO_2$ measurement**

The WMO $CO_2$ mole fraction scale is defined as the number of moles of $CO_2$ per mole of dry air, without regard to its isotopic composition. An isotopologue of $CO_2$ has a specific isotopic composition. The five most abundant

20     $CO_2$ isotopologues, in order of abundance, are: $^{16}O^{12}C^{16}O$, $^{16}O^{13}C^{16}O$,  $^{16}O^{12}C^{18}O$,  $^{16}O^{12}C^{17}O$ and  $^{16}O^{13}C^{18}O$ (referred to in equations in this work by the HITRAN (Rothman, 2013) shorthand notations 626, 636, 628, 627, and 638 respectively). For $CO_2$ the two oxygen positions are equivalent due to the symmetry of the molecule so the position of the oxygen isotopes does not matter. The abundance of the radioactive $^{14}C$ relative to $^{12}C$ is ~$10^{-12}$; which is too small to be of significance in this context. Analysts need to take

25     into account differences in the relative sensitivity of their analyzers to different isotopologues (or isotopomers, see below) as well as differences in the isotopic composition of sample and standard gases.

Isotopic composition is typically measured by isotope ratio mass spectroscopy (IRMS) and is reported as the difference in the minor isotope to major isotope ratio (i.e. $^{13}C/^{12}C$) from the ratio of an accepted standard reference material. For example, the $^{13}C$  isotope delta value ($\delta^{13}C$) is defined as:

$$\delta^{13}C = \frac{\left(\frac{^{13}C}{^{12}C}\right)_{Sample} - \left(\frac{^{13}C}{^{12}C}\right)_{Standard}}{\left(\frac{^{13}C}{^{12}C}\right)_{Standard}} * 1000$$

(1)

Where $(^{13}C/^{12}C)_{sample}$ and $(^{13}C/^{12}C)_{standard}$ are the $^{13}C$ to $^{12}C$ isotopice abundance ratios for the sample and the standard reference material respectively. The international accepted scale for $^{13}C$ is the Vienna Pee Dee Bellemnite (VPDB)

5    scale, realized as calcium carbonate. Oxygen isotopic ratios ($^{18}O/^{16}O$ or $^{17}O/^{16}O$) in $CO_2$ are described with a similar isotope delta notation relative to an accepted reference material. Isotope delta values for carbon and oxygen are typically reported in units of per mil (‰) by multiplying Eq. (1) by 1000. For many applications, the $^{17}O$ isotope is not actually measured but is assumed to follow a mass dependent relationship with $^{18}O$ where $\delta^{17}O \approx 0.528 * \delta^{18}O$. This approximation is adequate for the purpose of defining the oxygen isotopic effects on atmospheric $CO_2$

10   measurements. For more detailed descriptions of this relationship see Santrock et al. (1985), Assonov and Brenninkmeijer (2003), Brand et al. (2010) and references therein. Oxygen isotopes can be related to either Vienna Standard Mean Ocean Water (VSMOW) or to VPDB-$CO_2$, with the latter commonly used in the atmospheric $CO_2$ community. The VPDB-$CO_2$ scale relates to the $CO_2$ gas evolved from the calcium carbonate material itself during the reaction with phosphoric acid and accounts for oxygen fractionation that occurs during the reaction (Swart et al.,

15   1991). In this paper all oxygen isotope values are referenced to the VPDB-$CO_2$ scale unless otherwise noted.

$CO_2$ analysers are not equally sensitive to the isotopologues of $CO_2$. For example, gas chromatography where $CO_2$ is reduced to $CH_4$ and detected with a flame ionization detector (GC-FID) (Weiss, 1981) is equally sensitive to all isotopologues whereas laser based absorption techniques that measure an absorption line from the single major $^{16}O^{12}C^{16}O$ isotopologue are blind to all of the minor isotopologues. NDIR instruments are much more complicated

20   in their response to the various minor isotopologues of $CO_2$. Most NDIR analyzers use an optical band pass filter to limit the wavelengths of light reaching the detectors. These filters often exclude part of the absorption bands of the minor isotopologues (e.g. Tohjima et al. 2009), but are more sensitive to the $^{16}O^{13}C^{16}O$ lines within the pass band because absorption of the much stronger $^{16}O^{12}C^{16}O$ lines is partially saturated. The width and shape of the transmission window of the filter is generally not identical between instruments. Tohjima et al. (2009) found

25   significant differences in the sensitivity to the minor isotopologues between three different LI-COR NDIR analyzers. In addition, Lee et al. (2006) found the response of a Siemens ULTRAMAT 6E NDIR analyzer to be almost completely insensitive to the minor isotopologues.

[revised manuscript text omitted]

As used here the symbols $^{13}C$, $^{18}O$, etc. stand for amounts. It will simplify derivations below if we use isotopic ratios as amount ratios relative to all carbon, oxygen, etc., similar to mole fractions in air. We give these  isotope-amount fractions the symbol "$x$" instead of "$r$".

$$^{13}x \overset{\text{def}}{=} \frac{^{13}C}{^{12}C + ^{13}C} \qquad ^{18}x \overset{\text{def}}{=} \frac{^{18}O}{^{16}O + ^{17}O + ^{18}O} \quad ==> \quad ^{16}x = 1 - \frac{^{17}O + ^{18}O}{^{16}O + ^{17}O + ^{18}O}$$

These definitions lead to the following relationships:

$$^{13}x = \frac{^{13}r}{1 + ^{13}r} \qquad ^{13}r = \frac{^{13}x}{1 - ^{13}x} \tag{2}$$

The equivalents for oxygen are:

$$^{17}x = \frac{^{17}r}{1 + ^{17}r + ^{18}r} \qquad ^{17}r = \frac{^{17}O}{^{16}O} = \frac{^{17}O/(^{16}O + ^{17}O + ^{18}O)}{^{16}O/(^{16}O + ^{17}O + ^{18}O)} = \frac{^{17}x}{1 - ^{17}x - ^{18}x} \tag{3}$$

and similarly for $^{18}x$ and $^{18}r$ .

From here on we will abbreviate  VPDB-CO2 as VPDB to keep the notation manageable. Using Table 1 and these conventions gives us

$$^{13}x_{\text{VPDB}} = 0.0110564 \tag{4a}$$

$$^{17}x_{\text{VPDB}} = 395.11 \ 10^{-6} / (1 + 2088.35 \ 10^{-6} + 395.11 \ 10^{-6}) = 394.1 \ 10^{-6} \tag{4b}$$

$$^{18}x_{\text{VPDB}} = 2088.35 \ 10^{-6} / (1 + 2088.35 \ 10^{-6} - + 395.11 \ 10^{-6}) = 2083.2 \ 10^{-6} \tag{4c}$$

Isotopic ratio measurements have always been expressed in terms of their (typically) small difference from the standard reference materials, in the so-called delta notation:

$$^{13}\delta \overset{\text{def}}{=} (^{13}r - ^{13}r_{\text{VPDB}})/^{13}r_{\text{VPDB}} = ^{13}r/^{13}r_{\text{VPDB}} - 1, \text{ so that } ^{13}r - ^{13}r_{\text{VPDB}} = ^{13}r_{\text{VPDB}} \ ^{13}\delta \tag{5}$$

Comment [pt9]: This is correct – it is the isotope-amount ratio as described by Coplen. I like to keep the bold notation for better readability.

Comment [pt10]: I experimented here with increasing the font from 10 to 12 just in the equations to make them more readable

Comment [pt11]: Now we are following Not sure if we should use a new definition for VPDB

Comment [pt12]: I have been mucking with these values – please check

and similarly for $^{17}\delta$ and $^{18}\delta$

By analogy we define for the amount fractions $x$ :

$$^{13}\Delta \overset{\text{def}}{=} (^{13}x - {}^{13}x_{\text{VPDB}})/{}^{13}x_{\text{VPDB}} = {}^{13}x/{}^{13}x_{\text{VPDB}} - 1, \text{ so that } {}^{13}x - {}^{13}x_{\text{VPDB}} = {}^{13}x_{\text{VPDB}}\ {}^{13}\Delta \qquad (6)$$

and similarly for $^{17}\Delta$ and $^{18}\Delta$

5   In the above (Eqs. (5) and (6)) and the rest of this work we will express $\delta$ and $\Delta$ as small numbers, not in the "permil" (‰) notation, in which every delta value is multiplied by 1000. For example δ=0.020 would normally be written as 20 permil or 20 ‰. To keep the notation economical and the paper more readable we introduced simplified notations such as $^{13}r$ and $^{13}\delta$ instead of $r(^{13}\text{C}/^{12}\text{C})$ and $\delta^{13}\text{C}$ in equations. This produces no ambiguities. In addition, in this paper we need to distinguish between isotope-amount fractions *within* $CO_2$ (denoted "$x$" above in

10   accordance with Coplen (2011)) of isotopes (and isotopologues) from mole fraction in air. We normally denote mole fraction in air by "$x$" or "$X$", but here we use "$y$" (in accordance with notation recommendations for gas mixtures (IUPAC, 2006)) to distinguish mole fraction in air from isotope or isotopologue amount fraction. For example, $x(636)$ is the amount fraction of $^{16}\text{O}^{13}\text{C}^{16}\text{O}$ to all isotopologues of $CO_2$ whereas $y(636)$ is the mole fraction of the $^{16}\text{O}^{13}\text{C}^{16}\text{O}$ isotopologue in air.

**4 Fractional abundances of isotopologues in molecules.**

Converting measured $\delta^{13}\text{C}$ and $\delta^{18}\text{O}$ values into $^{16}\text{O}^{13}\text{C}^{16}\text{O}$  and  $^{16}\text{O}^{12}\text{C}^{18}\text{O}$  isotopologue abundances is not straightforward due to the rare $^{17}\text{O}^{12}\text{C}^{16}\text{O}$  and doubly substituted isotopologues. Isotope ratio mass spectrometry (IRMS) determines $\delta^{13}\text{C}$ and $\delta^{18}\text{O}$ values by measuring molecular

20   mass 45/44 and  46/44 ratios, with appropriate corrections for interfering masses, relative to a standard reference material. These mass ratios can be used with the accepted isotopic ratios of the standard reference materials to *approximate* the abundance as amount  fraction ($x$) of the three main isotopologues in $CO_2$ using:

$$x(636) \cong {}^{13}x, \quad x(628) \cong 2*{}^{18}x, \quad x(626) \cong 1 - x(636) - x(628) \qquad (7)$$

The oxygen abundance ratio is multiplied by a factor of two in Eq. (7) to convert the amount fractions

25    from atomic abundance (i.e. $^{18}\text{O}/^{16}\text{O}$) into molecular abundance. The approximations in Eqs. (7)  ignore the contribution of the oxygen isotopes to $x$(636) and of $^{13}\text{C}$ to $x$(628), as well as the portion of the total composed of the rare isotopologues. Depending on the level of uncertainty desired this may or may not be acceptable. As the WMO GAW CCL for $CO_2$, NOAA is obligated to minimize biases in the $CO_2$ calibration scale, and therefore we will correctly account for the apportionment of $CO_2$ through all isotopologues. The same technique

30   was developed independently by Flores et al. (2017) for use in calibrating spectroscopic instruments for $\delta^{13}\text{C}$ and $\delta^{18}\text{O}$ measurements. Here our focus is on total $CO_2$ measurements that account for isotopic differences between standards.

We start by assuming a purely statistical distribution of $^{13}C$, $^{18}O$, and other atoms when putting together a molecule starting from atomic amount fractions  as given for standard reference materials  (Table 1), namely, that the probability of picking a particular isotope is not affected by what is picked before or later. In general the other picks can affect the probability a little (called "clumped" isotopes), so that the thermodynamic

5  abundances are slightly different from the statistical distribution. We will ignore that for now, and construct a purely statistical baseline distribution for the reference. It is important to note that Thermodynamic and kinetic fractionation effects are reflected in actual measured delta values and fractionation factors relative to the agreed upon reference material. Thus the probability of picking a $^{13}C$ atom for a carbon position is defined as simply $^{13}R_x$.  However, a molecule may have more than one position for

10  C, O, N, etc.  For example, suppose there are N chemical positions for a particular atom in a molecule and we want to define the probability of M of those positions being filled with one particular isotope (denoted isotope a). If the locations of the M, as a subset of N, do not matter, as is the case for symmetrical molecules like $CO_2$ and $CH_4$, we could call the N positions equivalent. In that case the probability is

$$P = \binom{N}{M} * x_a{}^M * x_b{}^{N-M} \tag{$\cancel{10}$8}$$

15  $x_a$ is the amount fraction of isotope a, $x_b$ is the amount fraction  of other isotopes ($x_b = 1 - x_a$). The first term in  Eq. (8) is the statistical weight which equals the number of combinations (a statistical term, the order of picking the M does not matter) of M out of N, given as

$$\frac{N!}{M!(N-M)!} \stackrel{\text{def}}{=} \binom{N}{M} \tag{$\cancel{11}$9}$$

N! is the factorial notation, $N! \stackrel{\text{def}}{=} 1 * 2 * 3 * \ldots \ldots (N-1) * N$, with the special case $0! \stackrel{\text{def}}{=} 1$

20  Example: there are two equivalent positions for a single $^{18}O$ in $CO_2$, namely $^{18}O^{12}C^{16}O$ and $^{16}O^{12}C^{18}O$, jointly denoted as "628" (one $^{18}O^{16}O$, one $^{12}C$, one $^{16}O^{18}O$), so that the statistical weight is

$$\binom{2}{1} = \frac{2!}{1! * (2-1)!} = 2 .$$

Or for methane, a single or double substitution of deuterium ($^2H$) for $^1H$ has respective statistical weights:

$$\binom{4}{1} = \frac{4!}{1! * (4-1)!} = 4 \qquad \binom{4}{2} = \frac{4!}{2! * (4-2)!} = 6$$

It should be noted that whether positions can be considered equivalent depends on the symmetry of the molecule and the measurement method. For example for nitrous oxide the two positions for N in NNO would be equivalent when

25  mass 45 (one $^{14}N$, one $^{15}N$, one $^{16}O$) is measured in a mass spectrometer but they are not when an optical absorption method is used because the spectrum of $^{14}N^{15}N^{16}O$ is different from $^{15}N^{14}N^{16}O$. In the latter case we need to keep

separate track of the probabilities, denoted below as "P", of these two isotopomers. Isotopomers have the same number of specific isotopes, but they differ in their position in the molecule.

The probability for any particular $CO_2$ isotopologue is the product of the probability of picking the carbon isotope and the probability of picking the oxygen isotopes. Each of these probabilities is determined using Eq. (8). For example, the probability for the  $^{16}O^{13}C^{18}O$ isotopologue is the probability of picking one $^{13}C$ isotope for one carbon position times the probability of picking one $^{18}O$ isotope for one of the two oxygen positions and one $^{16}O$ for the other.

The equations below give the probabilities for individual $CO_2$ isotopologues. When the isotopic compositions of the standard reference materials (VPDB in Table 1) are filled in we obtain the numbers after the "=>" sign.

$$P(626) = (1-^{13}x)*(1-^{17}x-^{18}x)^2 \qquad => 0.98404985 = 1-0.01595015 \qquad (10)$$

$$P(636) = {}^{13}x*(1-^{17}x-^{18}x)^2 \qquad => 0.01101688 \qquad (11)$$

$$P(628) = (1-^{13}x)*2*^{18}x*(1-^{17}x-^{18}x) \qquad => \qquad 0.0041101273$$
$$(12)$$

The sum of the above three major abundances is $0.99916166 = 1- 0.00083834$

$$P(627) = (1-^{13}x)*2*^{17}x*(1-^{17}x-^{18}x) \qquad => \qquad 0.000777554$$
$$(13)$$

The sum of the above four major abundances is $0.99993922 = 1- 0.00006078$

$$P(638) = {}^{13}x*2*^{18}x*(1-^{17}x-^{18}x) \qquad => 4.59513 \ 10^{-5}$$

and so on, with progressively smaller probabilities. The sum of all probabilities equals 1, which was verified digitally in double precision. This example was for VPDB, but in any population of $CO_2$ molecules, (i.e. in a sample or standard cylinder) probabilities determined from the isotope-amount fractions of the population equate to the  fractional abundance of each isotopologue.

**5 An expression for potential effects of isotopic mismatches on measurements of $CO_2$**

In this section we derive some practical expressions for biases, and corrections, resulting from isotopic mismatches if they are ignored, for the case of $CO_2$. Similar considerations apply to other greenhouse gases such as $CH_4$, $N_2O$, etc. Such corrections can be generally applied to $CO_2$ measurements if desired. The unknown quantity of $CO_2$-in-air that we intend to measure is called "measurand". It can be a real air sample or an intermediate transfer standard.

Comment [pt15]: Reference: IUPAC, Compendium of Chemical Terminology, 2nd ed. (the "Gold Book") (1997)

Formatted

Formatted

 Here we give an example for an instrument that quantifies the mole fraction of total $CO_2$ in air, denoted $y_{CO2}$, by measuring only one isotopologue, namely $^{16}O^{12}C^{16}O$.  We assume that the instrument is calibrated by a $CO_2$ standard

5    with amount fraction $^{13}x_{VPDB}$ and $^{18}x_{VPDB}$ of the two main isotopologues, corresponding to the international VPDB reference points for $^{13}C$ and $^{18}O$. In almost all cases deviations of $^{17}x$ from VPDB are tightly correlated with deviations of $^{18}x$ from VPDB. The deviation of total $CO_2$ from being proportional to P(626) due to inconsistencies of $^{13}x$, $^{17}x$, $^{18}x$ between the measurand and VPDB, using Eq. (810), is

10   $$\Delta P(626) \overset{\text{def}}{=} P(626) - P_{VPDB}(626) =$$

$$\frac{\partial P(626)}{\partial ^{13}x}(^{13}x - ^{13}x_{VPDB}) + \frac{\partial P(626)}{\partial ^{17}x}(^{17}x - ^{17}x_{VPDB}) + \frac{\partial P(626)}{\partial ^{18}x}(^{18}x - ^{18}x_{VPDB})$$

(14)

The above are the first terms of a Taylor expansion around $P_{VPDB}(626)$. Inserting the first derivatives and using Eq. (6) gives:

15   $$\Delta P(626) = -\left(1 - ^{17}x_{VPDB} - ^{18}x_{VPDB}\right)^2 \left(^{13}x_{VPDB} {}^{13}\Delta\right) +$$

(15)

$$-2\left(1 - ^{13}x_{VPDB}\right)\left(1 - ^{17}x_{VPDB} - ^{18}x_{VPDB}\right)\left(^{17}x_{VPDB} {}^{17}\Delta + ^{18}x_{VPDB} {}^{18}\Delta\right)$$

If $^{13}\Delta$ is positive the air to be measured has a higher $^{13}C/^{12}C$ ratio than VPDB. Therefore P(626) is slightly lower than it is for VPDB, and the relative correction in the mole fraction assigned to the measured air will have to be positive, of opposite sign to the relative error of P(626):

20   $$\frac{\Delta y_{CO2}}{y_{CO2}} = -\frac{\Delta P(626)}{P(626)} = \frac{^{13}x_{VPDB} {}^{13}\Delta}{\left(1 - ^{13}x_{VPDB}\right)} + \frac{2\left(^{17}x_{VPDB} {}^{17}\Delta + ^{18}x_{VPDB} {}^{18}\Delta\right)}{\left(1 - ^{17}x_{VPDB} - ^{18}x_{VPDB}\right)}$$

(16)

We note here that we could have used a $^{16}O^{13}C^{16}O$ line for quantifying $y_{CO2}$, but an analogous derivation for $\Delta P(636)/P(636)$ shows that it is 90 times more sensitive to isotopic errors or mismatches.

Using Eqs. (2) and (3) gives

25   $$\frac{\Delta y_{CO2}}{y_{CO2}} = ^{13}r_{VPDB}\,^{13}\Delta + 2\left(^{17}r_{VPDB}\,^{17}\Delta + ^{18}r_{VPDB}\,^{18}\Delta\right)$$

(17)

Generally, one is not making atmospheric $CO_2$ measurements with standards that have isotopic abundances for C and O exactly like VPDB. Because the linear Eq. (17) applies to the measurement of a transfer standard itself as well as to an air sample, we can give an expression for corrections to be made when the standard (subscript "st") has

30   an isotopic composition different from air but not equal to VPDB:

$$\frac{\Delta xy_{CO2}}{xy_{CO2}} = {}^{13}r_{VPDB}\left({}^{13}\Delta_{air} - {}^{13}\Delta_{st}\right) + 2\left({}^{17}r_{VPDB}\left({}^{17}\Delta_{air} - {}^{17}\Delta_{st}\right) + {}^{18}r_{VPDB}\left({}^{18}\Delta_{air} - {}^{18}\Delta_{st}\right)\right)$$
(18)

In the Appendix we derive the following very close approximation to Eq. (18) in which the $\Delta$ values have been replaced by the familiar $\delta$ values:

$$\Delta xy_{CO2} = xy_{CO2}\left[0.011061\left({}^{13}\delta_{air} - {}^{13}\delta_{st}\right) + 2*0.0023\left({}^{18}\delta_{air} - {}^{18}\delta_{st}\right)\right]$$
(19)

This is an expression for $CO_2$ corrections when only the ${}^{16}O^{12}C^{16}O$ isotopologue is used to measure $xy_{CO2}$, and we are using VPDB scales. As an example, if we use a standard with $CO_2$ made from natural gas, it could have ${}^{13}\delta_{st}$ = −0.045 and ${}^{18}\delta_{st}$ = −0.017 on the VPDB scales, whereas air has ${}^{13}\delta_{air} \cong$ −0.008 and ${}^{18}\delta_{air} \cong$ 0.000. Assuming $xy_{CO2}$ = 400 μmol mol$^{-1}$, then $\Delta y$ = 0.164 +0.031 = 0.195 μmol mol$^{-1}$. ${}^{13}\delta_{air}$ is higher than ${}^{13}\delta_{st}$, so that the ${}^{16}O^{12}C^{16}O$ abundance of the standard is higher than assumed, resulting in the air measurement being too low. Therefore an upward correction is needed for ${}^{13}C$ and likewise for ${}^{18}O$.

**6 Practical calculations for definition and propagation of the $CO_2$ calibration scale**

Equations (18) and (19) gives the correction required when only the ${}^{16}O^{12}C^{16}O$ isotopologue is used to determine $xy_{CO2}$ as a function of the isotopic differences between the sample and a single standard. However, most $CO_2$ measurements are made vs a suite of standards that may have various isotopic compositions and the isotopic compositions may be a function of $CO_2$ (as is the case for the primary $CO_2$ standards used by the WMO GAW CCL). In this case the calibration curve that defines the response of the analyser may incorporate a systematic error making the idea of a simple "correction" impractical. Equations (18) and (19) can be used to estimate the potential offsets due to sample/standard isotopic differences but is not practical for making corrections when multiple standards are used. Therefore, we must instead use a calibration approach that fully accounts for the isotopic composition of the standards rather than using a post measurement correction.

A great advantage of our method is that it uses multiple standards covering a range of values to create a scale. "Scale contraction" can result from having a single standard reference, and mass spectrometer measurements have suffered from that. We do not have such "contraction" because we have a real scale over the full range of interest instead of a single point so that we can create a response curve for any analyzer. Secondly, having such isotopologue specific response curves over a large range also opens the possibility to make a new determination of the value of $({}^{13}C/{}^{12}C)_{VPDB}$. One could make for example 400 ppm and 800 ppm CO2 in air mixtures with their isotopic ratios

We have taken the approach of decomposing the total $CO_2$ in the primary standards, as defined by manometric measurements, into individual isotopologue mole fractions in air based on measured $\delta^{13}C$ and $\delta^{18}O$ values. The $\delta^{13}C$ and $\delta^{18}O$ values are determined by IRMS by the Stable Isotope Laboratory, Institute of Arctic and Alpine Research, University of Colorado, Boulder (INSTAAR) on their own realization of the VPDB scales. The current scales used by INSTAAR were set using NBS-19 and NBS-20 (carbonates) and VSMOW, GISP and SLAP (waters) (Trolier et al., 1996). These isotopologue specific mole fractions in air of the standards are used to calibrate laser-based spectroscopic instruments for the three major $CO_2$ isotopologues ($^{16}O^{12}C^{16}O$, $^{16}O^{13}C^{16}O$, and $^{18}O^{12}C^{16}O$ $^{16}O^{12}C^{18}O$) individually. The three major isotopologues in unknown cylinders are measured relative to these isotopologue specific calibration curves. The isotopologue mole fractions in air of the unknowns are then recombined into total $CO_2$ and conventional $\delta^{13}C$, and $\delta^{18}O$ values while properly accounting for the non-measured rare isotopologues.

A great advantage of our method is that it uses multiple standards covering a range of values to create a scale. "Scale contraction" can result from having extrapolating from a single standard reference, and mass spectrometer measurements have suffered from that. We do not have such "contraction" because we have a real scale calibrate with multiple standards over the full range of interest instead of using a single point so that we can create a response curve for any analyzer. Having such isotopologue specific response curves over a large range also opens the intriguing possibilities of making $CO_2$ isotopic scales that are traceable to SI and improving our understanding of VPDB and its relation with LSVEC. This may be beyond the scope of our laboratory but we offer it as an interesting aside. Secondly, having such isotopologue specific response curves over a large range also opens the possibility to make a new determination of the value of $(^{13}C/^{12}C)_{VPDB}$. One could make for example 400 ppm and 800 ppm $CO_2$ in air mixtures with their isotopic ratios close to the VPDB values. Then a small and well known amount of pure $^{13}CO_2$ could be added to the 400 ppm standard so that its $^{636}$ isotopologue mole fraction in air ends up close to, say, that of the 800 ppm mixture.added needed to double the amount of 636 that corresponds to VPDB provides a measure of VPDB itself.

Suppose we have one or more instruments measuring each isotopologue ($^{16}O^{12}C^{16}O$, $^{16}O^{13}C^{16}O$, $^{18}O^{12}C^{16}O$ $^{16}O^{12}C^{18}O$ and perhaps also $^{17}O^{12}C^{16}O$ $^{16}O^{12}C^{17}O$) individually. The response of the instrument(s) for each of the isotopologues needs to be calibrated separately. How often such calibrations need to be repeated depends on the instrument. For this purpose we need to have a series of reference gas standards with well defined total $CO_2$ ($X_{CO2}$ $x_{CO2}$) and with known conventional $\delta$-values for the isotopic ratios. Equations (12 10)-(15 13) can be used to convert that information to the fractional abundances of the isotopologues, by first writing them in terms of conventional delta values by using relations (2) and (3) and by writing $r_{sample}$ as $r_{PVPDB}(1+\delta)$ (see Eq. (5)).

$$P(626) = \frac{1}{1+^{13}r_{\text{VPDB}}(1+^{13}\delta)} \; \frac{1}{\left[1+^{17}r_{\text{VPDB}}(1+^{17}\delta)+^{18}r_{\text{VPDB}}(1+^{18}\delta)\right]^2}$$

(20)

$$P(636) = \frac{^{13}r_{\text{VPDB}}(1+^{13}\delta)}{1+^{13}r_{\text{VPDB}}(1+^{13}\delta)} \; \frac{1}{\left[1+^{17}r_{\text{VPDB}}(1+^{17}\delta)+^{18}r_{\text{VPDB}}(1+^{18}\delta)\right]^2}$$

(20)

$$P(\underline{628}) = \frac{1}{1+^{13}r_{\text{VPDB}}(1+^{13}\delta)} \; \frac{2*^{18}r_{\text{VPDB}}(1+^{18}\delta)}{\left[1+^{17}r_{\text{VPDB}}(1+^{17}\delta)+^{18}r_{\text{VPDB}}(1+^{18}\delta)\right]^2}$$

(21)

$$P(\underline{627}) = \frac{1}{1+^{13}r_{\text{VPDB}}(1+^{13}\delta)} \; \frac{2*^{17}r_{\text{VPDB}}(1+^{17}\delta)}{\left[1+^{17}r_{\text{VPDB}}(1+^{17}\delta)+^{18}r_{\text{VPDB}}(1+^{18}\delta)\right]^2}$$

(22)

If $\delta^{17}O$ has not been measured, we approximate $\delta^{17}O = 0.528 * \delta^{18}O$ to determine the fractional abundances above.

The fractional abundances (Eqs. (20)-(22)) are converted into mole fractions in dry air by multiplying with the total mole fraction of $CO_2$ in dry air ($\underline{y}_{CO2}$). The isotopologue mole fractions in air are written as $\underline{y}(626)$, etc. In other words, we have $\underline{y}(626) = \underline{y}_{CO2} * P(626)$ and similar for all isotopologues.

A series of standards can in this way be used to calibrate the instrument response for each isotopologue individually. With these response functions we can then assign mole fractions in air to the isotopologues of the unknown gas mixtures that are being measured, $\underline{y}(626)_{\text{unk}}$, etc.

Then we need to convert the measured isotopologue  mole fractions of the unknown ($\underline{y}(626)_{\text{unk}}$, $\underline{y}(636)_{\text{unk}}$, and $\underline{y}(628)_{\text{unk}}$) back to standard delta-notation using Eqs. (20)-(23) as follows:

$$\frac{\underline{y}(636)_{\text{unk}}}{\underline{y}(626)_{\text{unk}}} = \frac{P(636)}{P(626)} = {}^{13}r_{\text{VPDB}}(1+{}^{13}\delta) \;\;\Rightarrow\;\; {}^{13}\delta = \frac{\underline{y}(636)_{\text{unk}}}{^{13}r_{\text{VPDB}}*\underline{y}(626)_{\text{unk}}} - 1$$

(24)

 (24)

$$\frac{\underline{y}(826)_{\text{unk}}}{\underline{y}(626)_{\text{unk}}} = \frac{P(826)}{P(626)} = 2 * {}^{18}r_{\text{VPDB}}(1+{}^{18}\delta) \;\;\Rightarrow\;\; {}^{18}\delta = \frac{\underline{y}(826)_{\text{unk}}}{2*^{18}r_{\text{VPDB}}*\underline{y}(626)_{\text{unk}}} - 1$$

(25)

and similarly for $\delta^{17}O$. If $\delta^{17}O$ has not been measured we assume that $\delta^{17}O = 0.528 * \delta^{18}O$.

$$\frac{X(826)_{unk}}{2 * {}^{18}r_{PDB} * X(626)_{unk}} = 1 + [....] + {}^{18}\delta\,(1 + [....])$$

in which we defined $[....] = {}^{17}r_{PDB}(1 + {}^{17}\delta) + {}^{18}r_{PDB}(1 + {}^{18}\delta)$

and rearrange it further into:

$${}^{18}\delta = \left[\frac{X(826)_{unk}}{2*{}^{18}r_{PDB}*X(626)_{unk}} - 1\right] - [(1 + {}^{18}\delta)({}^{17}r_{PDB}(1 + {}^{17}\delta) + {}^{18}r_{PDB}(1 + {}^{18}\delta))] \qquad (25)$$

The first approximation to ${}^{18}\delta$ is to assume $\delta^{18}O$ and $\delta^{17}O = 0$ on the right hand side, i.e. equal to the standard reference material:

$${}^{18}\delta = \left[\frac{X(826)_{unk}}{2 * {}^{18}r_{PDB} * X(626)_{unk}} - 1\right] - {}^{17}r_{PDB} - {}^{18}r_{PDB}$$

Then we substitute this first approximation into Eq. (25), with the assumption of $\delta^{17}O = 0.528 * \delta^{18}O$, and iterate the solution for ${}^{18}\delta$ by continuing to substitute it in the right hand side of Eq. (25). The approximation is extremely close to the full solution unless the sample is highly depleted in $\delta^{18}O$.

If $X(726)_{unk}$ has been measured, an equation similar to Eq. (25) applies:

$${}^{17}\delta = \left[\frac{X(726)_{unk}}{2*{}^{17}r_{PDB}*X(626)_{unk}} - 1\right] - [(1 + {}^{17}\delta)({}^{17}r_{PDB}(1 + {}^{17}\delta) + {}^{18}r_{PDB}(1 + {}^{18}\delta))] \qquad (26)$$

In this case Eqs. (25) and (26) can be iterated together substituting updated values for both ${}^{18}\delta$ and ${}^{17}\delta$.

The total $CO_2$ in dry air is given by

$$\mathbf{X}y_{CO2,unk} = \frac{\mathbf{X}y(626)_{unk} + \mathbf{X}y(636)_{unk} + \mathbf{X}y(826628)_{unk} + \mathbf{X}y(726627)_{unk}}{P(626)_{unk} + P(636)_{unk} + P(826628)_{unk} + P(726627)_{unk}}$$

(2667)

Dividing by the sum of the probabilities (fractional abundances P) corrects the sum of the measured isotopologues for the unmeasured rare isotopologues. The sum of the probabilities in Eq (26) which would be equal to 0.99993894 99993922 if the isotopic ratios are equal to the standard reference materials for carbon and oxygen. This would adds 0.024 µmol mol$^{-1}$ to the sum of the measured isotopologues $X_{CO2}$, assuming $X_{CO2}$ $y_{CO2}$ ~ 400 µmol mol$^{-1}$. This small difference accounts for the rare isotopologues with multiple isotopic substitutions that are not being measured. The correction in Eq. (2667) that applies for actual the unknowns will in general be *very* slightly different from 1- 0.000061060.99993932 (see above, the sum of the four major molecular abundances assuming VPDB values). We calculate actual P values for the unknown using Eqs. (1920)-(2223) with the δ-values from Eqs. (2324)-(2556)

and then use those in Eq. (2$\underline{66}$$\overline{7}$) instead of the standard reference material values to account for this small discrepancy, but it is not necessary in most cases.

**7 Analytical methods**

NOAA's new $CO_2$ calibration system is based on multiple laser spectroscopic techniques. It uses a combination of cavity ring-down spectroscopy (CRDS, Picarro, Inc. $CO_2/CH_4/H_2O$ analyzer, model number G2301) (O'Keefe and Deacon, 1988; Crosson, 2008), off-axis integrated cavity output spectroscopy (ICOS, Los Gatos Research, Inc., carbon dioxide isotope analyzer, CCIA-46-EP, model number 913-0033-0000) (Paul et al., 2001; Baer et al., 2002), and quantum cascade tunable infrared laser differential absorption spectroscopy (QC-TILDAS, Aerodyne Research, Inc., carbon dioxide isotope analyzer, model QCTILDAS-CS) (Tuzson et al., 2008; McManus et al., 2015).

The CRDS instrument measures a single absorption line from the $^{16}O^{12}C^{16}O$ isotopologue at 1603 nm (Crosson, 2008). For most of the data presented here, the instrument operated in an enhanced $CO_2$ mode where it did not measure $CH_4$ and instead focussed exclusively on the $CO_2$ absorption line with periodic measurements of $H_2O$ as a diagnostic. However, we have since determined this enhanced $CO_2$ mode does not improve the reproducibility of $CO_2$ measurements. We are currently testing the ability to do $CH_4$ calibrations at the same time as the $CO_2$ calibrations using the standard operating mode of the CRDS.

The ICOS and QC-TILDAS analyzers both measure absorption lines of $^{16}O^{12}C^{16}O$, $^{16}O^{13}C^{16}O$, and $\underline{^{18}O^{12}C^{16}O}$ $\underline{^{16}O^{12}C^{18}O}$ isotopologues individually (using lines $\sout{\text{in the}}$$\underline{\text{at}}$ 2309$\underline{0}$ $cm^{-1}$ $\sout{CO_2 \text{ absorption bands}}$). Both analyzers also measure the $\sout{^{17}O^{12}C^{16}O}$ $\underline{^{16}O^{12}C^{17}O}$ isotopologue but we cannot independently calibrate this measurement $\underline{\text{because our}}$ $\underline{\text{standards have not been measured for } \delta^{17}O. W}$$\sout{\text{so w}}$e assume that $\delta^{17}O$ follows the mass dependent fractionation relative to $\delta^{18}O$. $\underline{\text{Deviations from this relationship would be small and be insignificant when calculating total } CO_2.}$ The two analyzers have comparable performance and serve as backups for each other since only one is installed and used at a time. In the following discussion they are designated collectively as the $CO_2$ isotope analyzer. The $^{16}O^{12}C^{16}O$ measurement in the isotope analyzers uses a weak absorption line to match the measured absorption with the low abundance minor isotopologues. They are therefore not as precise as the measurement on the CRDS. The $\sout{\text{internal}}$ $^{16}O^{12}C^{16}O$ measurement from the isotope analyzer is not used to calculate total $CO_2$ but is used as $\underline{\boldsymbol{X}}\boldsymbol{y}$(626)$_{unk}$ in the calculation of $\delta^{13}C$ and $\delta^{18}O$ (see Eqs. (2$\underline{3}$$\overline{24}$) and (2$\underline{5}$$\overline{25}$) in the discussion above). Using this "internal" $\underline{\boldsymbol{X}}\boldsymbol{y}$(626)$_{unk}$ measurement gives slightly more precise $\delta^{13}C$ and $\delta^{18}O$ results than using the "external" $\underline{\boldsymbol{X}}\boldsymbol{y}$(626)$_{unk}$ measurement from the CRDS $\sout{\text{system}}$ $\underline{\text{instrument}}$ since it accounts for some instrument bias common to both the $^{16}O^{12}C^{16}O$ and the $^{16}O^{13}C^{16}O$ and $\sout{^{18}O^{12}C^{16}O}$ $\underline{^{16}O^{12}C^{18}O}$ isotopologue measurements. $\underline{\boldsymbol{X}}\boldsymbol{y}$(626)$_{unk}$ from the CRDS system is used in Eq. (2$\underline{7}$$\overline{26}$) to calculate total $CO_2$.

Figure 1 is a plumbing diagram for the $CO_2$ calibration system. The system uses the CRDS analyzer plus one of the $CO_2$ isotope analyzers. All measurements on the system are relative to a reference tank of compressed, unmodified natural air. A 4-port, 2-position switching valve (Valco Instruments Co, Inc. (VICI), model EUDA-24UWE) is used

to send sample/standard gas to one analyzer while the other analyzer simultaneously measures the reference tank. Sample and standard tanks are introduced to the system via two identical sample manifolds composed of 16-port multi-position selection valves (VICI, model EUTA-2CSD16MWE). A 4-port multi-position stream selection valve (VICI, model EUTA-2SD4MWE) is used to select either manifold A, B, or, optionally, for expansion to a third

5      manifold C. A plugged port on the manifold selection valve is used as a safe off port during shutdown. Sample/standard and reference gas pressures are controlled at 760 ±-1 Torr by two electronic pressure controllers with integrated mass flow meters (MKS Instruments, type 649B electronic pressure controller, model number 649B00813T13C2MR). The analyzers themselves control their internal cell pressures. However, controlling the inlet pressure prevents large inlet pressure swings due to inconsistent cylinder regulator set points and allows the internal

10    pressure control to be more consistent. All three instruments are continuous flow instruments so an idle gas is provided through a 3-way solenoid valve (Parker, model 009-0143-900) just upstream of the instrument inlet.  This idle gas is partially dried room air drawn through a Nafion drier (Perma Pure LLC.) for extended system idle time (e.g. on weekends) but is a cylinder of dried ambient natural air (dew point ~ -80 °C) for short idle times during and just prior to actual

15    calibrations. This cylinder ensures that the system downstream of the water traps does not get exposed to elevated levels of water vapor during short idle times between analyses. Each analyzer has a $H_2O$ trap up-stream of the inlet that normalizes any differences in water content among cylinders analyzed. These traps are 3.2 mm OD stainless steel tubing loops immersed in a -78 °C ethanol bath (SP Scientific Inc., MultiCool, model number MC480A). Both analyzers have individual sampling pumps to pull gas through the sample cell at partial vacuum. All tubing in the

20    system is 3.2 mm or 1.6 mm OD stainless steel.

The flow rates are set to 130 - 150 mL min$^{-1}$ by using a critical flow orifice downstream of the isotope analyzer cell or by partially closing the upstream solenoid valve in the CRDS instrument and relying on a stable pressure at the instrument inlet. The analysis sequence starts with a 4 minute flush of the sample/standard regulator (and sample/standard electronic pressure controller) and then alternates reference and sample through the two analyzers

25    for 8 cycles before moving to the next sample or standard. Each measurement cycle is 2.5 minutes of flushing and a 30 second signal average.

**8 Calibration and system performance**

Analyzers are calibrated approximately every two weeks in an offline calibration mode using a suite of 14 secondary

30    standards, covering the range 250 to 600 μmol mol$^{-1}$ total $CO_2$. The system is calibrated routinely to 600 μmol mol$^{-1}$ in expectation of a scale expansion in 2017. Each isotopologue is calibrated independently after decomposing the standard's total $CO_2$ into its component isotopologue mole fractions using the method discussed above. The secondary standards have assigned total $CO_2$ values by calibration against the entire set of primary standards (plus two additional standards that will extend the scale to 600 μmol mol$^{-1}$) in an analogous manner as described here.

35    This is a significant change from our previous NDIR calibration system where subsets of standards were used. It

makes the new calibration system less likely to have $CO_2$ dependent biases. The secondary standard's $\delta^{13}C$ and $\delta^{18}O$ values were assigned by IRMS measurement at INSTAAR. Primary standards also have $\delta^{13}C$ and $\delta^{18}O$ values assigned by INSTAAR, which we use when primary standards are used to calibrate secondary standards. The use of INSTAAR $\delta^{13}C$ and $\delta^{18}O$ assigned values on the secondary standards rather than the values from measurement versus the primary standards, shortens the traceability of the delta measurements to a true IRMS measurement. A comparison of the INSTAAR assignments with the NOAA measured isotopic values for the secondary standards is discussed below.

As mentioned in section 6, INSTAAR $\delta^{13}C$ and $\delta^{18}O$ measurements are relative to their own realization of the VPBD scales rather than on the WMO GAW scale for isotopic measurements of $CO_2$ (Jena Reference Air Set (JRAS-06) maintained by the Max Planck Institute for Biogeochemistry, Jena Germany) (Wendeberg et al., 2013). INSTAAR has scale contraction issues relative to JRAS. The relationships between INSTAAR and JRAS published by Wendeberg et al. (2013) indicate that while the offsets are significant for isotopic studies, the use of the INSTAAR realization for accounting for isotopic differences when determining total $CO_2$ will not add significant bias. When we use primary standards to calibrate secondary standards, the apportionment of the total $CO_2$ into component isotopologues will be slightly off. However, this is partially corrected when we recombine the resulting measured isotopologue mole fractions of the secondary standards into total $CO_2$. Using approximate JRAS values for our primary standards based on the Wendeberg et al. (2013) relationships, we see changes in the apportionment of the $^{16}O^{12}C^{16}O$ isotopologues on the order of 0.000 to 0.004 μmol mol$^{-1}$ with corresponding but opposite sign changes in the other isotopologues.

The instrument readings are absorption measurements corrected for cell pressure and temperature and converted into nominal mole fraction units. However, we treat them purely as an instrument response in arbitrary units. They could also be a voltage or a current. The responses from the analyzers are subsequently used in an offline calibration of each instrument. We do not use the internal calibration capabilities of the instruments; this ensures that the measurements are directly traceable to the WMO primary standards and can be reprocessed for future scale revisions. Each standard is measured relative to a reference cylinder to correct for slow drift of the analyzers. For the CRDS and ICOS analyzers the instrument response to each standard is divided by the average instrument response of the bracketing reference aliquots. For the QC-TILDAS, the difference between the response to the standard and the reference is used. In both cases we term the resulting values "response ratios". The choice of division vs subtraction is made due to the characteristics of the drift in each analyser. For example, the division operation does a better job when there is a slow span drift (perhaps due to variations in cell temperature and pressure) causing relative changes that are proportional to $X_{CO2}$ $CO_2$, whereas the difference operation is more appropriate when the majority of the drift is caused by a uniform shift in the output that does not depend on $X_{CO2}$ is not proportional to $CO_2$. Rather

than characterize the source of drift in each analyzer we use the reproducibility of target tank measurements to empirically determine which method gives more consistent results between calibration episodes.

The calibration curves are $CO_2$ isotopologue mole fractions as a function of response ratios. The CRDS instrument response is linear within the uncertainty of the standards (typical uncertainty of the primary standards is ±0.1 μmol mol$^{-1}$ 68 % CI). However, both isotope analyzers are slightly non-linear in their response and are fit with a quadratic polynomial. Non-linearity in the isotope analyzers may be partially due to incomplete flushing of the sample cell, caused by un-swept dead volumes, as the system switches from reference to standard. Memory of theResidual reference gas (ambient air from Niwot Ridge, ~400 μmol mol$^{-1}$ $CO_2$) in the sample cell influences the standards on the ends of the scale more than those close to the reference gas value potentially leading to a slight non-linear response. The difference in $^{16}O^{13}C^{16}O$ calibration curve residuals at 600 μmol/mol using a quadratic fit (0.0005 μmol/mol) and a linear fit (0.003 μmol/mol) indicate the memory effect is small in terms of total $CO_2$. Since all standards and all samples are treated identically and measured against the same reference gas, small memory effects should cancel out. Longer flushing times would reduce the memory effect but would decrease the lifetime of the standards.

Sample measurements are made relative to the same reference tank to account for drift in the analyzers between calibration episodes. The sample response ratios are used with the isotopologue specific calibration curves to determine isotopologue mole fractions for the sample cylinder which are combined into total $CO_2$, $\delta^{13}C$, and $\delta^{18}O$ values using the method discussed above. These values (total $CO_2$, $\delta^{13}C$, and $\delta^{18}O$) are stored in the NOAA database and are reported to the user via certificates and the web interface. Isotopologue specific mole fractions are not provided, however the equations described in this paper can be used to regenerate them.

Performance of the new calibration system has been evaluated over approximately one year by repeated measurements of target tanks (cylinders repeatedly measured as a diagnostic of system performance). Figure 2 shows the time series of total $CO_2$ measured for 4 target tanks with $CO_2$ ranging from 357 to 456 μmol mol$^{-1}$. Standard deviations of the measurements are approximately ±0.007 μmol mol$^{-1}$. Reproducibility of the target tanks close to the reference tank (typically ~ 400 μmol mol$^{-1}$ $CO_2$) are a little better than those farther out on the ends of the calibration range but the difference is small. While one year is not a long enough time series to fully quantify the reproducibility of the system, we estimate it to be ±0.01 μmol mol$^{-1}$ (68 % CI) based on these target tank measurements. This is a significant improvement over the NDIR system where reproducibility is ±0.03 μmol mol$^{-1}$ (68 % CI) ("Carbon Dioxide WMO Scale", 2017).

Prior to this new $CO_2$ calibration system, NOAA provided informational isotopic values for tertiary standards delivered to outside organizations by taking discrete samples from cylinders in flasks and having them measured by INSTAAR. This continued during the 6 months period when both calibration systems were run in parallel. Comparisons of these measurements with the isotopic results from the new calibration system are show in Figs. 3 and 4. The top plot in each figure is differences of measured delta values (NOAA – INSTAAR) vs INSTAAR values and the bottom plot in each figure is differences as a function of total $CO_2$ measured by NOAA. There is no

systematic bias between the NOAA and INSTAAR measurements for either species, except for highly depleted cylinders ($\delta^{13}C$ or $\delta^{18}O$ less than -20 ‰, shown by open symbols in both figures) and $\delta^{18}O$ in very high $CO_2$ cylinders (> 490 µmol mol$^{-1}$). The average offset (NOAA – INSTAAR) of non-depleted tanks is 0.0 ± 0.1 ‰ $\delta^{13}C$ and 0.0 ± 0.2 ‰ $\delta^{18}O$. The offset in the highly depleted cylinders most likely occurs as a result of the large

5   extrapolation in the INSTAAR IRMS measurements from the working standard at ambient $\delta^{13}C$ and $\delta^{18}O$. These offsets are roughly consistent with the INSTAAR JRAS offsets (Wendeberg et al., 2013) which are attributed to scale contraction issues at INSTAAR. The secondary standards used to routinely calibrate the NOAA system have isotopic assignments made by direct measurement by INSTAAR and are all relatively close to ambient (see Figs. 5 and 6) where the INSTAAR scale contraction is very small. By using these standards and calibrating our

10   measurements in mole fraction space we are not sensitive to the scale contraction issues in the INSTAAR measurement of depleted tanks. The $\delta^{18}O$ data do show a pronounced "hook" above ~ 490 µmol mol$^{-1}$. This is thought to be due to issues when sampling air from cylinders into flasks and not to the measurements either at INSTAAR or NOAA. A tertiary standard with 497 µmol mol$^{-1}$ $CO_2$ showed excellent agreement when measured directly by both NOAA ($\delta^{18}O$ = -8.92 ± 0.04 ‰) and  INSTAAR ($\delta^{18}O$ = -8.94 ± 0.1 ‰). A comparison can also

15   be made using the secondary standards which were calibrated directly by INSTAAR and by NOAA verses the primary standards. The assigned values of the secondary standards (as measured directly by INSTAAR) and the NOAA minus INSTAAR differences are shown in Figs. 5 and 6 for $\delta^{13}C$ and $\delta^{18}O$ respectively. Agreement is very good but there is a loss of precision on the NOAA calibration system near the wings of the $CO_2$ scale. NOAA measurements show some decrease in performance as total $CO_2$ moves away from the reference cylinder, which is

20   always an ambient $CO_2$ cylinder. However, even on the wings of the range the performance is more than adequate for the purpose of correcting total $CO_2$ for isotopic differences. The reproducibility of $\delta^{13}C$ (±-0.2 ‰, 68 % CI) and $\delta^{18}O$ (±-0.2 ‰, 68 % CI) measurements are again estimated from target tanks measurements. The uncertainty of the $\delta^{13}C$ and $\delta^{18}O$ measurements is dependent on the uncertainty of the total $CO_2$ values of the standards in addition to the reproducibility of the measurement system (Flores et al., 2017). This will be treated in an upcoming

25   publication describing the $CO_2$ scale revision (Hall et al., in preparation). The uncertainties of our measurement results for $\delta^{13}C$ and $\delta^{18}O$ are more than adequate for correcting atmospheric $CO_2$ measurements for standard vs sample isotopic differences. However, we caution against using them as standards for high precision $CO_2$ isotopic measurements.

[revised manuscript text omitted]

**Appendix**

We will derive expressions for Δ in terms of conventional δ values because we currently supply standards to users within the greenhouse gas measurement community with their δ values as information in addition to the total $x_{CO2}$ calibration.

$$^{13}\Delta = \frac{^{13}Rx}{^{13}Rx_{\mathrm{VPDB}}} - 1 = \frac{^{13}r}{^{13}r_{\mathrm{VPDB}}}\frac{1 + {}^{13}r_{\mathrm{VPDB}}}{1 + {}^{13}r} - 1 = (1 + {}^{13}\delta)(1 + {}^{13}r_{\mathrm{VPDB}})(1 - {}^{13}r + {}^{13}r^2) - 1$$

Where we have used the first 3 terms of the series expansion $(1+r)^{-1} = 1 - r + r^2 - r^3 + \ldots$ and the definitions of $r$, $Rx$, δ, and Δ. Expanding,

$$^{13}\Delta = (1 - {}^{13}r + {}^{13}r^2) + {}^{13}r_{\mathrm{VPDB}}(1 - {}^{13}r + {}^{13}r^2) +$$
$$^{13}\delta(1 - {}^{13}r + {}^{13}r^2) + {}^{13}\delta\,{}^{13}r_{\mathrm{VPDB}}(1 - {}^{13}r + {}^{13}r^2) - 1$$

and rearranging, we get

$$^{13}\Delta = (-{}^{13}r + {}^{13}r^2) + ({}^{13}r_{\mathrm{VPDB}} + {}^{13}\delta)(1 - {}^{13}r + {}^{13}r^2) + {}^{13}\delta\,{}^{13}r_{\mathrm{VPDB}}(1 - {}^{13}r + {}^{13}r^2)$$

Rearranging further,

$$^{13}\Delta = {}^{13}\delta - ({}^{13}r - {}^{13}r_{\mathrm{VPDB}}) + {}^{13}r({}^{13}r - {}^{13}r_{\mathrm{VPDB}}) - {}^{13}\delta({}^{13}r - {}^{13}r_{\mathrm{VPDB}}) - {}^{13}\delta\,{}^{13}r_{\mathrm{VPDB}}\,{}^{13}r$$

Then, using Eq. (5),

$$^{13}\Delta = {}^{13}\delta - {}^{13}r_{\mathrm{VPDB}}\,{}^{13}\delta + {}^{13}r\,{}^{13}r_{\mathrm{VPDB}}\,{}^{13}\delta - {}^{13}\delta\,{}^{13}r_{\mathrm{VPDB}}\,{}^{13}\delta - {}^{13}\delta\,{}^{13}r_{\mathrm{VPDB}}\,{}^{13}r$$

The third and the last term cancel, and then keeping only the two leading terms, we obtain

$$^{13}\Delta = {}^{13}\delta(1 - {}^{13}r_{\mathrm{VPDB}}) \qquad (A1)$$

Equation (A1) is an excellent approximation. Using the values for $^{13}r_{\mathrm{VPDB}}$ in Table 1 and assuming that $^{13}\delta = -0.00800$ (−8.00 permil, an approximate value for $CO_2$-in-air) we calculate both $^{13}Rx$ for the air sample and $^{13}Rx_{\mathrm{VPDB}}$, and using the definition (Eq. (6)) for $^{13}\Delta$, we obtain $^{13}\Delta = -0.00791202$. Equation (A1) gives us $-0.00791064$.

A very similar derivation holds for $^{17}\Delta$ and $^{18}\Delta$ but it is a bit more complicated because the terms for $^{17}Rx$ and $^{18}Rx$ get mixed.

$$^{17}\Delta = \frac{^{17}r}{^{17}r_{\mathrm{VPDB}}}\frac{1 + {}^{17}r + {}^{18}r}{1 + {}^{17}r_{\mathrm{VPDB}} + {}^{18}r_{\mathrm{VPDB}}} - 1 = \frac{^{17}r}{^{17}r_{\mathrm{VPDB}}}\frac{1 + {}^{78}r}{1 + {}^{78}r_{\mathrm{VPDB}}} - 1$$

To keep the notation simpler and stressing the analogy with the derivation for $^{13}\Delta$ we have written in the above $^{78}r = {}^{17}r + {}^{18}r$ for the air sample and $^{78}r_{\mathrm{VPDB}} = {}^{17}r_{\mathrm{VPDB}} + {}^{18}r_{\mathrm{VPDB}}$ for the standard.

After keeping only the leading terms we have

$$^{17}\Delta = {}^{17}\delta - ({}^{78}r - {}^{78}r_{\mathrm{VPDB}}) = {}^{17}\delta - {}^{17}r_{\mathrm{VPDB}}\,{}^{17}\delta - {}^{18}r_{\mathrm{VPDB}}\,{}^{18}\delta$$
(A2)

And similarly for $^{18}O$:

$$^{18}\Delta = {}^{18}\delta - {}^{17}r_{\text{VPDB}}{}^{17}\delta - {}^{18}r_{\text{VPDB}}{}^{18}\delta \qquad\qquad —(A3)$$

These are the equivalents of Eq. (A1) for $^{13}\Delta$. Because $^{17}r$ and $^{18}r$ are significantly smaller than $^{13}r$, we approximate further $^{17}\Delta \cong {}^{17}\delta$ and $^{18}\Delta \cong {}^{18}\delta$. Since $^{17}\delta$ is not usually measured and also is often very closely related as $^{17}\delta = 0.53$

5    $^{18}\delta$, we can write for the oxygen correction terms in Eq. (16) $\frac{^{17}r_{\text{PDB}}}{^{17}r_{\text{VPDB}}}{}^{17}\delta + \frac{^{18}r_{\text{PDB}}}{^{18}r_{\text{VPDB}}}{}^{18}\delta = (0.53$ $\frac{^{17}r_{\text{PDB}}}{^{17}r_{\text{VPDB}}} + \frac{^{18}r_{\text{PDB}}}{^{18}r_{\text{VPDB}}}){}^{18}\delta$, and filling in the $r_{\text{PVPDB}}$ values from Table 1,

$0.53*395*10^{-6}*{}^{18}\delta + 2088*10^{-6}*{}^{18}\delta = 2297*10^{-6}*{}^{18}\delta \cong 0.0023*{}^{18}\delta$

Now we return to Eq. (18) in the main text, applicable when the isotopic composition of measured air is different from the standard that is used. We restate it as

10   $\Delta \cancel{X} y_{\text{totCO2}} = \cancel{X} y_{\text{totCO2}} \left[0.0110\underline{06}\cancel{11}\left({}^{13}\delta_{\text{air}} - {}^{13}\delta_{\text{st}}\right) + 2*0.0023\left({}^{18}\delta_{\text{air}} - {}^{18}\delta_{\text{st}}\right)\right]$
     (A4)

**Supplemental links**

None

**Author contribution**

P. Tans derived the equations used to apportion total $CO_2$ into component isotopologues and their application. A. Crotwell designed and built the calibration system. K. Thoning wrote software for operating the calibration system and managing the data.

**Competing interests**

The authors declare that they have no conflict of interest.

**Disclaimer**

None

**Acknowledgements**

We thank Thomas Mefford for running the calibration system, Duane Kitzis for standards preparation, and Ed Dlugokencky and three anonymous reviewers for reviewing this manuscript. In addition, we thank Sylvia Michel, Natalie Cristo, and the other members of the INSTAAR Stable Isotope Laboratory for isotopic measurements.

[revised manuscript text omitted]